# Double Descent Meets Out-of-Distribution Detection: Theoretical Insights and Empirical Analysis on the role of model complexity

## Abstract

While overparameterization is known to benefit generalization, its impact on Out-Of-Distribution (OOD) detection is less understood. This paper investigates the influence of model complexity in OOD detection. We propose an expected OOD risk metric to evaluate classifiers confidence on both training and OOD samples. Leveraging Random Matrix Theory, we derive bounds for the expected OOD risk of binary least-squares classifiers applied to Gaussian data. We show that the OOD risk depicts an infinite peak, when the number of parameters is equal to the number of samples, which we associate with the double descent phenomenon. Our experimental study on different OOD detection methods across multiple neural architectures extends our theoretical insights and highlights a double descent curve. Our observations suggest that overparameterization does not necessarily lead to better OOD detection. Using the Neural Collapse framework, we provide insights to better understand this behavior. To facilitate reproducibility, our code will be made publicly available upon publication.

## 1 Introduction

In recent years, large neural networks have seen increased use in Machine Learning due to their impressive generalization properties (Brown, 2020; Dubey et al., 2024). While empirical evidence suggests that rich machine learning systems obtain near-optimal generalization results when trained to interpolate training data (Zhang et al., 2021), the classical bias-variance trade-off theory (Geman et al., 1992) suggests that such models overfit and generalize poorly. Indeed, the classical literature describes the generalization error with respect to the model complexity as a U-shaped curve and suggests finding a model between underfitting and overfitting, *i.e.*, a model rich enough to express underlying structure in the data and simple enough to avoid fitting spurious patterns. To bridge the gap between the classical theory and the modern practice, Belkin et al. (2019) introduced the concept of "double descent" within a unified generalization error curve. In this setting, for "small" model complexities, the generalization error curve exhibits the U-shaped curve described by the bias-variance trade-off. However, when the model complexity is higher than the interpolation threshold, *i.e.*, when the model is rich enough to fit the training data, increasing the model complexity leads to a second decrease in the generalization error. A popular intuitive explanation of this phenomenon is that by considering large model complexities that contain more candidate predictors compatible with the training data, we are also able to find interpolating functions that are "simpler" and are smoother to follow a form of Occam's razor (Belkin et al., 2019).

Although the double descent phenomenon provides valuable insights to understand generalization of rich models on unseen data, its understanding on Out-Of-Distribution (OOD) detection has received less attention. OOD detection addresses a distinct challenge in deep neural networks (DNNs): their tendency to make high-confidence predictions, even for inputs that differ significantly from the training data. While generalization focuses on the model's ability to classify data that has shifted, OOD detection emphasizes the model's capacity to recognize when a shift is too large and refrain from confident predictions. In real-world applications, reliable OOD detection is crucial to ensuring the safety and reliability of AI systems. This includes fields such as healthcare (Schlegl et al., 2017), industrial inspection (Paul Bergmann & Stege, 2019), and autonomous driving (Kitt et al., 2010).

Therefore, understanding the influence of the model complexity on OOD detection is crucial for model selection in critical tasks.

While overparameterization is known to benefit generalization, its effects on OOD detection still remain limited. In this work, we investigate the role of the model complexity in OOD detection. In particular, we highlight a double descent phenomenon similar to the one observed for generalization error. To the best of our knowledge, the double descent phenomenon has never been observed in OOD detection. The results of our theoretical and empirical analyses are outlined below.

**Contributions.** We make the following contributions, taking a step towards a better theoretical and empirical understanding of the influence of model complexity on OOD detection:

1. We propose an expected OOD risk metric to evaluate classifiers confidence on both training and OOD samples.

2. Using Random Matrix Theory, we derive bounds for both the expected risk and OOD risk of binary least-squares classifiers applied to Gaussian data with respect to the model complexity. We show that both risks exhibit an infinite peak, when the number of parameters is equal to the number of samples, which we associate with the double descent phenomenon.

3. We empirically observe a double descent phenomenon curve in various OOD detection methods and across multiple neural architectures, including transformer-based (ViT, Swin) and convolutional-based models (ResNet, CNN).

4. We also observe that OOD detection in the overparametrized regime is not guaranteed to be better than in the underparametrized regime. Using the Neural Collapse framework Papyan et al. (2020), we propose to better explain this architecture-dependant improvement on OOD detection with overparametrization.

To facilitate reproducibility, our code will be made publicly available upon publication.

## 2 RELATED WORK

**OOD Detection.** The OOD detection research focuses on two primary directions: supervised and unsupervised approaches. We will focus on latter ones, also called post-hoc methods. These can be categorized based on the essential feature used for the scoring function. First, the logit- or confidence-based methods leverage network logits to derive a confidence measure used as an OOD scoring metric (Hendrycks & Gimpel, 2017; DeVries & Taylor, 2018; Liu et al., 2020; Huang & Li, 2021b; Hendrycks et al., 2022). A common baseline for this methods is the softmax score (Hendrycks & Gimpel, 2017), which simply uses the model softmax prediction as the OOD score. Then, the Energy (Liu et al., 2020) elaborates on it by computing the the LogSumExp on the logits, thus offering empirical and theoretical advantages over the Softmax confidence score. Second, feature-based and hybrid methods (Lee et al., 2018b; Wang et al., 2022; Sun et al., 2022; Ming et al., 2023; Djurisic et al., 2023; Ammar et al., 2024) exploit the model's final representation to derive the scoring function. Mahalanobis (Lee et al., 2018b) estimates density on ID training samples using a mixture of class-conditional Gaussians based on the feature distributions. NECO (Ammar et al., 2024), on the contrary, leverages the geometric properties of Neural Collapse to construct a scoring function based on the relative norm of a sample within the subspace defined by the Simplex Equiangular Tight Frame (ETF) formed by the ID data. Hybrid methods are characterized by the fact that they can be augmented by using the logits as weighting factors on the scoring metrics defined on the features.

**Double Descent.** The double descent risk curve was introduced by Belkin et al. (2019) to explain the good performance observed in practice by overparameterized models (Zhang et al., 2021; Belkin et al., 2018; Nakkiran et al., 2021) and to bridge the gap between the classical bias-variance trade-off theory and modern practices. Theoretical investigation into this phenomenon mainly focuses on various linear models in both regression and classification problems through the Random Matrix Theory (Louart et al., 2018; Liao et al., 2020; Jacot et al., 2020; Derezinski et al., 2020; Kini & Thrampoulidis, 2020; Mei & Montanari, 2022; Deng et al., 2022; Bach, 2024; Brellmann et al., 2024), techniques from statistical mechanics (d'Ascoli et al., 2020; Canatar et al., 2021), the VC theory (Lee & Cherkassky, 2022; Cherkassky & Lee, 2024), or novel bias-variance decomposition

in deep neural networks (Yang et al., 2020). The double descent phenomenon has also been observed in experiments with popular neural network architectures (Belkin et al., 2019; Nakkiran et al., 2021). In addition to depending on the model complexity, the double descent phenomenon also depends on other dimensions such as the level of regularization (Liao et al., 2020; Mei & Montanari, 2022), the number of epochs (Nakkiran et al., 2021; Stephenson & Lee, 2021; Olmin & Lindsten, 2024), or the data eigen-profile (Liu et al., 2021a). Finally, the theoretical background on double descent and benign overparametrization developed by Bartlett et al. (2020) inspired subsequent works that focused on generalization under dataset shifts (Tripuraneni et al., 2021; Hao et al., 2024; Kausik et al., 2024; Hao & Zhang, 2024). It is important to note that these dataset shifts concern scenarios where the model can generalize, *e.g.*, the class labels are the same as in the training. This studies do not address the issue of presenting the model with samples that differ significantly beyond the point of generalization, *i.e.*, OOD detection.

## 3 PRELIMINARIES

**Notations.** For a real vector $\boldsymbol{v}$, we denote by $\|\boldsymbol{v}\|_2$ the euclidean norm of $\boldsymbol{v}$. When the matrix $\boldsymbol{A}$ is full rank, we denote by $\boldsymbol{A}^+$ the Moore-Penrose inverse of $\boldsymbol{A}$. We depict by $[d] := \{1, \ldots, d\}$ the set of the $d$ first natural integers. For a subset $\mathcal{T} \subseteq [d]$, we denote by $\mathcal{T}^c := [d] \setminus \mathcal{T}$ its complement set. For a subset $\mathcal{T} \subseteq [d]$, a $d$-dimensional vector $\boldsymbol{v} \in \mathbb{R}^d$ and a $n \times d$ matrix $\boldsymbol{A} = \left[ \boldsymbol{a}^{(1)} | \ldots | \boldsymbol{a}^{(n)} \right]^T \in \mathbb{R}^{n \times d}$, we use $\boldsymbol{v}_{\mathcal{T}} = [v_j : j \in \mathcal{T}]$ to denote its $|\mathcal{T}|$-dimensional subvector of entries from $\mathcal{T}$ and $\boldsymbol{A}_{\mathcal{T}} = \left[ \boldsymbol{a}_{\mathcal{T}}^{(1)} | \ldots | \boldsymbol{a}_{\mathcal{T}}^{(n)} \right]^T$ to denote the $n \times |\mathcal{T}|$ design matrix with variables from $\mathcal{T}$. We use $\lambda_{\min}(\boldsymbol{A})$ and $\lambda_{\max}(\boldsymbol{A})$ to depict the minimum and maximum eigenvalues of $\boldsymbol{A}$, respectively. $\mathcal{N}(0, \boldsymbol{I}_d)$ denotes the standard multivariate Gaussian distribution of $d$ random variables.

**Supervised Learning Problems.** In supervised learning, we employ training dataset $\mathcal{D} := \left\{ (\boldsymbol{x}_1, y_1), \cdots, (\boldsymbol{x}_n, y_n) \right\}$ of $n$ independent and identically distributed (i.i.d.) samples drawn from an unknown distribution $P_{\mathcal{X}, \mathcal{Y}}$ over $\mathcal{X} \times \mathcal{Y}$. Using samples from the dataset $\mathcal{D}$, the objective is to find a predictor $\hat{f} : \mathcal{X} \to \mathcal{Y}$ among a class of functions $\mathcal{F}$ to predict the target $y \in \mathcal{Y}$ of a new sample $\boldsymbol{x} \in \mathcal{X}$. In particular, given a loss function $\ell : \mathcal{Y} \times \mathcal{Y} \to \mathbb{R}$, the objective is to minimize the expected risk (or loss) defined, for all $\hat{f} \in \mathcal{F}$, as:

$$R(\hat{f}) = \mathbb{E}_{(\boldsymbol{x}, y) \sim P_{\mathcal{X}, \mathcal{Y}}} \left[ \ell \big( \hat{f}(\boldsymbol{x}), y \big) \right] = \int_{\mathcal{X} \times \mathcal{Y}} \ell \big( \hat{f}(\boldsymbol{x}), y \big) dP_{\mathcal{X}, \mathcal{Y}}(\boldsymbol{x}, y). \tag{1}$$

Typically, we choose the mean-squared loss $\ell(\hat{f}(\boldsymbol{x}), y) = \big( \hat{f}(\boldsymbol{x}) - y \big)^2$ for regression problems or the zero-one loss $\ell(\hat{f}(\boldsymbol{x}), y) = \mathbf{1}_{\hat{f}(\boldsymbol{x}) \neq y}$ for classification problems. We denote the optimal predictor by $f^* := \arg\min_{f \in \mathcal{F}} R(f)$. Since the distribution $P_{\mathcal{X}, \mathcal{Y}}$ is unknown in practice, we instead try to minimize an empirical version of the expected risk based on the dataset $\mathcal{D} := \left\{ (\boldsymbol{x}_i, y_i) \right\}_{i=1}^n$ and defined as

$$R_{emp}(\hat{f}) = \tfrac{1}{n} \sum_{i=1}^n \ell \big( \hat{f}(\boldsymbol{x}_i), y_i \big). \tag{2}$$

**Out-of-Distribution Detection.** In machine learning problems, we usually assume that the test data distribution is similar to the training data distribution (the closed-world assumption). As this is not the case in real-world applications, the Out-of-Distribution (OOD) detection aims to flag inputs that significantly deviate from the training data to prevent unreliable predictions. In the following, we denote by $P_{\mathcal{X}, \mathcal{Y}}^{\text{OOD}}$ a distribution over $\mathcal{X} \times \mathcal{Y}$ that differs from the training distribution $P_{\mathcal{X}, \mathcal{Y}}$. OOD data typically involve a semantic shift and represent concepts or labels not seen during training. A popular class of OOD detection techniques relies on the definition of a scoring function $s(\cdot \, ; \hat{f})$, which uses the probability predictions of the classifier $\hat{f}(\cdot)$ as scores to flag an instance $\boldsymbol{x}$ as OOD when the score $s(\boldsymbol{x}; \hat{f})$ is below a certain threshold $\lambda$. A common approach is to use the Maximum Softmax Probability that returns the higher softmax probabilities of the predictor $\hat{f}(\cdot)$ as a scoring function to measure the prediction confidence.

# 4 DOUBLE DESCENT FOR THE BINARY CLASSIFICATION IN GAUSSIAN COVARIATE MODEL

In this section, we introduce the expected OOD risk metric and we present our main theoretical results on binary least-squares classifiers applied to Gaussian data. We assume that $\mathcal{X} \subseteq \mathbb{R}^d$ and $\mathcal{Y} := [0, 1]$. Let $\phi : \mathbb{R} \to \mathcal{Y}$ be a mapping, we denote by $\mathcal{F}_d := \{ f : \mathcal{X} \to \mathcal{Y}, \boldsymbol{x} \mapsto \phi(\boldsymbol{x}^T \boldsymbol{w}) \mid \boldsymbol{w} \in \mathbb{R}^d \}$ the class of functions considered in this study.

## 4.1 SYSTEM MODEL

**Gaussian Covariate Model & Binary Classification.** We assume we have a training dataset $\mathcal{D} := \{ (\boldsymbol{x}_i, \boldsymbol{y}_i) \}_{i=1}^n$ of $n$ i.i.d samples drawn from a Gaussian covariate model, *i.e.*, from a distribution $P_{\mathcal{X}, \mathcal{Y}}$ over $\mathcal{X} \times \mathcal{Y}$; where $\boldsymbol{x}_i \sim \mathcal{N}(0, \boldsymbol{I}_d)$ and $\boldsymbol{y}_i$ is a noisy response of $\boldsymbol{x}_i$ with respect to the function $f^* : \boldsymbol{x} \mapsto \phi(\boldsymbol{x}^T \boldsymbol{w}^*)$ that is defined as

$$\boldsymbol{y}_i = f^*(\boldsymbol{x}_i) + \boldsymbol{\epsilon}_i = \phi(\boldsymbol{x}_i^T \boldsymbol{w}^*) + \boldsymbol{\epsilon}_i,$$

with $\boldsymbol{\epsilon}_i \sim \mathcal{N}(0, \sigma^2)$ and $\sigma > 0$. The objective is to find a classifier $\hat{f}(\cdot) \in \mathcal{F}_d$ that fits $f^*(\cdot)$. Without loss of generality, this problem can be interpreted as a binary classification problem where $f^*(\cdot)$ returns a probability. Let $\boldsymbol{X} = [\boldsymbol{x}_1, \ldots, \boldsymbol{x}_n]^T \in \mathbb{R}^{n \times d}$ be the data matrix containing the $n$ samples $\boldsymbol{x}_i \in \mathbb{R}^d$ as column vectors and $\boldsymbol{y} = [\boldsymbol{y}_1, \ldots, \boldsymbol{y}_n]^T \in \mathbb{R}^n$ be the target vector of probabilities. Given the loss function $\ell : (\hat{y}, y) \mapsto (\hat{y} - y)^2$, in order to find $\boldsymbol{w}^*$ (and thus $f^*(\cdot)$), we want to minimize the empirical risk $R_{emp} : \mathcal{F}_d \to \mathbb{R}$ defined in equation 2 as

$$R_{emp}(\hat{f}) = \tfrac{1}{n} \sum_{i=1}^n \ell(\hat{f}(\boldsymbol{x}_i), \boldsymbol{y}_i) = \tfrac{1}{n} \sum_{i=1}^n (\phi(\boldsymbol{x}_i^T \boldsymbol{w}) - \boldsymbol{y}_i)^2 = \tfrac{1}{n} \| \phi(\boldsymbol{X} \boldsymbol{w}) - \boldsymbol{y} \|_2^2. \tag{3}$$

**Least-Squares Binary Classifiers.** To analytically solve equation 3, we assume that $n \ll d$ and that the data matrix $\boldsymbol{X}$ is full row rank. We consider a particular selection of classifiers $\hat{\mathcal{F}}_d := \{ \hat{f}_{\mathcal{T}} : \boldsymbol{x} \mapsto \phi(\boldsymbol{x}^T \hat{\boldsymbol{w}}) \in \mathcal{F}_d \mid \mathcal{T} \subseteq [d] \}$, in which $\hat{f}_{\mathcal{T}} \in \hat{\mathcal{F}}_d$ uses a subset $\mathcal{T} \subseteq [d]$ of features that fits coefficients $\hat{\boldsymbol{w}} \in \mathbb{R}^d$ as

$$\hat{\boldsymbol{w}}_{\mathcal{T}} = \boldsymbol{X}_{\mathcal{T}}^+ \boldsymbol{y} \quad \text{and} \quad \hat{\boldsymbol{w}}_{\mathcal{T}^c} = \boldsymbol{0}. \tag{4}$$

**Out-of-Distribution Risk.** To measure the ability of binary classifiers $\hat{f}(\cdot) \in \hat{\mathcal{F}}_d$ to provide prediction confidence on samples drawn from both the training distribution $P_{\mathcal{X}, \mathcal{Y}}$ and the OOD distribution $P_{\mathcal{X}, \mathcal{Y}}^{\text{OOD}}$, we introduce an OOD risk function similar to the expected risk defined in equation 1. Let $f_{\text{OOD}}^* : \mathcal{X} \to \mathcal{Y}$ be the mapping chosen from $\mathcal{F}_d$ such that $f_{\text{OOD}}^*(\boldsymbol{x})$ is close to $0.5$ when the sample $\boldsymbol{x}$ is more likely drawn from the $P_{\mathcal{X}, \mathcal{Y}}^{\text{OOD}}$ and close to $f^*(\boldsymbol{x})$ when $\boldsymbol{x}$ is more likely drawn from $P_{\mathcal{X}, \mathcal{Y}}$. We define the noisy response $z(\boldsymbol{x})$ to a given sample $\boldsymbol{x} \in \mathcal{X}$ for the mapping $f_{\text{OOD}}^*(\cdot)$ as:

$$z(\boldsymbol{x}) = 2 f_{\text{OOD}}^*(\boldsymbol{x}) - 1 + \epsilon' = 2 \phi(\boldsymbol{x}^T \boldsymbol{w}_{\text{OOD}}^*) - 1 + \epsilon', \tag{5}$$

where $\epsilon' \sim \mathcal{N}(0, \sigma')$ and $\sigma' > 0$. To measure whether prediction confidences of a binary classifier $\hat{f}(\cdot)$ can be used for defining an OOD scoring function, we define the Out-of-Distribution Risk $R_{\text{OOD}} : \mathcal{F}_d \to \mathbb{R}$ as:

$$R_{\text{OOD}}(\hat{f}) = \mathbb{E}_{(\boldsymbol{x}, \cdot) \sim P_{\mathcal{X}, \mathcal{Y}}} \big[ (2 \hat{f}(\boldsymbol{x}) - 1 - z(\boldsymbol{x}))^2 \big] + \mathbb{E}_{(\boldsymbol{x}, \cdot) \sim P_{\mathcal{X}, \mathcal{Y}}^{\text{OOD}}} \big[ (2 \hat{f}(\boldsymbol{x}) - 1 - z(\boldsymbol{x}))^2 \big], \tag{6}$$

which depicts the expected risk of the predictor $2 \hat{f}(\cdot) - 1$ on the loss function $\ell : (\hat{y}, y) \mapsto (\hat{y} - y)^2$ and distributions $P_{\mathcal{X}, \mathcal{Y}}$ and $P_{\mathcal{X}, \mathcal{Y}}^{\text{OOD}}$.

**Remark 4.1.** From equation 5, we have $z(\boldsymbol{x}) \approx \pm 1 + \epsilon'$ if $\boldsymbol{x} \in P_{\mathcal{X}, \mathcal{Y}}$ and $z(\boldsymbol{x}) \approx \epsilon'$ if $\boldsymbol{x} \in P_{\mathcal{X}, \mathcal{Y}}^{\text{OOD}}$. A low value for $R_{\text{OOD}}(\hat{f})$ indicates thus two aspects: $(i)$ the classifier $\hat{f}(\cdot)$ is confident on predictions over the training distribution $P_{\mathcal{X}, \mathcal{Y}}$, which corresponds to a low $\mathbb{E}_{(\boldsymbol{x}, \cdot) \sim P_{\mathcal{X}, \mathcal{Y}}} \big[ (2 \hat{f}(\boldsymbol{x}) - 1 - z(\boldsymbol{x}))^2 \big]$; and/or $(ii)$ the classifier $\hat{f}(\cdot)$ is not confident on predictions over the distribution $P_{\mathcal{X}, \mathcal{Y}}^{\text{OOD}}$, which is

reflected by a low $\mathbb{E}_{(\boldsymbol{x},\cdot)\sim P_{\mathcal{X},\mathcal{Y}}^{\text{OOD}}}\left[(2\hat{f}(\boldsymbol{x})-1-z(\boldsymbol{x}))^2\right]$. In particular, $R_{\text{OOD}}(\hat{f})$ is minimized when logits are maximally confident on ID samples (only one logit is non-zero) and uniformly distributed on OOD samples.

**Remark 4.2.** Note that the OOD risk function $R_{\text{OOD}}(\cdot)$ defined in equation 6 can be extended to multi-class classifiers $\hat{f}(\cdot)$ using the softmax function as

$$R_{\text{OOD}}(\hat{f}) = \mathbb{E}_{(\boldsymbol{x},\cdot)\sim P_{\mathcal{X},\mathcal{Y}}}\left[(\|\hat{f}(\boldsymbol{x})\|_\infty - z(\boldsymbol{x}))^2\right] + \mathbb{E}_{(\boldsymbol{x},\cdot)\sim P_{\mathcal{X},\mathcal{Y}}^{\text{OOD}}}\left[(\|\hat{f}(\boldsymbol{x})\|_\infty - z(\boldsymbol{x}))^2\right],$$

where $z(\boldsymbol{x}) \approx 1 + \epsilon'$ if $\boldsymbol{x} \in P_{\mathcal{X},\mathcal{Y}}$ and $z(\boldsymbol{x}) \approx \frac{1}{C} + \epsilon'$ if $\boldsymbol{x} \in P_{\mathcal{X},\mathcal{Y}}^{\text{OOD}}$, $C$ depicts the number of classes, and $\|\cdot\|_\infty$ denotes the infinity norm.

In order to use the Random Matrix Theory, we make the following assumption on the activation function $\phi : \mathbb{R} \to \mathcal{Y}$.

**Assumption 4.1.** The activation function $\Phi(\cdot)$ is strictly monotonically non-decreasing and its derivative Lipschitz continous.

**Remark 4.3.** This assumption holds for many of the activation functions traditionally considered in neural networks, such as sigmoid functions.

## 4.2 PREDICTION RISK

Leveraging the Random Matrix Theory and following the same line of arguments of Theorem 1 in Belkin et al. (2020), we derive bounds for the risk of the subset of classifiers defined in $\hat{\mathcal{F}}_d$ with equation 4 (see proof in Appendix A.1).

**Theorem 1.** Let $(p,q) \in [\![1,d]\!]^2$ such that $p + q = d$, $\mathcal{T} \subseteq [d]$, be an arbitrary subset of the $d$ first natural integers, and $\mathcal{T}^c := [d] \setminus \mathcal{T}$ its complement set. Let $\hat{\boldsymbol{w}} \in \mathbb{R}^d$ such that $\hat{\boldsymbol{w}}_\mathcal{T} = \boldsymbol{X}_\mathcal{T}^+ \boldsymbol{y} \in \mathbb{R}^p$ and $\hat{\boldsymbol{w}}_{\mathcal{T}^c} = \boldsymbol{0} \in \mathbb{R}^q$. Then the expected risk with respect to the loss function $\ell : (\hat{y},y) \mapsto (\hat{y}-y)^2$ of the predictor $\hat{f}_\mathcal{T} : \boldsymbol{x} \mapsto \phi(\boldsymbol{x}^T\hat{\boldsymbol{w}}) \in \hat{\mathcal{F}}_d$ satisfies

$$\lambda_{\min}(\boldsymbol{\Sigma})c(n,p,\sigma) + \sigma^2 \le \mathbb{E}_{\boldsymbol{X}}\left[R(\hat{f}_\mathcal{T})\right] \le \lambda_{\max}(\boldsymbol{\Sigma})c(n,p,\sigma) + \sigma^2,$$

*where*

$$c(n,p,\sigma) = \begin{cases} \frac{n}{n-p-1}\left(\|\boldsymbol{w}_{\mathcal{T}^c}^*\|_2^2 + \sigma^2\right) + \|\boldsymbol{w}_{\mathcal{T}^c}^*\|_2^2 & \text{if } p \le n-2, \\ +\infty & \text{if } n-1 \le p \le n+1, \\ \left(1-\frac{n}{p}\right)\|\boldsymbol{w}_\mathcal{T}^*\|_2^2 + \frac{n}{p-n-1}\left(\|\boldsymbol{w}_{\mathcal{T}^c}^*\|_2^2 + \sigma^2\right) + \|\boldsymbol{w}_{\mathcal{T}^c}^*\|_2^2 & \text{if } p \ge n+2, \end{cases} \tag{7}$$

*and*

$$\boldsymbol{\Sigma} = \mathbb{E}_{(\boldsymbol{x},\cdot)\sim P_{\mathcal{X},\mathcal{Y}}}\left[\left(\frac{\Phi(\boldsymbol{x}^T\hat{\boldsymbol{w}})-\Phi(\boldsymbol{x}^T\boldsymbol{w}^*)}{\boldsymbol{x}^T\hat{\boldsymbol{w}}-\boldsymbol{x}^T\boldsymbol{w}^*}\right)^2\boldsymbol{x}\boldsymbol{x}^T\right]. \tag{8}$$

**Remark 4.4.** From Lemma 4.1, the matrix $\boldsymbol{\Sigma}$ defined in equation 8 is nonsingular and positive-definite. Note that Theorem 1 in Belkin et al. (2020) constitutes a special case of Theorem 1 for $\phi : \boldsymbol{x} \mapsto \boldsymbol{x}$, which corresponds to the case where $\boldsymbol{\Sigma} = \boldsymbol{I}_d$.

**Remark 4.5.** From Theorem 1, we have $\mathbb{E}_{\boldsymbol{X}}\left[R(\hat{f}_\mathcal{T})\right] = \infty$ around $p = n$. The expected risk decreases again as $p$ increases beyond $n$ and highlights a double descent phenomenon. This result is consistent with the literature of double descent (Mei & Montanari, 2022; Louart et al., 2018; Liao et al., 2020; Bach, 2024), which identifies the ratio $p/n$ as the model complexity of a linear model to describe an under-($p/n < 1$) and an over-($p/n > 1$) parameterized regimes for the expected risk with a phase transition around $p/n = 1$ characterized by a peak.

## 4.3 OUT-OF-DISTRIBUTION RISK

Using a similar approach to that Theorem 1, we obtain the following result on the subset of classifiers defined in $\hat{\mathcal{F}}_d$ with equation 4 (see proof in Appendix A.2).

**Theorem 2.** *Let $(p, q) \in [\![1, d]\!]^2$ such that $p + q = d$, $\mathcal{T} \subseteq [d]$, be an arbitrary subset of the $d$ first natural integers, and $\mathcal{T}^c := [d] \setminus \mathcal{T}$ its complement set. Let $\hat{\boldsymbol{w}} \in \mathbb{R}^d$ such that $\hat{\boldsymbol{w}}_{\mathcal{T}} = \boldsymbol{X}_{\mathcal{T}}^+ \boldsymbol{y} \in \mathbb{R}^p$ and $\hat{\boldsymbol{w}}_{\mathcal{T}^c} = \boldsymbol{0} \in \mathbb{R}^q$. If $(\boldsymbol{x}, \cdot) \sim P_{\mathcal{X},\mathcal{Y}}^{OOD}$, then the expected OOD risk of the predictor $\hat{f}_{\mathcal{T}} : \boldsymbol{x} \mapsto \phi(\boldsymbol{x}^T \hat{\boldsymbol{w}}) \in \hat{\mathcal{F}}_d$ satisfies*

$$\mathbb{E}_{\boldsymbol{X}}\left[R_{OOD}(\hat{f})\right] \geq \left(\lambda_{\min}(\boldsymbol{\Sigma}) + \lambda_{\min}(\boldsymbol{\Sigma}^{OOD})\right)c(n, p, \sigma') + 2\sigma'^2$$

*and*

$$\mathbb{E}_{\boldsymbol{X}}\left[R_{OOD}(\hat{f})\right] \leq \left(\lambda_{\max}(\boldsymbol{\Sigma}) + \lambda_{\max}(\boldsymbol{\Sigma}^{OOD})\right)c(n, p, \sigma') + \sigma'^2,$$

*where $c(n, p, \sigma')$ is defined in equation 7, $\boldsymbol{\Sigma} \in \mathbb{R}^{d \times d}$ is defined in equation 8, and $\boldsymbol{\Sigma}^{OOD} \in \mathbb{R}^{d \times d}$ is defined as*

$$\boldsymbol{\Sigma}^{OOD} = \mathbb{E}_{(\boldsymbol{x}, \cdot) \sim P_{\mathcal{X},\mathcal{Y}}^{OOD}}\left[\left(\frac{\Phi(\boldsymbol{x}^T \hat{\boldsymbol{w}}) - \Phi(\boldsymbol{x}^T \boldsymbol{w}^*)}{\boldsymbol{x}^T \hat{\boldsymbol{w}} - \boldsymbol{x}^T \boldsymbol{w}^*}\right)^2 \boldsymbol{x}\boldsymbol{x}^T\right].$$

**Remark 4.6.** Like the expected risk in Theorem 1, we find $\mathbb{E}_{\boldsymbol{X}}\left[R_{\text{OOD}}(\hat{f})\right] = \infty$ around $p = n$, which is characteristic of a double descent phenomenon. This results suggests that OOD scoring functions based on the prediction confidence of binary classifiers $\hat{f}(\cdot) \in \hat{\mathcal{F}}_d$ exhibit a double descent phenomenon similar to what has been reported for the expected risk.

## 5 EXPERIMENTS

In this section, we provide an empirical evaluation of different OOD detection methods with respect to the model width across multiple neural network architectures.

### 5.1 SETUP

**General Setup.** We aim to investigate whether the double descent phenomenon, widely observed in model generalization setup, also extends to OOD detection. To explore this, we perform experiments on multiple DNN architectures: ResNet-18 (He et al., 2016), ResNet-34 (Appendix D.3), a 4-block convolutional neural network (CNN), Vision Transformers (ViTs) (Dosovitskiy et al., 2020) and Swin Transformers (Liu et al., 2021b).

**Model Setup.** To replicate double descent, we follow the experimental setup from Nakkiran et al. (2021), which uses ResNet-18 as the baseline architecture. We apply a similar setup to the 4-block CNN model, ViTs and Swin. We vary the model capacity by altering the number of filters (denoted as $k$) per layer, with values ranging from $k = 1$ to $k = 128$. ResNet-18, which uses 64 filters, operates within the overparameterized regime. The depth of the models is kept constant to isolate the effects of width (effective model complexity). The convolutional models are trained using the cross-entropy loss function, with a learning rate of 0.0001 and the Adam optimizer for 4 000 epochs. This extended training regime ensures that models converge for all explored model widths. Moreover, each experiment is conducted five times (with different random seeds). Finally, further details on the experimental setup for the Transformers are given in the Appendix B.2.

**Label Noise.** To induce the double descent effect, we introduce label noise into the training set by randomly swapping 20% of the labels. This setup simulates real-world scenarios, where noisy data is common. The models are trained on this noisy dataset but evaluated on a clean test set. Random data augmentations, including random cropping and horizontal flipping, are applied during training. Noiseless experiments and discussions on the imbalanced dataset case are presented in D.5.

### 5.2 EVALUATION METRICS

We evaluate both generalization and OOD detection using multiple metrics:

- **Generalization**: We report the test accuracy for in-distribution (ID) classification tasks.
- **OOD Detection**: We measure OOD detection performance using the area under the receiver operating characteristic curve (AUC), which is threshold-free and widely adopted in OOD detection research. A higher AUC indicates better performance.

- **Neural Collapse**: We report NC metrics, which, as noted in Ammar et al. (2024); Haas et al. (2023); Zhao & Cao (2023); Zhang et al. (2024), are associated with certain aspects of OOD detection.

## 5.3 OOD DATASETS

For OOD detection, we evaluate each model using six well-established OOD benchmark datasets: **Textures** (Cimpoi et al., 2014), **Places365** (Zhou et al., 2017), **iNaturalist** (Van Horn et al., 2018), a 10 000 image subset from (Huang & Li, 2021a), **ImageNet-O** (Hendrycks et al., 2021) and **SUN** (Xiao et al., 2010). For experiments where CIFAR-10 (or CIFAR-100) is the in-distribution dataset, we also include CIFAR-100 (or CIFAR-10) as an additional OOD benchmark. CIFAR-10/100 contains 50 000 training images and 10 000 test images.

## 5.4 OOD DETECTION METHODS

In order to have a discussion that generalises across different OOD Detection methods, we evaluate several state-of-the-art methods, categorized by the information they rely on:

- **Logit-based methods**: Maximum Softmax Probability (MSP) (Hendrycks & Gimpel, 2017), Energy scores (Liu et al., 2020), React (Sun et al., 2021), MaxLogit and KL-Matching (Hendrycks et al., 2022),
- **Feature-based methods**: Mahalanobis distance (Lee et al., 2018b) and Residual score (Wang et al., 2022).
- **Hybrid methods**: ViM (Wang et al., 2022), ASH (Djurisic et al., 2023) and NECO (Ammar et al., 2024).

Although the double descent effect is observed in all of our experiments, results from only a few representative methods will be presented in the main paper. Additional results can be found in Appendix D.

## 5.5 OOD DETECTION AND DOUBLE DESCENT

**Double Descent & OOD Detection.** The primary question addressed in this section is whether the double descent phenomenon extends to OOD detection, as suggested by our theoretical framework. The results focus on the relative performance across underparameterized and overparameterized regimes. We conduct experiments using CIFAR-10 and CIFAR-100 as ID datasets, and assess OOD detection across increasing model widths. Figure 1 presents the evolution of generalization error and OOD detection performance (AUC) for a challenging covariate shift scenario between CIFAR-10 and CIFAR-100. Refer to Appendix D for more results on multiple OOD datasets. This figure illustrates a generalization double descent phenomenon in all models, with logit-based and hybrid OOD detection methods exhibiting a similar curve. This demonstrates that this phenomenon is not exclusive to generalization, but it extends to OOD detection as well. Moreover, the figure displays the average result (from the five runs) as well as the associated variance. These can be seen to be very narrow, which confirms the prevalence of the phenomena.

**Feature-Based Techniques & Interpolation Threshold.** In some cases, no double descent curve is observed for feature-based techniques. This result suggests that the double descent depends either on the used architecture or the data, as discussed in Appendix E.3. Furthermore, we observe that the interpolation threshold is not always perfectly consistent across OOD datasets or techniques. Those observations are consistent with the Nakkiran et al. (2021)'s results on the CIFAR-10 and CIFAR-100 datasets. Those results suggest that theAmmar et al. (2024); Haas et al. (2023); Zhao & Cao (2023); Zhang et al. (2024) effective model complexity (EMC) framework (Nakkiran et al., 2021) defined for the generalization error can be extended to OOD detection.

**Smaller Models for OOD Detection.** Interestingly, in many cases, smaller models are very good OOD detectors. This suggests that in applications where model pruning or DNN simplification is important, using smaller models may offer advantages for detecting OOD samples. Similarly, resource-constrained environments may benefit from lighter models as a viable option for robust

OOD detection. The conditions under which this choice becomes optimal will be discussed in Section 5.6.

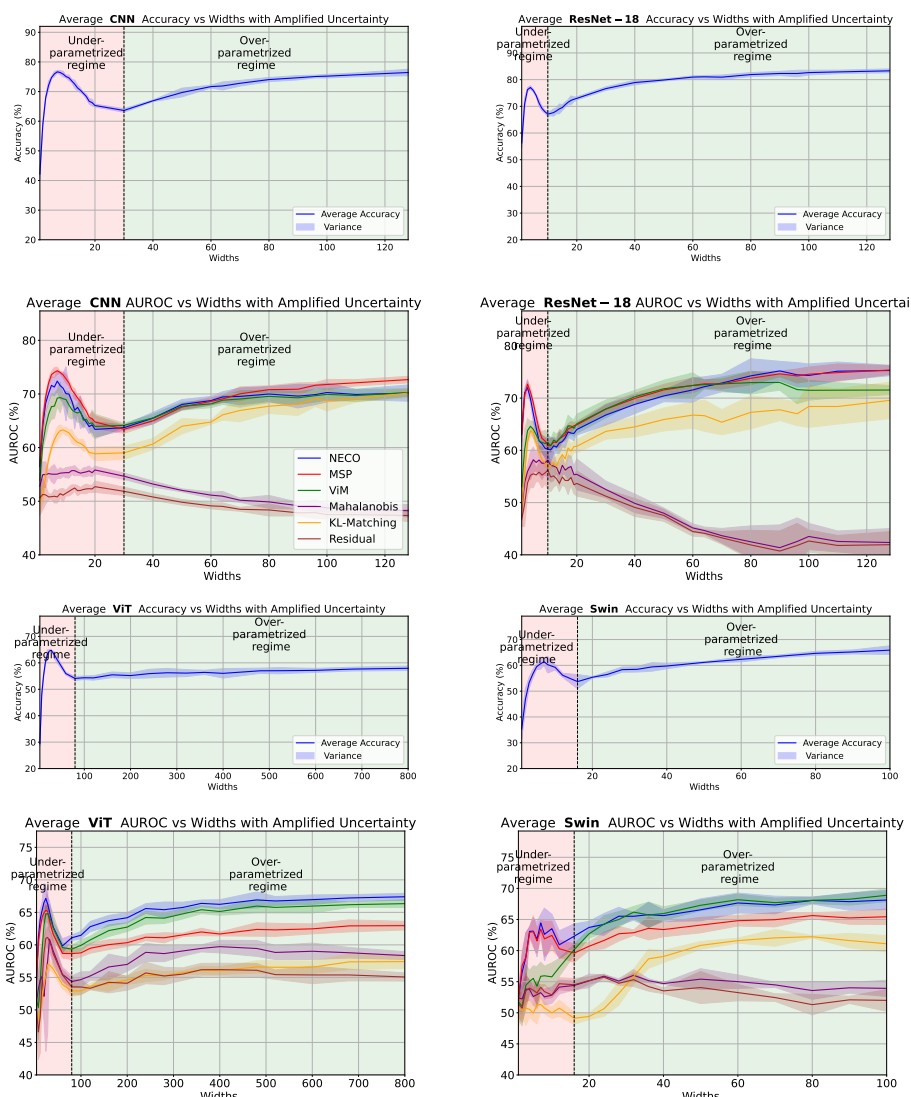

Figure 1: Generalisation (single curve) and OOD detection (multiple curves) evolution plots w.r.t model's width (x-axis), in terms of accuracy and `AUC` respectively. With CIFAR10 as ID and CIFAR100 as OOD, and for (from top-left to bottom-right) CNN, ResNet-18, ViT and Swin.

**Discussion on OOD Methods.** It is important to note that effective OOD detection depends on two main factors: the quality of the learned representations (which shape the feature space) and a reliable confidence score. Different OOD detection methods emphasize one of these factors over the other. Logit-based methods rely primarily on the confidence score, which is determined by the model's output logits. These logits are typically sensitive to the model size and complexity, making them closely tied to the double descent phenomenon. As a result, logit-based methods tend to exhibit smoother double descent curves, with fewer drastic shifts at the interpolation threshold. In contrast, feature space-based techniques rely more heavily on the model's ability to learn high-quality representations that can effectively separate ID from OOD data. However, there is no guarantee that the quality of the latent space discriminative power will be impacted by the double descent phenomenon in the same manner as the output logits.

**Discussion on Different Architectures.** The learned representation is highly complex, with its properties and structure determined by the intrinsic biases of the architecture. This complexity goes

beyond what can be directly inferred from performance metrics alone. As a result, even though all architectures display a double descent pattern, their performance variability remains significant. Notably, some architectures, such as ResNet-18 and Swin, consistently achieve higher performance in the overparameterized regime for both of our metrics of interest. In contrast, the CNN model performs comparably in both regimes, while the ViT struggles with generalization when overparameterized. This performance variability is expected due to the fundamental differences between architectures. To gain deeper insight into the causes of this variability, we study the model's learned representation, which captures its intrinsic biases. In particular, we will use the Neural Collapse framework in our analysis, examining its influence in the double descent setting, and how it can inform us on the possible improvement in OOD detection performance across various architectures.

## 5.6 REPRESENTATION ANALYSIS AND NEURAL COLLAPSE ROLE

**From Double Descent to Neural Collapse.** One of the primary arguments for the interest in double descent in DNNs is that increasing model complexity beyond the interpolation threshold can lead to improved models, compared to those found in underparameterized local minima. However, this improvement does not occur uniformly across all architectures. Although the OOD double descent curve is consistently observed, each architecture exhibits a unique behavior in both generalization and OOD detection. These differences can be attributed to the complexities of the learned representations. We analyze the learned representations using the NC framework to understand this behavior. Previous works (Papyan et al., 2020; Ming et al., 2023; Haas et al., 2023; Ammar et al., 2024) have empirically shown that NC might positively influences both generalization and OOD detection by ensuring stability and strong performance as models converge.

**Background on Neural Collapse.** Neural Collapse (NC) describes the convergence of model representations during the late phases of training towards a low-dimensional and highly structured configuration known as the Equiangular-Tight Frame Simplex (ETF). This structure is characterized by data clustering within each category, with low intra-class covariance, high inter-class separation, and equiangular and equinorm relationships between class representations. Appendix C provides further details on the Neural Collapse phenomenom.

**NC1-based Metric for Overparameterization Analysis.** We will analyze the data clustering and separation properties by leveraging the NC1 metric on the clean test set. NC1 measures the signal-to-noise ratio, where lower values indicate more compact intra-class clustering and greater inter-class separation. The NC1 metric is computed as follows:

$$\text{NC1} = \text{Tr}\left[\frac{\boldsymbol{\Sigma}_W \boldsymbol{\Sigma}_B^+}{C}\right];$$

where $\boldsymbol{\Sigma}_W$ is the intra-class covariance matrix of the penultimate layer of the DNN that depicts noise, $\boldsymbol{\Sigma}_B$ is the inter-class covariance matrix of the penultimate layer of the DNN that represents the signal, and $C$ is the number of classes. As the NC1 value converges towards a lower value, the activations of samples collapse toward their respective class means (see Appendix C for more details). To quantify the influence of overparameterization on the NC1 property, we compute the following ratio:

$$NC1_{u/o} = \frac{NC1_u}{NC1_o},$$

where $NC1_u$ represents the NC1 value at the underparameterized local minimum, and $NC1_o$ is the NC1 value for the most overparameterized model. Values of $NC1_{u/o} > 1$ indicate improved data separation with increased model complexity.

**Analysis of the Results** Table 1 shows that the $NC1_{u/o}$ ratio strongly correlates with improvements in OOD detection. Models that achieve better overparameterized NC1 values tend to improve as their complexity increases. In contrast, the CNN model either stagnates or performs worse with overparameterization, as its $NC1_{u/o}$ metric degrades. We also observe that logit-based methods are well correlated with NC1, with the exception of the MSP method on the ViT model, due to the degradation in generalization. Since NC1 reflects class variability collapse and improved clustering, our results suggest that the separation and clustering effects of the latent space, as measured by NC, can indicate OOD detection performance in the overparameterized regime. The ViT model is an

exception, as its accuracy suffers due to the lack of pretraining. In Appendix E.1, we will study this structure through the eigenvalues to gain further insights into the performance variability.

Table 1: Models performance in terms of AUC in the underparametrized local minima ($\text{AUC}_u$) and the overparametrized maximum width ($\text{AUC}_o$), *w.r.t* $NC1_{u/o}$ value. Best is highlighted in green when $\text{AUC}_o$ is higher, red when $\text{AUC}_u$ is higher and blue if both AUC are within standard deviation range. The highest AUC value per-dataset and per-architecture is highlighted in **bold**.

| Model | $NC1_{u/o}$ | Method | SUN | | Places365 | | CIFAR-100 | |
|---|---|---|---|---|---|---|---|---|
| | | | $\text{AUC}_u \uparrow$ | $\text{AUC}_o \uparrow$ | $\text{AUC}_u \uparrow$ | $\text{AUC}_o \uparrow$ | $\text{AUC}_u \uparrow$ | $\text{AUC}_o \uparrow$ |
| CNN | 0.88 | Softmax score | **76.09±0.96** | 75.08±0.75 | **75.95±0.76** | 74.59±0.45 | **74.33±0.32** | 72.68±0.31 |
| | | MaxLogit | 72.98±2.22 | 60.13±0.35 | 73.33±2.17 | 61.25±0.42 | 73.38±0.39 | 70.37±0.34 |
| | | Energy | 68.08±3.66 | 59.73±0.36 | 69.00±3.56 | 60.90±0.43 | 70.78±0.73 | 70.24±0.35 |
| | | Energy+ReAct | 59.12±4.73 | 47.49±0.58 | 60.79±4.53 | 49.45±0.52 | 66.00±1.58 | 63.57±0.51 |
| | | NECO | 70.43±2.53 | 64.22±1.36 | 71.20±2.59 | 63.59±0.98 | 72.40±0.95 | 70.17±0.76 |
| | | ViM | 59.94±2.01 | 59.77±0.66 | 61.88±1.89 | 60.93±0.40 | 69.23±0.78 | 70.25±0.35 |
| | | ASH-P | 68.60±3.59 | 60.36±0.37 | 69.35±3.50 | 61.48±0.43 | 71.11±0.53 | 70.45±0.41 |
| ResNet | 1.96 | Softmax score | 71.18±0.93 | 75.82±0.89 | 71.22±0.93 | **75.52±0.88** | 71.21±0.48 | **75.37±0.42** |
| | | MaxLogit | 70.64±1.53 | 72.51±1.03 | 70.69±1.29 | 72.64±0.94 | 69.76±0.39 | 73.65±0.38 |
| | | Energy | 69.11±2.49 | 72.46±1.03 | 69.19±2.08 | 72.59±0.94 | 67.58±0.46 | 73.61±0.39 |
| | | Energy+ReAct | 69.57±2.35 | 71.83±0.88 | 69.63±1.93 | 71.97±0.78 | 67.25±0.91 | 72.63±0.45 |
| | | NECO | 70.39±2.30 | **75.60±1.56** | 70.46±1.85 | 75.20±1.42 | 69.92±0.36 | 75.28±0.50 |
| | | ViM | 66.54±1.99 | 74.44±0.65 | 65.38±1.99 | 73.42±0.65 | 64.61±0.47 | 71.54±0.44 |
| | | ASH-P | 69.11±2.49 | 71.73 ±1.09 | 69.19±2.08 | 71.85±0.98 | 67.58±0.46 | 72.89±0.35 |
| Swin | 1.70 | Softmax score | 58.82±2.98 | 67.91±0.69 | 59.01±2.90 | 67.66±0.59 | 61.89±1.42 | 65.44±0.62 |
| | | MaxLogit | 59.91±3.11 | 70.75±0.45 | 59.84±3.40 | 70.46±0.46 | 61.79±1.73 | 66.95±0.53 |
| | | Energy | 60.12±3.68 | 70.79±0.42 | 58.77±3.98 | 70.50±0.44 | 55.52±2.10 | 66.92±0.51 |
| | | Energy+ReAct | 60.16±3.79 | 71.15±0.44 | 58.85±4.03 | 70.83±0.48 | 55.53±2.10 | 67.27±0.52 |
| | | NECO | 64.26±3.20 | **73.29±0.75** | 63.88±3.37 | **72.38±0.66** | 62.62±2.10 | 68.13±0.76 |
| | | ViM | 60.68±2.61 | 71.69±0.19 | 58.34±2.60 | 71.39±0.15 | 55.95±0.77 | 68.89±0.51 |
| | | ASH-P | 59.49±4.21 | 70.74±0.44 | 58.15±4.42 | 70.42±0.40 | 55.10±2.28 | **66.89±0.55** |
| ViT | 2.32 | Softmax score | 66.28±0.19 | 64.87±0.27 | 66.26±0.36 | 64.61±0.26 | 65.18±0.38 | 62.96±0.33 |
| | | MaxLogit | 66.09±1.48 | 70.30±0.46 | 66.13±1.50 | 69.79±0.26 | 64.60±0.35 | 66.69±0.39 |
| | | Energy | 64.79±2.81 | 70.50±0.48 | 64.86±2.65 | 69.98±0.26 | 63.08±0.44 | 66.79±038 |
| | | Energy+ReAct | 64.51±2.93 | 70.51±0.49 | 64.65±2.75 | 69.97±0.26 | 62.86±0.58 | 66.78±039 |
| | | NECO | 67.61±1.61 | **75.89±0.47** | 67.47±1.68 | **74.29±0.29** | 66.28±0.54 | **67.40±0.27** |
| | | ViM | 63.14±3.54 | 72.25±0.37 | 63.30±3.36 | 71.41±0.15 | 64.81±0.65 | 66.34±0.30 |
| | | ASH-P | 64.79±2.81 | 70.27±0.50 | 64.86±2.65 | 69.79±0.25 | 63.08±0.44 | 66.61±0.36 |

## 6 CONCLUSION

In this work, we conducted a theoretical and empirical study on the double descent phenomenon in both classification and OOD detection. Our findings indicate that the double descent phenomenon also occurs in OOD detection, with significant implications for model performance. We introduced the expected OOD risk to evaluate classifiers' confidence on both training and OOD samples. Using Random Matrix Theory, we demonstrated that both the expected risk and OOD risk of least-squares binary classifiers applied to Gaussian models exhibit an infinite peak, when the number of parameters is equal to the number of samples, which we associate with the double descent phenomenon. Our experimental study on different OOD detection methods revealed a similar double descent phenomenon across multiple neural architectures. However, we observed significant variability in performance among different models, with some showing no advantages from overparameterization. Using the Neural Collapse (NC) framework, we revealed that OOD detection improves with overparameterization only when it enhances NC convergence, boosting the performance of OOD detection methods. This emphasizes the crucial role of learned representations in the performance of overparameterized models and their significance in model selection.

We hope our insights and extensive experiments will benefit practitioners in OOD detection and inspire further theoretical research into this aspect of DNNs. Ultimately, although this paper introduces a novel theoretical framework for understanding the double descent phenomenon in OOD detection, its theoretical scope has some limitations including a focus on binary classification, the choice of loss function, and specific model architectures. Hence solving these limitations would be a valuable direction for future work.

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

TABLE OF CONTENTS - SUPPLEMENTARY MATERIAL

# A  PROOF OF THEOREMS

## A.1  PROOF OF THEOREM 1

This section is dedicated to the proof of Theorem 1 and follows the same line of arguments of Theorem 1 in Belkin et al. (2020).

**Theorem 3.** *Let $(p, q) \in [\![1, d]\!]^2$ such that $p + q = d$, $\mathcal{T} \subseteq [d]$ an arbitrary subset of the $d$ first natural integers, and $\mathcal{T}^c := [d] \setminus \mathcal{T}$ its complement set. Let $\hat{\boldsymbol{w}} \in \mathbb{R}^d$ such that $\hat{\boldsymbol{w}}_\mathcal{T} = \boldsymbol{X}_\mathcal{T}^+ \boldsymbol{y} \in \mathbb{R}^p$ and $\hat{\boldsymbol{w}}_{\mathcal{T}^c} = \boldsymbol{0} \in \mathbb{R}^q$. Then the expected risk with respect to the loss function $\ell : (\hat{y}, y) \mapsto (\hat{y} - y)^2$ of the predictor $\hat{f}_\mathcal{T} : \boldsymbol{x} \mapsto \phi(\boldsymbol{x}^T \hat{\boldsymbol{w}}) \in \hat{\mathcal{F}}_d$ satisfies*

$$\lambda_{\min}(\boldsymbol{\Sigma})c(n, p) + \sigma^2 \leq \mathbb{E}_{\boldsymbol{X}}\left[R(\hat{f}_\mathcal{T})\right] \leq \lambda_{\max}(\boldsymbol{\Sigma})c(n, p) + \sigma^2,$$

*where*

$$
c(n, p) = \begin{cases}
\frac{n}{n-p-1}\left(\|\boldsymbol{w}_{\mathcal{T}^c}^*\|_2^2 + \sigma^2\right) + \|\boldsymbol{w}_{\mathcal{T}^c}^*\|_2^2 & \text{if } p \leq n - 2, \\
+\infty & \text{if } n - 1 \leq p \leq n + 1, \\
\left(1 - \frac{n}{p}\right)\|\boldsymbol{w}_\mathcal{T}^*\|_2^2 + \frac{n}{p-n-1}\left(\|\boldsymbol{w}_{\mathcal{T}^c}^*\|_2^2 + \sigma^2\right) + \|\boldsymbol{w}_{\mathcal{T}^c}^*\|_2^2 & \text{if } p \geq n + 2,
\end{cases}
$$

*and*

$$\boldsymbol{\Sigma} = \mathbb{E}_{(\boldsymbol{x}, \cdot) \sim P_{\mathcal{X}, \mathcal{Y}}}\left[\left(\frac{\Phi(\boldsymbol{x}^T \hat{\boldsymbol{w}}) - \Phi(\boldsymbol{x}^T \boldsymbol{w}^*)}{\boldsymbol{x}^T \hat{\boldsymbol{w}} - \boldsymbol{x}^T \boldsymbol{w}^*}\right)^2 \boldsymbol{x}\boldsymbol{x}^T\right].$$

*Proof.* Let $\boldsymbol{x} \in \mathcal{X}$ and

$$\boldsymbol{\Delta}_{\boldsymbol{w}} = \hat{\boldsymbol{w}} - \boldsymbol{w}^*.$$

We have

$$\hat{f}_\mathcal{T}(\boldsymbol{x}) = \phi(\boldsymbol{x}^T \hat{\boldsymbol{w}}) = \phi(\boldsymbol{x}^T \boldsymbol{w}^*) + \boldsymbol{x}^T \boldsymbol{\Delta}_{\boldsymbol{w}} \phi'\big(c(\boldsymbol{x}, \hat{\boldsymbol{w}}, \boldsymbol{w}^*)\big).$$

From the mean-value theorem, there exists

$$c(\boldsymbol{x}, \hat{\boldsymbol{w}}, \boldsymbol{w}^*) \in \big(\min(\boldsymbol{x}^T \boldsymbol{w}, \boldsymbol{x}^T \boldsymbol{w}^*), \max(\boldsymbol{x}^T \hat{\boldsymbol{w}}, \boldsymbol{x}^T \boldsymbol{w}^*)\big),$$

such that

$$\phi\big(\boldsymbol{x}^T(\boldsymbol{w}^* + \boldsymbol{\Delta}_{\boldsymbol{w}})\big) = \phi(\boldsymbol{x}^T \boldsymbol{w}^*) + \boldsymbol{x}^T \boldsymbol{\Delta}_{\boldsymbol{w}} \phi'\big(c(\boldsymbol{x}, \hat{\boldsymbol{w}}, \boldsymbol{w}^*)\big).$$

We have thus

$$
\begin{aligned}
\mathbb{E}_{\boldsymbol{X}}\left[R(\hat{f}_\mathcal{T})\right] = \mathbb{E}_{\boldsymbol{x}, \boldsymbol{X}}\left[\big(\phi(\boldsymbol{x}^T \hat{\boldsymbol{w}}) - y\big)^2\right] &= \mathbb{E}_{\boldsymbol{x}, \boldsymbol{X}}\left[\big(\phi(\boldsymbol{x}^T \hat{\boldsymbol{w}}) - \phi(\boldsymbol{x}^T \boldsymbol{w}^*) - \epsilon\big)^2\right] \\
&= \mathbb{E}_{\boldsymbol{x}, \boldsymbol{X}}\left[\big(\boldsymbol{x}^T \boldsymbol{\Delta}_{\boldsymbol{w}} \phi'\big(c(\boldsymbol{x}, \hat{\boldsymbol{w}}, \boldsymbol{w}^*)\big)\big)^2\right] + \sigma^2 \\
&= \mathbb{E}_{\boldsymbol{x}, \boldsymbol{X}}\left[\boldsymbol{\Delta}_{\boldsymbol{w}}^T \boldsymbol{x}\boldsymbol{x}^T \boldsymbol{\Delta}_{\boldsymbol{w}} \phi'\big(c(\boldsymbol{x}, \hat{\boldsymbol{w}}, \boldsymbol{w}^*)\big)^2\right] + \sigma^2 \\
&= \text{Tr}\left(\mathbb{E}_{\boldsymbol{x}, \boldsymbol{X}}\left[\phi'\big(c(\boldsymbol{x}, \hat{\boldsymbol{w}}, \boldsymbol{w}^*)\big)^2 \boldsymbol{x}\boldsymbol{x}^T \boldsymbol{\Delta}_{\boldsymbol{w}} \boldsymbol{\Delta}_{\boldsymbol{w}}^T\right]\right) + \sigma^2 \\
&= \text{Tr}\left(\mathbb{E}_{\boldsymbol{x}}\left[\phi'\big(c(\boldsymbol{x}, \hat{\boldsymbol{w}}, \boldsymbol{w}^*)\big)^2 \boldsymbol{x}\boldsymbol{x}^T\right] \mathbb{E}_{\boldsymbol{X}}\left[\boldsymbol{\Delta}_{\boldsymbol{w}} \boldsymbol{\Delta}_{\boldsymbol{w}}^T\right]\right) + \sigma^2 \\
&= \text{Tr}\left(\boldsymbol{\Sigma} \mathbb{E}_{\boldsymbol{X}}\left[\boldsymbol{\Delta}_{\boldsymbol{w}} \boldsymbol{\Delta}_{\boldsymbol{w}}^T\right]\right) + \sigma^2 \\
&= \mathbb{E}_{\boldsymbol{X}}\left[\boldsymbol{\Delta}_{\boldsymbol{w}}^T \boldsymbol{\Sigma} \boldsymbol{\Delta}_{\boldsymbol{w}}\right] + \sigma^2,
\end{aligned}
$$

where

$$
\begin{aligned}
\boldsymbol{\Sigma} &= \mathbb{E}_{\boldsymbol{x}}\left[\phi'\big(c(\boldsymbol{x}, \hat{\boldsymbol{w}}, \boldsymbol{w}^*)\big)^2 \boldsymbol{x}\boldsymbol{x}^T\right] \\
&= \mathbb{E}_{\boldsymbol{x}}\left[\left(\frac{\Phi(\boldsymbol{x}^T \hat{\boldsymbol{w}}) - \Phi(\boldsymbol{x}^T \boldsymbol{w}^*)}{\boldsymbol{x}^T \hat{\boldsymbol{w}} - \boldsymbol{x}^T \boldsymbol{w}^*}\right)^2 \boldsymbol{x}\boldsymbol{x}^T\right].
\end{aligned}
$$

From the min-max theorem, we have

$$\lambda_{\min}(\boldsymbol{\Sigma})\mathbb{E}_{\boldsymbol{X}}\left[\boldsymbol{\Delta}_{\boldsymbol{w}}^T\boldsymbol{\Delta}_{\boldsymbol{w}}\right] + \sigma^2 \leq \mathbb{E}_{\boldsymbol{X}}\left[\boldsymbol{\Delta}_{\boldsymbol{w}}^T\boldsymbol{\Sigma}\boldsymbol{\Delta}_{\boldsymbol{w}}\right] + \sigma^2 \leq \lambda_{\max}(\boldsymbol{\Sigma})\mathbb{E}_{\boldsymbol{X}}\left[\boldsymbol{\Delta}_{\boldsymbol{w}}^T\boldsymbol{\Delta}_{\boldsymbol{w}}\right] + \sigma^2.$$

From Lemma 4.1, we have $\lambda_{\min}(\boldsymbol{\Sigma}) > 0.$ For $\mathbb{E}_{\boldsymbol{X}}\left[\boldsymbol{\Delta}_{\boldsymbol{w}}^T\boldsymbol{\Delta}_{\boldsymbol{w}}\right]$, we have

$$\begin{aligned}
\mathbb{E}_{\boldsymbol{X}}\left[\boldsymbol{\Delta}_{\boldsymbol{w}}^T\boldsymbol{\Delta}_{\boldsymbol{w}}\right] &= \mathbb{E}_{\boldsymbol{X}}\left[\|\hat{\boldsymbol{w}} - \boldsymbol{w}^*\|_2^2\right] \\
&= \mathbb{E}_{\boldsymbol{X}}\left[\|\hat{\boldsymbol{w}}_{\mathcal{T}} - \boldsymbol{w}_{\mathcal{T}}^*\|_2^2\right] + \mathbb{E}_{\boldsymbol{X}}\left[\|\hat{\boldsymbol{w}}_{\mathcal{T}^c} - \boldsymbol{w}_{\mathcal{T}^c}^*\|_2^2\right] \\
&= \mathbb{E}_{\boldsymbol{X}}\left[\|\hat{\boldsymbol{w}}_{\mathcal{T}} - \boldsymbol{w}_{\mathcal{T}}^*\|_2^2\right] + \|\boldsymbol{w}_{\mathcal{T}^c}^*\|_2^2.
\end{aligned}$$

as $\hat{\boldsymbol{w}}_{\mathcal{T}^c} = \boldsymbol{0}$. In the following, we provide a decomposition of $\mathbb{E}_{\boldsymbol{X}}\left[\|\hat{\boldsymbol{w}}_{\mathcal{T}} - \boldsymbol{w}_{\mathcal{T}}^*\|_2^2\right]$. Let $\boldsymbol{\eta} = \boldsymbol{y} - \boldsymbol{X}_{\mathcal{T}}\boldsymbol{w}_{\mathcal{T}}^*$. Since $\hat{\boldsymbol{w}}_{\mathcal{T}} = \boldsymbol{X}_{\mathcal{T}}^+\boldsymbol{y}$, we have $\hat{\boldsymbol{w}}_{\mathcal{T}} = \boldsymbol{X}_{\mathcal{T}}^+(\boldsymbol{\eta} + \boldsymbol{X}_{\mathcal{T}}\boldsymbol{w}_{\mathcal{T}}^*)$. Therefore,

$$\begin{aligned}
\mathbb{E}_{\boldsymbol{X}}\left[\|\boldsymbol{w}_{\mathcal{T}}^* - \hat{\boldsymbol{w}}_{\mathcal{T}}\|_2^2\right] &= \mathbb{E}_{\boldsymbol{X}}\left[\|(\boldsymbol{I}_p - \boldsymbol{X}_{\mathcal{T}}^+\boldsymbol{X}_{\mathcal{T}})\boldsymbol{w}_{\mathcal{T}}^* - \boldsymbol{X}_{\mathcal{T}}^+\boldsymbol{\eta}\|_2^2\right] \\
&= \mathbb{E}_{\boldsymbol{X}}\left[\|(\boldsymbol{I}_p - \boldsymbol{X}_{\mathcal{T}}^+\boldsymbol{X}_{\mathcal{T}})\boldsymbol{w}_{\mathcal{T}}^*\|_2^2 + \|\boldsymbol{X}_{\mathcal{T}}^+\boldsymbol{\eta}\|_2^2 - 2\langle(\boldsymbol{I}_p - \boldsymbol{X}_{\mathcal{T}}^+\boldsymbol{X}_{\mathcal{T}})\,\boldsymbol{w}_{\mathcal{T}}^*, \boldsymbol{X}_{\mathcal{T}}^+\boldsymbol{\eta}\rangle\right].
\end{aligned}$$

Since $(\boldsymbol{X}_{\mathcal{T}}^+\boldsymbol{X}_{\mathcal{T}})^T = \boldsymbol{X}_{\mathcal{T}}^+\boldsymbol{X}_{\mathcal{T}}$ and $(\boldsymbol{X}_{\mathcal{T}}^+\boldsymbol{X}_{\mathcal{T}})^T\boldsymbol{X}_{\mathcal{T}}^+ = \boldsymbol{X}_{\mathcal{T}}^+$, we have

$$\begin{aligned}
\langle(\boldsymbol{I}_p - \boldsymbol{X}_{\mathcal{T}}^+\boldsymbol{X}_{\mathcal{T}})\boldsymbol{w}_{\mathcal{T}}^*, \boldsymbol{X}_{\mathcal{T}}^+\boldsymbol{\eta}\rangle_2 &= \left((\boldsymbol{I}_p - \boldsymbol{X}_{\mathcal{T}}^+\boldsymbol{X}_{\mathcal{T}})\boldsymbol{w}_{\mathcal{T}}^*\right)^T\boldsymbol{X}_{\mathcal{T}}^+\boldsymbol{\eta} \\
&= \boldsymbol{w}_{\mathcal{T}}^{*T}(\boldsymbol{I}_p - \boldsymbol{X}_{\mathcal{T}}^+\boldsymbol{X}_{\mathcal{T}})^T\boldsymbol{X}_{\mathcal{T}}^+\boldsymbol{\eta} \\
&= \boldsymbol{w}_{\mathcal{T}}^{*T}\boldsymbol{X}_{\mathcal{T}}^+\boldsymbol{\eta} - \boldsymbol{w}_{\mathcal{T}}^{*T}(\boldsymbol{X}_{\mathcal{T}}^+\boldsymbol{X}_{\mathcal{T}})\boldsymbol{X}_{\mathcal{T}}^+\boldsymbol{\eta} \\
&= \boldsymbol{w}_{\mathcal{T}}^{*T}\boldsymbol{X}_{\mathcal{T}}^+\boldsymbol{\eta} - \boldsymbol{w}_{\mathcal{T}}^{*T}\boldsymbol{X}_{\mathcal{T}}^+\boldsymbol{\eta} \\
&= 0.
\end{aligned}$$

$(\boldsymbol{I}_p - \boldsymbol{X}_{\mathcal{T}}^+\boldsymbol{X}_{\mathcal{T}})\boldsymbol{w}_{\mathcal{T}}^*$ and $\boldsymbol{X}_{\mathcal{T}}^+\boldsymbol{\eta}$ are thus orthogonal. We deduce that

$$\begin{aligned}
\mathbb{E}_{\boldsymbol{X}}\left[\|\hat{\boldsymbol{w}}_{\mathcal{T}} - \boldsymbol{w}_{\mathcal{T}}^*\|_2^2\right] &= \mathbb{E}_{\boldsymbol{X}}\left[\|(\boldsymbol{I}_p - \boldsymbol{X}_{\mathcal{T}}^+\boldsymbol{X}_{\mathcal{T}})\,\boldsymbol{w}_{\mathcal{T}}^*\|_2^2 + \|\boldsymbol{X}_{\mathcal{T}}^+\boldsymbol{\eta}\|_2^2\right] \\
&= \mathbb{E}_{\boldsymbol{X}}\left[\|(\boldsymbol{I}_p - \boldsymbol{X}_{\mathcal{T}}^+\boldsymbol{X}_{\mathcal{T}})\,\boldsymbol{w}_{\mathcal{T}}^*\|_2^2\right] + \mathbb{E}_{\boldsymbol{X}}\left[\|\boldsymbol{X}_{\mathcal{T}}^+\boldsymbol{\eta}\|_2^2\right].
\end{aligned} \tag{9}$$

Leveraging the same arguments used by Belkin et al. (2020) to prove the existence of the double descent phenomenon in the regression problem, we distinguish two cases depending on $n$ and $p$ to derive equation 9.

**Classical Regime** ($p < n$). Breiman & Freedman (1983) studied this regime for the regression problem. In the classical regime, the Moore-Penrose inverse is equal to:

$$\boldsymbol{X}_{\mathcal{T}}^+ = (\boldsymbol{X}_{\mathcal{T}}^T\boldsymbol{X}_{\mathcal{T}})^{-1}\boldsymbol{X}_{\mathcal{T}}^T,$$

which implies that

$$\mathbb{E}_{\boldsymbol{X}}\left[\|(\boldsymbol{I}_p - \boldsymbol{X}_{\mathcal{T}}^+\boldsymbol{X}_{\mathcal{T}})\,\boldsymbol{w}_{\mathcal{T}}^*\|_2^2\right] = \mathbb{E}_{\boldsymbol{X}}\left[\|(\boldsymbol{I}_p - (\boldsymbol{X}_{\mathcal{T}}^T\boldsymbol{X}_{\mathcal{T}})^{-1}\boldsymbol{X}_{\mathcal{T}}^T\boldsymbol{X}_{\mathcal{T}})\boldsymbol{w}_{\mathcal{T}}^*\|_2^2\right] = 0.$$

From equation 9, we deduce that

$$\mathbb{E}_{\boldsymbol{X}}\left[\|\hat{\boldsymbol{w}}_{\mathcal{T}} - \boldsymbol{w}_{\mathcal{T}}^*\|_2^2\right] = \mathbb{E}_{\boldsymbol{X}}\left[\|\boldsymbol{X}_{\mathcal{T}}^+\boldsymbol{\eta}\|_2^2\right] = \mathbb{E}_{\boldsymbol{X}}\left[\boldsymbol{\eta}^T\boldsymbol{X}_{\mathcal{T}}^{+T}\boldsymbol{X}_{\mathcal{T}}^+\boldsymbol{\eta}\right] = \mathrm{Tr}\left(\mathbb{E}_{\boldsymbol{X}}\left[\boldsymbol{X}_{\mathcal{T}}^{+T}\boldsymbol{X}_{\mathcal{T}}^+\boldsymbol{\eta}\boldsymbol{\eta}^T\right]\right).$$

We observe that

$$\begin{aligned}
\boldsymbol{\eta} &= \boldsymbol{y} - \boldsymbol{X}_{\mathcal{T}}\boldsymbol{w}_{\mathcal{T}}^* \\
&= \boldsymbol{X}_{\mathcal{T}}\boldsymbol{w}_{\mathcal{T}}^* + \boldsymbol{X}_{\mathcal{T}^c}\boldsymbol{w}_{\mathcal{T}^c}^* + \boldsymbol{\epsilon} - \boldsymbol{X}_{\mathcal{T}}\boldsymbol{w}_{\mathcal{T}}^* \\
&= \boldsymbol{X}_{\mathcal{T}^c}\boldsymbol{w}_{\mathcal{T}^c}^* + \boldsymbol{\epsilon},
\end{aligned} \tag{10}$$

where $\boldsymbol{\epsilon} = [\epsilon_1, \ldots, \epsilon_n]^T$. Because $\boldsymbol{X}_{\mathcal{T}}\boldsymbol{w}_{\mathcal{T}}^*$ and $\boldsymbol{X}_{\mathcal{T}^c}\boldsymbol{w}_{\mathcal{T}^c}^* + \boldsymbol{\epsilon}$ are both uncorrelated, we have

$$\mathbb{E}_{\boldsymbol{X}}\left[\|\hat{\boldsymbol{w}}_{\mathcal{T}} - \boldsymbol{w}_{\mathcal{T}}^*\|_2^2\right] = \mathrm{Tr}\left(\mathbb{E}_{\boldsymbol{X}}\left[\boldsymbol{X}_{\mathcal{T}}^{+T}\boldsymbol{X}_{\mathcal{T}}^+\right]\mathbb{E}_{\boldsymbol{X}}\left[\boldsymbol{\eta}\boldsymbol{\eta}^T\right]\right). \tag{11}$$

We have

$$\mathbb{E}_{\boldsymbol{X}}\big[\boldsymbol{X}_{\mathcal{T}}^{+T}\boldsymbol{X}_{\mathcal{T}}^{+}\big] = \mathbb{E}_{\boldsymbol{X}}\big[\boldsymbol{X}_{\mathcal{T}}(\boldsymbol{X}_{\mathcal{T}}^{T}\boldsymbol{X}_{\mathcal{T}})^{-2}\boldsymbol{X}_{\mathcal{T}}^{T}\big]$$

Let $\boldsymbol{S}$ be the unique positive definite square root of $\boldsymbol{X}_{\mathcal{T}}^{T}\boldsymbol{X}_{\mathcal{T}}$ and $\boldsymbol{\Psi} = \boldsymbol{X}_{\mathcal{T}}\boldsymbol{S}^{-1}$, an $n \times p$ matrix and $\boldsymbol{X}_{\mathcal{T}} = \boldsymbol{\Psi}\boldsymbol{S}$. $\boldsymbol{\Psi}$ is an orthonormal such that $\boldsymbol{\Psi}^{T}\boldsymbol{\Psi} = \boldsymbol{I}_p$ and $\boldsymbol{\Psi}\boldsymbol{\Psi}^{T} = \boldsymbol{I}_n$. Following the same arguments than Breiman & Freedman (1983), we obtain

$$\begin{aligned}
\mathbb{E}_{\boldsymbol{X}}\big[\boldsymbol{X}_{\mathcal{T}}^{+T}\boldsymbol{X}_{\mathcal{T}}^{+}\big] &= \mathbb{E}_{\boldsymbol{X}}\big[\boldsymbol{X}_{\mathcal{T}}(\boldsymbol{X}_{\mathcal{T}}^{T}\boldsymbol{X}_{\mathcal{T}})^{-2}\boldsymbol{X}_{\mathcal{T}}^{T}\big] \\
&= \mathbb{E}_{\boldsymbol{X}}\big[\boldsymbol{\Psi}\boldsymbol{S}\boldsymbol{S}^{-4}\boldsymbol{S}\boldsymbol{\Psi}^{T}\big] \\
&= \mathbb{E}_{\boldsymbol{X}}\big[\boldsymbol{\Psi}\boldsymbol{S}^{-2}\boldsymbol{\Psi}^{T}\big] \\
&= \boldsymbol{\Psi}\mathbb{E}_{\boldsymbol{X}}\big[(\boldsymbol{X}_{\mathcal{T}}^{T}\boldsymbol{X}_{\mathcal{T}})^{-1}\big]\boldsymbol{\Psi}^{T}.
\end{aligned}$$

$\boldsymbol{X}_{\mathcal{T}}^{T}\boldsymbol{X}_{\mathcal{T}} \in \mathbb{R}^{p \times p}$ follows a Wishart distribution: $\boldsymbol{X}_{\mathcal{T}}^{T}\boldsymbol{X}_{\mathcal{T}} \sim \mathcal{W}_p(n, \boldsymbol{I}_p)$, where $\mathcal{W}_p(n, \boldsymbol{I}_p)$ denotes a Wishart distribution with $n$ degrees of freedom and scale matrix $\boldsymbol{I}_p$. The inverse of a Wishart-distributed matrix $\mathcal{W}_p(n, \boldsymbol{I}_p)$ follows an inverse Wishart distribution: $(\boldsymbol{X}_{\mathcal{T}}^{T}\boldsymbol{X}_{\mathcal{T}})^{-1} \sim \mathcal{W}_p^{-1}(n, \boldsymbol{I}_p)$, where $\mathcal{W}_p^{-1}(n, \boldsymbol{I}_p)$ denotes an inverse Wishart distribution with $n$ degrees of freedom and scale matrix $\boldsymbol{I}_p$. As a consequence,

$$\mathbb{E}_{\boldsymbol{X}}\Big[\big(\boldsymbol{X}_{\mathcal{T}}^{T}\boldsymbol{X}_{\mathcal{T}}\big)^{-1}\Big] = \tfrac{1}{n-p-1}\boldsymbol{I}_p$$

and thus

$$\mathbb{E}_{\boldsymbol{X}}\big[\boldsymbol{X}_{\mathcal{T}}^{+T}\boldsymbol{X}_{\mathcal{T}}^{+}\big] = \tfrac{1}{n-p-1}\boldsymbol{I}_p. \tag{12}$$

Putting equation 12 into equation 11, we get

$$\mathbb{E}_{\boldsymbol{X}}\Big[\big\|\hat{\boldsymbol{w}}_{\mathcal{T}} - \boldsymbol{w}_{\mathcal{T}}^{*}\big\|_2^2\Big] = \tfrac{1}{n-p-1}\mathbb{E}_{\boldsymbol{X}}\big[\boldsymbol{\eta}^{T}\boldsymbol{\eta}\big].$$

From equation 10, we have

$$\begin{aligned}
\mathbb{E}_{\boldsymbol{X}}\big[\boldsymbol{\eta}^{T}\boldsymbol{\eta}\big] &= \mathbb{E}_{\boldsymbol{X}}\big[(\boldsymbol{y} - \boldsymbol{X}_{\mathcal{T}}\boldsymbol{w}_{\mathcal{T}}^{*})^{T}(\boldsymbol{y} - \boldsymbol{X}_{\mathcal{T}}\boldsymbol{w}_{\mathcal{T}}^{*})\big] \\
&= \mathbb{E}_{\boldsymbol{X}}\Big[(\boldsymbol{X}_{\mathcal{T}^c}\boldsymbol{w}_{\mathcal{T}^c}^{*} + \boldsymbol{\epsilon})^{T}(\boldsymbol{X}_{\mathcal{T}^c}\boldsymbol{w}_{\mathcal{T}^c}^{*} + \boldsymbol{\epsilon})\Big] \\
&= \boldsymbol{w}_{\mathcal{T}^c}^{*T}\mathbb{E}_{\boldsymbol{X}}\big[\boldsymbol{X}_{\mathcal{T}^c}^{T}\boldsymbol{X}_{\mathcal{T}^c}\big]\boldsymbol{w}_{\mathcal{T}^c}^{*} + \underbrace{\mathbb{E}_{\boldsymbol{X}}\big[\boldsymbol{\epsilon}^{T}\big]}_{=\boldsymbol{0}}\underbrace{\mathbb{E}_{\boldsymbol{X}}\big[\boldsymbol{X}_{\mathcal{T}^c}\big]}_{=\boldsymbol{0}}\boldsymbol{w}_{\mathcal{T}^c}^{*} + \boldsymbol{w}_{\mathcal{T}^c}^{*T}\underbrace{\mathbb{E}_{\boldsymbol{X}}\big[\boldsymbol{X}_{\mathcal{T}^c}^{T}\big]}_{=\boldsymbol{0}}\underbrace{\mathbb{E}_{\boldsymbol{X}}\big[\boldsymbol{\epsilon}\big]}_{=\boldsymbol{0}} + \mathbb{E}_{\boldsymbol{X}}\big[\boldsymbol{\epsilon}^{T}\boldsymbol{\epsilon}\big].
\end{aligned}$$

$\boldsymbol{X}_{\mathcal{T}^c}^{T}\boldsymbol{X}_{\mathcal{T}^c}$ follows a Wishart distribution, *i.e.*, $\boldsymbol{X}_{\mathcal{T}^c}^{T}\boldsymbol{X}_{\mathcal{T}^c} \sim \mathcal{W}_q(n, \boldsymbol{I}_q)$, where $\mathcal{W}_q(n, \boldsymbol{I}_q)$ denotes the Wishart distributions with $n$ degrees of freedom and scale matrix $\boldsymbol{I}_q$, respectively. We obtain thus

$$\begin{aligned}
\mathbb{E}_{\boldsymbol{X}}\big[\boldsymbol{\eta}^{T}\boldsymbol{\eta}\big] &= n\boldsymbol{w}_{\mathcal{T}^c}^{*T}\boldsymbol{w}_{\mathcal{T}^c}^{*} + n\sigma^2 \\
&= n\|\boldsymbol{w}_{\mathcal{T}^c}^{*}\|_2^2 + n\sigma^2.
\end{aligned} \tag{13}$$

As a consequence, we obtain

$$\mathbb{E}_{\boldsymbol{X}}\Big[\big\|\hat{\boldsymbol{w}}_{\mathcal{T}} - \boldsymbol{w}_{\mathcal{T}}^{*}\big\|_F^2\Big] = \tfrac{n}{n-p-1}\Big(\|\boldsymbol{w}_{\mathcal{T}^c}^{*}\|_2^2 + \sigma^2\Big).$$

**Interpolating Regime** ($p \geq n$). **The interpolating regime** has been considered in Belkin et al. (2020) for the regression problem. We can first observe that:

$$\boldsymbol{w}_{\mathcal{T}}^{*} = (\boldsymbol{I}_p - \boldsymbol{X}_{\mathcal{T}}^{+}\boldsymbol{X}_{\mathcal{T}})\boldsymbol{w}_{\mathcal{T}}^{*} + \boldsymbol{X}_{\mathcal{T}}^{+}\boldsymbol{X}_{\mathcal{T}}\boldsymbol{w}_{\mathcal{T}}^{*}$$

and

$$\big\langle(\boldsymbol{I}_p - \boldsymbol{X}_{\mathcal{T}}^{+}\boldsymbol{X}_{\mathcal{T}})\boldsymbol{w}_{\mathcal{T}}^{*}, \boldsymbol{X}_{\mathcal{T}}^{+}\boldsymbol{X}_{\mathcal{T}}\boldsymbol{w}_{\mathcal{T}}^{*}\big\rangle = 0.$$

Indeed, since $\boldsymbol{X}_{\mathcal{T}}^{+}\boldsymbol{X}_{\mathcal{T}} = (\boldsymbol{X}_{\mathcal{T}}^{+}\boldsymbol{X}_{\mathcal{T}})(\boldsymbol{X}_{\mathcal{T}}^{+}\boldsymbol{X}_{\mathcal{T}})$, we have

$$\begin{aligned}
\big\langle(\boldsymbol{I}_p - \boldsymbol{X}_{\mathcal{T}}^{+}\boldsymbol{X}_{\mathcal{T}})\boldsymbol{w}_{\mathcal{T}}^{*}, \boldsymbol{X}_{\mathcal{T}}^{+}\boldsymbol{X}_{\mathcal{T}}\boldsymbol{w}_{\mathcal{T}}^{*}\big\rangle &= (\boldsymbol{I}_p - \boldsymbol{X}_{\mathcal{T}}^{+}\boldsymbol{X}_{\mathcal{T}})\boldsymbol{w}_{\mathcal{T}}^{*}\big)^{T}(\boldsymbol{X}_{\mathcal{T}}^{+}\boldsymbol{X}_{\mathcal{T}}\boldsymbol{w}_{\mathcal{T}}^{*}) \\
&= \boldsymbol{w}_{\mathcal{T}}^{*T}\big(\boldsymbol{X}_{\mathcal{T}}^{+}\boldsymbol{X}_{\mathcal{T}} - (\boldsymbol{X}_{\mathcal{T}}^{+}\boldsymbol{X}_{\mathcal{T}})(\boldsymbol{X}_{\mathcal{T}}^{+}\boldsymbol{X}_{\mathcal{T}})\big)\boldsymbol{w}_{\mathcal{T}}^{*} \\
&= 0.
\end{aligned}$$

We deduce that $(I_p - X_{\mathcal{T}}^+ X_{\mathcal{T}}) w_{\mathcal{T}}^*$ and $X_{\mathcal{T}}^+ X_{\mathcal{T}} w_{\mathcal{T}}^*$ are orthogonal. From the Pythagorean theorem, we have

$$\left\| w_{\mathcal{T}}^* \right\|_2^2 = \left\| (I_p - X_{\mathcal{T}}^+ X_{\mathcal{T}}) w_{\mathcal{T}}^* \right\|_2^2 + \left\| X_{\mathcal{T}}^+ X_{\mathcal{T}} w_{\mathcal{T}}^* \right\|_2^2$$

and thus

$$\left\| (I_p - X_{\mathcal{T}}^+ X_{\mathcal{T}}) w_{\mathcal{T}}^* \right\|_2^2 = \left\| w_{\mathcal{T}}^* \right\|_2^2 - \left\| X_{\mathcal{T}}^+ X_{\mathcal{T}} w_{\mathcal{T}}^* \right\|_2^2.$$

Putting the equation above into equation 9, we obtain

$$\mathbb{E}_{X}\left[ \left\| \hat{w}_{\mathcal{T}} - w_{\mathcal{T}}^* \right\|_2^2 \right] = \mathbb{E}_{X}\left[ \left\| w_{\mathcal{T}}^* \right\|_2^2 \right] - \mathbb{E}_{X}\left[ \left\| X_{\mathcal{T}}^+ X_{\mathcal{T}} w_{\mathcal{T}}^* \right\|_2^2 \right] + \mathbb{E}_{X}\left[ \left\| X_{\mathcal{T}}^+ \eta \right\|_2^2 \right]. \quad (14)$$

Note that $\Pi_{\mathcal{T}} = X_{\mathcal{T}}^+ X_{\mathcal{T}} = X_{\mathcal{T}}^T (X_{\mathcal{T}} X_{\mathcal{T}}^T)^{-1} X_{\mathcal{T}}$ is the orthogonal projection matrix for the row space of $X_{\mathcal{T}}$. We can thus write $X_{\mathcal{T}}^+ X_{\mathcal{T}} w_{\mathcal{T}} = \Pi_{\mathcal{T}} w_{\mathcal{T}}^*$ as a linear combination of rows of $X_{\mathcal{T}}$. Then, using the fact that the $x_i$ in $X$ are i.i.d. and drawn from a standard normal distribution and by rotational symmetry of the standard normal distribution, it follows that we have

$$\mathbb{E}_{X}\left[ \left\| X_{\mathcal{T}}^+ X_{\mathcal{T}} w_{\mathcal{T}}^* \right\|_2^2 \right] = \tfrac{n}{p} \left\| w_{\mathcal{T}}^* \right\|_2^2$$

and thus

$$\mathbb{E}_{X}\left[ \left\| w_{\mathcal{T}}^* \right\|_2^2 \right] - \mathbb{E}_{X}\left[ \left\| X_{\mathcal{T}}^+ X_{\mathcal{T}} w_{\mathcal{T}}^* \right\|_2^2 \right] = \left\| w_{\mathcal{T}}^* \right\|_2^2 \left( 1 - \tfrac{n}{p} \right).$$

For $\mathbb{E}_{X}\left[ \left\| X_{\mathcal{T}}^+ \eta \right\|_2^2 \right]$ in equation 14, we have

$$\mathbb{E}_{X}\left[ \left\| X_{\mathcal{T}}^+ \eta \right\|_2^2 \right] = \mathrm{Tr}\left( \mathbb{E}_{X}\left[ X_{\mathcal{T}}^{+T} X_{\mathcal{T}}^+ \right] \mathbb{E}_{X}\left[ \eta \eta^T \right] \right).$$

As $p > n$, we have $X_{\mathcal{T}}^+ = X_{\mathcal{T}}^T (X_{\mathcal{T}} X_{\mathcal{T}}^T)^{-1}$ and thus

$$\mathbb{E}_{X}\left[ X_{\mathcal{T}}^{+T} X_{\mathcal{T}}^+ \right] = \mathbb{E}_{X}\left[ (X_{\mathcal{T}} X_{\mathcal{T}}^T)^{-1} \right]$$

Similarly, $X_{\mathcal{T}} X_{\mathcal{T}}^T$ follows a Wishart distribution: $X_{\mathcal{T}} X_{\mathcal{T}}^T \sim \mathcal{W}_n(p, I_n)$, and $(X_{\mathcal{T}} X_{\mathcal{T}}^T)^{-1}$ follows an inverse Wishart distribution: $(X_{\mathcal{T}} X_{\mathcal{T}}^T)^{-1} \sim \mathcal{W}_n^{-1}(p, I_n)$. Its expectation is given by:

$$\mathbb{E}[(X_{\mathcal{T}} X_{\mathcal{T}}^T)^{-1}] = \tfrac{I_n}{p - n - 1}$$

From equation 13, we deduce that

$$\mathbb{E}_{X}\left[ \left\| X_{\mathcal{T}}^+ \eta \right\|_2^2 \right] = \mathrm{Tr}\left( \mathbb{E}_{X}\left[ X_{\mathcal{T}}^{+T} X_{\mathcal{T}}^+ \right] \mathbb{E}_{X}\left[ \eta \eta^T \right] \right)$$

$$= \tfrac{n}{p - n - 1} \left( \| w_{\mathcal{T}^c}^* \|_2^2 + \sigma^2 \right).$$

For equation 14, using equations above, we have

$$\mathbb{E}_{X}\left[ \left\| \hat{w}_{\mathcal{T}} - w_{\mathcal{T}}^* \right\|_2^2 \right] = \left\| w_{\mathcal{T}}^* \right\|_2^2 \left( 1 - \tfrac{n}{p} \right) + \tfrac{n}{p - n - 1} \left( \| w_{\mathcal{T}^c}^* \|_2^2 + \sigma^2 \right).$$

$\square$

## A.2 Proof of Theorem 2

This section is dedicated to the proof of Theorem 2.

**Theorem 4.** *Let $(p, q) \in [\![1, d]\!]^2$ such that $p + q = d$, $\mathcal{T} \subseteq [d]$ an arbitrary subset of the $d$ first natural integers, and $\mathcal{T}^c := [d] \setminus \mathcal{T}$ its complement set. Let $\hat{w} \in \mathbb{R}^d$ such that $\hat{w}_{\mathcal{T}} = X_{\mathcal{T}}^+ y \in \mathbb{R}^p$ and $\hat{w}_{\mathcal{T}^c} = 0 \in \mathbb{R}^q$. If $(x, \cdot) \sim P_{\mathcal{X}, \mathcal{Y}}^{OOD}$, then the expected OOD risk of the predictor $\hat{f}_{\mathcal{T}} : x \mapsto \phi(x^T \hat{w}) \in \hat{\mathcal{F}}_d$ satisfies*

$$\mathbb{E}_{X}\left[ R_{OOD}(\hat{f}) \right] \geq \left( \lambda_{\min}(\Sigma) + \lambda_{\min}(\Sigma^{OOD}) \right) c(n, p, \sigma') + 2\sigma'^2$$

*and*

$$\mathbb{E}_{X}\left[ R_{OOD}(\hat{f}) \right] \leq \left( \lambda_{\max}(\Sigma) + \lambda_{\max}(\Sigma^{OOD}) \right) c(n, p, \sigma') + \sigma'^2,$$

*where $c(n, p, \sigma')$ is defined in equation 7, $\Sigma \in \mathbb{R}^{d \times d}$ is defined in equation 8, and $\Sigma^{OOD} \in \mathbb{R}^{d \times d}$ is defined as*

$$\Sigma^{OOD} = \mathbb{E}_{(x, \cdot) \sim P_{\mathcal{X}, \mathcal{Y}}^{OOD}}\left[ \left( \tfrac{\Phi(x^T \hat{w}) - \Phi(x^T w^*)}{x^T \hat{w} - x^T w^*} \right)^2 x x^T \right].$$

*Proof.* For ease of notation, we denote the weight vector $\boldsymbol{w}^*_{\text{OOD}}$ defined in equation 5 by $\boldsymbol{w}^*$. The layout of the proof is similar to the proof of Theorem 1. Let $\boldsymbol{x} \in \mathcal{X}$ and

$$\boldsymbol{\Delta}_{\boldsymbol{w}} = \hat{\boldsymbol{w}} - \boldsymbol{w}^*.$$

From the mean-value theorem, there exists

$$c(\boldsymbol{x}, \hat{\boldsymbol{w}}, \boldsymbol{w}^*) \in \big(\min(\boldsymbol{x}^T\boldsymbol{w}, \boldsymbol{x}^T\boldsymbol{w}^*), \max(\boldsymbol{x}^T\hat{\boldsymbol{w}}, \boldsymbol{x}^T\boldsymbol{w}^*)\big),$$

such that

$$\hat{f}_{\mathcal{T}}(\boldsymbol{x}) = \phi(\boldsymbol{x}^T\hat{\boldsymbol{w}}) = \phi\big(\boldsymbol{x}^T(\boldsymbol{w}^* + \boldsymbol{\Delta}_{\boldsymbol{w}})\big)$$
$$= \phi(\boldsymbol{x}^T\boldsymbol{w}^*) + \boldsymbol{x}^T\boldsymbol{\Delta}_{\boldsymbol{w}}\phi'\big(c(\boldsymbol{x}, \hat{\boldsymbol{w}}, \boldsymbol{w}^*)\big).$$

We have

$$\mathbb{E}_{\boldsymbol{X}}\big[R_{\text{OOD}}(\hat{f})\big] = \mathbb{E}_{(\boldsymbol{x},\cdot)\sim P_{\mathcal{X},\mathcal{Y}},\boldsymbol{X}}\big[(2\hat{f}(\boldsymbol{x}) - 1 - z(\boldsymbol{x}))^2\big] + \mathbb{E}_{(\boldsymbol{x},\cdot)\sim P^{\text{OOD}}_{\mathcal{X},\mathcal{Y}},\boldsymbol{X}}\big[(2\hat{f}(\boldsymbol{x}) - 1 - z(\boldsymbol{x}))^2\big]$$

$$= \mathbb{E}_{(\boldsymbol{x},\cdot)\sim P_{\mathcal{X},\mathcal{Y}},\boldsymbol{X}}\Big[\big(2\big(\phi(\boldsymbol{x}^T\hat{\boldsymbol{w}}) - \phi(\boldsymbol{x}^T\boldsymbol{w}^*)\big) - \epsilon'\big)^2\Big]$$

$$+ \mathbb{E}_{(\boldsymbol{x},\cdot)\sim P^{\text{OOD}}_{\mathcal{X},\mathcal{Y}},\boldsymbol{X}}\Big[\big((2\big(\phi(\boldsymbol{x}^T\hat{\boldsymbol{w}}) - \phi(\boldsymbol{x}^T\boldsymbol{w}^*)\big) - \epsilon'\big)^2\Big]$$

$$= \mathbb{E}_{(\boldsymbol{x},\cdot)\sim P_{\mathcal{X},\mathcal{Y}},\boldsymbol{X}}\Big[\big(2\boldsymbol{x}^T\boldsymbol{\Delta}_{\boldsymbol{w}}\phi'\big(c(\boldsymbol{x}, \hat{\boldsymbol{w}}, \boldsymbol{w}^*)\big)\big)^2\Big]$$

$$+ \mathbb{E}_{(\boldsymbol{x},\cdot)\sim P^{\text{OOD}}_{\mathcal{X},\mathcal{Y}},\boldsymbol{X}}\Big[\big(2\boldsymbol{x}^T\boldsymbol{\Delta}_{\boldsymbol{w}}\phi'\big(c(\boldsymbol{x}, \hat{\boldsymbol{w}}, \boldsymbol{w}^*)\big)\big)^2\Big] + 2\sigma'^2$$

$$= \text{Tr}\Big(\big(\mathbb{E}_{(\boldsymbol{x},\cdot)\sim P_{\mathcal{X},\mathcal{Y}}}\big[\phi'\big(c(\boldsymbol{x}, \hat{\boldsymbol{w}}, \boldsymbol{w}^*)\big)^2\boldsymbol{x}\boldsymbol{x}^T\big] + \mathbb{E}_{(\boldsymbol{x},\cdot)\sim P^{\text{OOD}}_{\mathcal{X},\mathcal{Y}}}\big[\phi'\big(c(\boldsymbol{x}, \hat{\boldsymbol{w}}, \boldsymbol{w}^*)\big)^2\boldsymbol{x}\boldsymbol{x}^T\big]\big)\mathbb{E}_{\boldsymbol{X}}\big[\boldsymbol{\Delta}_{\boldsymbol{w}}\boldsymbol{\Delta}_{\boldsymbol{w}}^T\big]\Big)$$

$$+ 2\sigma'^2$$

$$= \text{Tr}\Big(\big(\boldsymbol{\Sigma} + \boldsymbol{\Sigma}^{\text{OOD}}\big)\mathbb{E}_{\boldsymbol{X}}\big[\boldsymbol{\Delta}_{\boldsymbol{w}}\boldsymbol{\Delta}_{\boldsymbol{w}}^T\big]\Big) + 2\sigma'^2$$

$$= \mathbb{E}_{\boldsymbol{X}}\big[\boldsymbol{\Delta}_{\boldsymbol{w}}^T\boldsymbol{\Sigma}\boldsymbol{\Delta}_{\boldsymbol{w}} + \boldsymbol{\Delta}_{\boldsymbol{w}}^T\boldsymbol{\Sigma}^{\text{OOD}}\boldsymbol{\Delta}_{\boldsymbol{w}}\big] + 2\sigma'^2,$$

where

$$\boldsymbol{\Sigma} = \mathbb{E}_{(\boldsymbol{x},\cdot)\sim P_{\mathcal{X},\mathcal{Y}}}\big[\phi'\big(c(\boldsymbol{x}, \hat{\boldsymbol{w}}, \boldsymbol{w}^*)\big)^2\boldsymbol{x}\boldsymbol{x}^T\big]$$

$$= \mathbb{E}_{(\boldsymbol{x},\cdot)\sim P_{\mathcal{X},\mathcal{Y}}}\Bigg[\bigg(\frac{\Phi(\boldsymbol{x}^T\hat{\boldsymbol{w}})-\Phi(\boldsymbol{x}^T\boldsymbol{w}^*)}{\boldsymbol{x}^T\hat{\boldsymbol{w}}-\boldsymbol{x}^T\boldsymbol{w}^*}\bigg)^2\boldsymbol{x}\boldsymbol{x}^T\Bigg].$$

and

$$\boldsymbol{\Sigma}^{\text{OOD}} = \mathbb{E}_{(\boldsymbol{x},\cdot)\sim P^{\text{OOD}}_{\mathcal{X},\mathcal{Y}}}\big[\phi'\big(c(\boldsymbol{x}, \hat{\boldsymbol{w}}, \boldsymbol{w}^*)\big)^2\boldsymbol{x}\boldsymbol{x}^T\big]$$

$$= \mathbb{E}_{(\boldsymbol{x},\cdot)\sim P^{\text{OOD}}_{\mathcal{X},\mathcal{Y}}}\Bigg[\bigg(\frac{\Phi(\boldsymbol{x}^T\hat{\boldsymbol{w}})-\Phi(\boldsymbol{x}^T\boldsymbol{w}^*)}{\boldsymbol{x}^T\hat{\boldsymbol{w}}-\boldsymbol{x}^T\boldsymbol{w}^*}\bigg)^2\boldsymbol{x}\boldsymbol{x}^T\Bigg].$$

From the min-max theorem, we have

$$\mathbb{E}_{\boldsymbol{X}}\big[\boldsymbol{\Delta}_{\boldsymbol{w}}^T\boldsymbol{\Sigma}\boldsymbol{\Delta}_{\boldsymbol{w}} + \boldsymbol{\Delta}_{\boldsymbol{w}}^T\boldsymbol{\Sigma}^{\text{OOD}}\boldsymbol{\Delta}_{\boldsymbol{w}}\big] \geq \big(\lambda_{\min}(\boldsymbol{\Sigma}) + \lambda_{\min}(\boldsymbol{\Sigma}^{\text{OOD}})\big)\mathbb{E}_{\boldsymbol{X}}\big[\boldsymbol{\Delta}_{\boldsymbol{w}}^T\boldsymbol{\Delta}_{\boldsymbol{w}}\big]$$

and

$$\mathbb{E}_{\boldsymbol{X}}\big[\boldsymbol{\Delta}_{\boldsymbol{w}}^T\boldsymbol{\Sigma}\boldsymbol{\Delta}_{\boldsymbol{w}} + \boldsymbol{\Delta}_{\boldsymbol{w}}^T\boldsymbol{\Sigma}^{\text{OOD}}\boldsymbol{\Delta}_{\boldsymbol{w}}\big] \leq \big(\lambda_{\max}(\boldsymbol{\Sigma}) + \lambda_{\max}(\boldsymbol{\Sigma}^{\text{OOD}})\big)\mathbb{E}_{\boldsymbol{X}}\big[\boldsymbol{\Delta}_{\boldsymbol{w}}^T\boldsymbol{\Delta}_{\boldsymbol{w}}\big].$$

From Lemma 4.1, we have $\lambda_{\min}(\boldsymbol{\Sigma}) > 0$. For $\mathbb{E}_{\boldsymbol{X}}\big[\boldsymbol{\Delta}_{\boldsymbol{w}}^T\boldsymbol{\Delta}_{\boldsymbol{w}}\big]$, we have

$$\mathbb{E}_{\boldsymbol{X}}\big[\boldsymbol{\Delta}_{\boldsymbol{w}}^T\boldsymbol{\Delta}_{\boldsymbol{w}}\big] = \mathbb{E}_{\boldsymbol{X}}\Big[\|\hat{\boldsymbol{w}} - \boldsymbol{w}^*\|_2^2\Big]$$

$$= \mathbb{E}_{\boldsymbol{X}}\Big[\|\hat{\boldsymbol{w}}_{\mathcal{T}} - \boldsymbol{w}^*_{\mathcal{T}}\|_2^2\Big] + \mathbb{E}_{\boldsymbol{X}}\Big[\|\hat{\boldsymbol{w}}_{\mathcal{T}^c} - \boldsymbol{w}^*_{\mathcal{T}^c}\|_2^2\Big]$$

$$= \mathbb{E}_{\boldsymbol{X}}\Big[\|\hat{\boldsymbol{w}}_{\mathcal{T}} - \boldsymbol{w}^*_{\mathcal{T}}\|_2^2\Big] + \|\boldsymbol{w}^*_{\mathcal{T}^c}\|_2^2.$$

as $\hat{\boldsymbol{w}}_{\mathcal{T}^c} = \boldsymbol{0}$. The remainder of the proof follows the proof of Theorem 1 with $\boldsymbol{\eta} = \boldsymbol{y} - \boldsymbol{X}_{\mathcal{T}}\boldsymbol{w}^*_{\mathcal{T}}$, where $\boldsymbol{w}^*_{\text{OOD}} = \boldsymbol{w}^*$. $\qquad\square$

**Lemma 4.1.** Under Assumption 4.1, the matrix

$$\mathbf{\Sigma} = \mathbb{E}_{(\boldsymbol{x},\cdot)\sim P_{\mathcal{X},\mathcal{Y}}}\left[\left(\frac{\Phi(\boldsymbol{x}^T\hat{\boldsymbol{w}})-\Phi(\boldsymbol{x}^T\boldsymbol{w}^*)}{\boldsymbol{x}^T\hat{\boldsymbol{w}}-\boldsymbol{x}^T\boldsymbol{w}^*}\right)^2\boldsymbol{x}\boldsymbol{x}^T\right]$$

is nonsingular.

*Proof.* From the mean-value theorem, there exists

$$c(\boldsymbol{x},\hat{\boldsymbol{w}},\boldsymbol{w}^*) \in \big(\min(\boldsymbol{x}^T\boldsymbol{w},\boldsymbol{x}^T\boldsymbol{w}^*),\max(\boldsymbol{x}^T\hat{\boldsymbol{w}},\boldsymbol{x}^T\boldsymbol{w}^*)\big),$$

such that

$$\phi'\big(c(\boldsymbol{x},\hat{\boldsymbol{w}},\boldsymbol{w}^*)\big) = \frac{\Phi(\boldsymbol{x}^T\hat{\boldsymbol{w}})-\Phi(\boldsymbol{x}^T\boldsymbol{w}^*)}{\boldsymbol{x}^T\hat{\boldsymbol{w}}-\boldsymbol{x}^T\boldsymbol{w}^*}.$$

Using $c(\boldsymbol{x},\hat{\boldsymbol{w}},\boldsymbol{w}^*)$, we rewrite the matrix $\mathbf{\Sigma}$ as

$$\mathbf{\Sigma} = \mathbb{E}_{(\boldsymbol{x},\cdot)\sim P_{\mathcal{X},\mathcal{Y}}}\big[\phi'\big(c(\boldsymbol{x},\hat{\boldsymbol{w}},\boldsymbol{w}^*)\big)^2\boldsymbol{x}\boldsymbol{x}^T\big].$$

The matrix $\mathbf{\Sigma}$ is semi-positive-definite. We want to show that $\mathbf{\Sigma}$ is nonsingular. From the min-max theorem, the matrix $\mathbf{\Sigma}$ is nonsingular iff for any $\boldsymbol{a} \in \mathbb{R}$, we have

$$\boldsymbol{a}^T\mathbf{\Sigma}\boldsymbol{a} > 0.$$

Since the activation function $\Phi(\cdot)$ is strictly monotonically non-decreasing, there exists $\epsilon > 0$ such that, for any $\boldsymbol{a} \in \mathbb{R}$, we have $\Phi'(\boldsymbol{a}) \geq \epsilon$. Therefore,

$$\boldsymbol{a}^T\mathbf{\Sigma}\boldsymbol{a} = \boldsymbol{a}^T\mathbb{E}_{(\boldsymbol{x},\cdot)\sim P_{\mathcal{X},\mathcal{Y}}}\big[\boldsymbol{x}\boldsymbol{x}^T\phi'c(\boldsymbol{x},\hat{\boldsymbol{w}},\boldsymbol{w}^*)^2\big]\boldsymbol{a} \geq \epsilon^2\boldsymbol{a}^T\mathbb{E}_{(\boldsymbol{x},\cdot)\sim P_{\mathcal{X},\mathcal{Y}}}\big[\boldsymbol{x}\boldsymbol{x}^T\big]\boldsymbol{a} = \epsilon^2\|\boldsymbol{a}\|_2^2 > 0.$$

$\square$

**Lemma 4.2** (Normal Concentration). ((Ledoux, 2001, Corollary 2.6, Propositions 1.3, 1.8) or (Tao, 2012, Theorem 2.1.12)) For $d \in \mathbb{N}$, consider $\mu$ the canonical Gaussian probability on $\mathbb{R}^d$ defined through its density $d\mu(\boldsymbol{w}) = (2\pi)^{-\frac{d}{2}}e^{-\frac{1}{2}\|\boldsymbol{w}\|^2}$ and $f : \mathbb{R}^d \to \mathbb{R}$ a $L_f$-Lipschitz function. Then

$$Pr\left(\left\{\left|f - \int f d\mu\right| \geq t\right\}\right) \leq Ce^{-c\frac{t^2}{L_f^2}}, \tag{15}$$

where $C, c > 0$ are independent of $d$ and $L_f$.

# B    DETAILS ABOUT BASELINES AND ARCHITECTURES IMPLEMENTATION

## B.1    BASELINES OOD METHODS

In this section, we present an overview of the baseline methods used in our experiments.We describe the principles behind these baselines, and the chosen hyperparameters. It is worth noting that extensive hyperparameter search for each method were not performed to maintain stability. Hence, once the final model is selected, hybrid methods like ViM, ASH and NECO performance may increase if such task is performed

**Softmax Score.**    This score uses the maximum softmax probability (MSP) of the model as an OOD scoring function (Hendrycks & Gimpel, 2017).

**Energy.**    Liu et al. (2020) proposes using the energy score for OOD detection, where the energy function maps the logit outputs to a scalar. To maintain the convention that lower scores correspond to in-distribution (ID) data, (Liu et al., 2020) uses the negative energy as the OOD score.

**ReAct.**    Sun et al. (2021) propose clipping extreme-valued activations. The original paper found that clipping activations at the $90th$ percentile of ID data was optimal. Moreover, as the authors propose, we employ the ReAct+Energy configuration.

**KL-Matching & MaxLogit.** KL-Matching computes the class-wise average probability using the entire training dataset. Consistent with the approach outlined in (Hendrycks et al., 2022), this calculation is based on the predicted class rather than the ground-truth labels. MaxLogit employs the maximum logit value of the model as an OOD scoring function.

**Mahalanobis.** This score leverages the feature vector from the layer preceding the final classification layer (Lee et al., 2018a). To estimate the precision matrix and the class-wise mean vector, we used the entire training dataset. It's important to note that we incorporated ground-truth labels during this computation process.

**ViM & Residual.** Wang et al. (2022) decomposes the latent space into a principal space $P$ and a null space $P^\perp$. The ViM score is calculated by projecting the features onto the null space to create a virtual logit, which is then combined with the logits using the norm of this projection. To enhance performance, they calibrate this norm with a constant which is determined by dividing the sum of the maximum logits by the sum of the norms of the null space projections, both measured on the training set. The Residual score is derived by computing the norm of the latent vector's projection onto the null space. We followed the author's suggestions for the null space, by setting it to half the size of the full feature vector, adapted to each model width.

**ASH.** Djurisic et al. (2023) employs activation pruning at the penultimate layer, just before the application of the DNN classifier. This pruning threshold is determined on a per-sample basis, eliminating the need for pre-computation of ID data statistics. The original paper presents three different post-hoc scoring functions, with the only distinction among them being the imputation method applied after pruning. We employ ASH-P in our experiments as it performed the best, in which the clipped values are replaced with zeros. As specified in the original paper, we fix the pruning threshold value to $90\%$.

**NECO.** Ammar et al. (2024) leverages the geometric properties of Neural Collapse, measuring the relative norm of a sample within the subspace defined by the ETF to identify OOD samples. NC typically involves a collapse in the variability of class representations, leading to a more structured and simplified feature space. It is hypothesized that this collapse also impacts OOD detection, particularly through the emerging orthogonality between ID and OOD data. NECO utilizes this orthogonality to effectively distinguish between ID and OOD data by measuring the relative norm of each data point within the approximated ETF space scaled by the maximum logit value as the OOD score. We use a dimension $d = c$ to approximate the ETF sub-space for all architectures, with $c$ being the number of classes.

### B.2 EXPERIMENTS SETUP

**ViT Experimental Setup.** For all experiments, we trained a set of ViT models with widths [4, 8, 12, 16, 20, 24, 28, 32, 36, 40, 60, 80, 100, 120, 160, 200, 240, 280, 320, 360, 400, 480, 520, 600, 680, 760, 800]. The width is used as the last dimension of the output layer after the linear transformation (the class-token size). The dimension of the FeedForward layer is the width multiplied by 4. The input size is set to 32 and the patch size to 8, no dropout is used and we use 4 heads with 4 Transformer blocks. The ViT models are first randomly initialized and then trained on CIFAR-10 using stochastic gradient descent with CE loss. The weights are fine-tuned for 60 000 steps, with no warm-up steps, 1 024 batch size, 0.9 momentum, and a learning rate of 0.03.

**Swin Experimental Setup.** We used a standard 4 block Swin architecture, with a downscaling factor of (2,2,2,1) for each block respectively. The width ranges from 1 to 100, with a window size of 4, an input size of 32, and a filter size of 4. The model is randomly initialized and then optimized using an Adam optimizer with CE loss for a 1 000 epoch using a batch size of 1 024. The initial learning rate is 0.0001.

**CNN Experimental Setup.** Similar to Nakkiran et al. (2021), we define a standard family CNN models formed by 4 convolutional stages of controlled base width $[k, 2k, 4k, 8k]$, for $k$ in the range of [1, 128], with a fully connected layer as classifier. The MaxPool is set to [2, 2, 2, 4] for the four blocks respectively. For all the convolution layers, the kernel size is set to 3, stride and padding to 1.

# C    Details about Neural collapse

For overparametrised model trained through the terminal phase of training (TPT), Neural Collapse (NC) phenomenon emerges, particularly in the penultimate layer and in the linear classifier of DNNs (Papyan et al., 2020; Ammar et al., 2024). NC is characterized by five main properties:

1. **Variability Collapse (NC1):** the within-class variation in activations becomes negligible as each activation collapses toward its respective class mean.

2. **Convergence to Simplex ETF (NC2):** the class-mean vectors converge to having equal lengths, as well as having equal-sized angles between any pair of class means. This configuration corresponds to a Simplex Equiangular Tight Frame (ETF).

3. **Convergence to Self-Duality (NC3):** in the limit of an ideal classifier, the class means and linear classifiers of a neural network converge to each other up to rescaling, implying that the decision regions become geometrically similar and that the class means lie at the centers of their respective regions.

4. **Simplification to Nearest Class-Center (NC4):** The network classifier progressively tends to select the class with the nearest class mean for a given activation, typically based on standard Euclidean distance.

5. **ID/OOD Orthogonality (NC5):** As the training procedure advances, OOD and ID data tend to become increasingly more orthogonal to each other. In other words, the clusters of OOD data become more perpendicular to the configuration adopted by ID data (*i.e.*, the Simplex ETF).

These NC properties provide valuable insights into DNNs learned representation structure and properties, which allows for a considerable simplification. Additionally, the convergence of NC can be linked to OOD detection Ammar et al. (2024); Haas et al. (2023); Zhao & Cao (2023); Zhang et al. (2024). For further details refer to (Papyan et al., 2020; Ammar et al., 2024)

# D    Complementary Results on Double descent and OOD detection

## D.1    CIFAR-10 additional results

To further show the consistency of double descent for OOD detection, Figures D.1, D.2 D.3, D.4 and D.5 show the OOD detection metrics performance on six more semantic-shift OOD datasets. To illustrate the performance of other OOD-methods while maintaining visibility, we show two different methods at each dataset alongside the better-performing and most stable three: MSP, NECO, and ASH.

## D.2    CIFAR-100 results

In this section, we present results for the CIFAR-100 dataset as ID for a ResNet-18 model. Figure D.6 illustrates the OOD detection metrics performance, Figure D.7 shows the accuracy and eigenvalues distribution (see section E.1 for discussion about eigenvalues). We can observe similar behaviors between the ResNet-18 trained on CIFAR-10, and this current configuration on a harder dataset (CIFAR-100).

Table 2 illustrates the evolution of $AUC$ between the underparametrized and overparametrized regime and its correlation with the $NC1_{u/o}$ for the remaining OOD datasets. As the CNN's $NC1_{u/o} < 1$, its performance stagnates or deteriorates with overparametrization, while the other models improve. Additionally, we can see how the hybrid-based methods improve considerably and become competitive with logit-methods when $NC1_{u/o} > 1$.

## D.3    ResNet-34 results

In this section, we present the results for the ResNet-34 architecture, a deeper version of the ResNet family. Results include both CIFAR-10 and CIFAR-100 datasets as ID. Figure D.9 shows the accu-

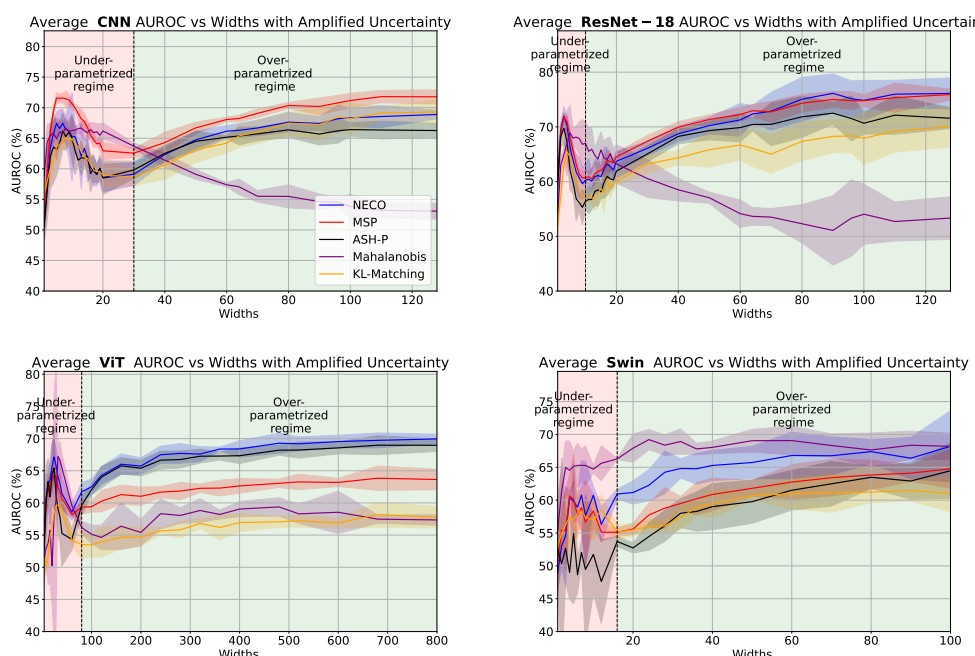

Figure D.1: OOD detection evolution curve w.r.t model's width (x-axis) in terms AUC. With CIFAR10 as ID and ImageNet-O as OOD, for (from top-left to bottom-right) CNN, ResNet-18, ViT, and Swin.

racy curve and Figure D.8 illustrates the OOD detection metrics performance, We observe a similar curve to that of ResNet-18 for both datasets and for all OOD methods, with slightly higher performances. We highlight that the interpolation threshold occurs at a smaller width for ResNet-34 (k=8), compared to ResNet-18 (k=10) for the cifar 10 case. This is simply due to the increased complexity of ResNet-34 at similar width values, due to its increased width, further highlighting the dependence of the interpolation threshold on model capacity. This higher complexity contributes to its lower NC1 values in Figure E.2.

## D.4 OUT-OF-DISTRIBUTION RISQUE EXPERIMENTS

In Section 4.1, we introduced $R_{\text{OOD}}(\hat{f})$ as a way to assess the expected OOD detection performance. Here, we present the evolution curve for this risk, which closely mirrors the double descent behavior across all OOD datasets. Figures D.10, D.11, D.12, D.13, and D.14 illustrate these results for each model.

## D.5 EXPERIMENTS WITHOUT LABEL-NOISE RESULTS

Usually, double descent is most pronounced when there is a mismatch between model complexity and data quality (Nakkiran et al., 2021). Introducing label noise accentuates this effect by increasing the effective complexity needed for the model to fit the training data, particularly in the over-parameterized regime. However, it is also valuable to examine results in noiseless settings. Figure D.15 illustrates the model's accuracy in a noiseless setup. Instead of the characteristic peak in generalization error, a plateau or stagnation appears near the interpolation threshold. Similarly, Figure D.16 presents the OOD detection performance under this setup, comparing a semantic shift case (CIFAR-10 vs. SUN) and a covariate shift case (CIFAR-10 vs. CIFAR-100). The curves exhibit a pattern similar to the model accuracy. Moreover, we highlight that removing label noise makes

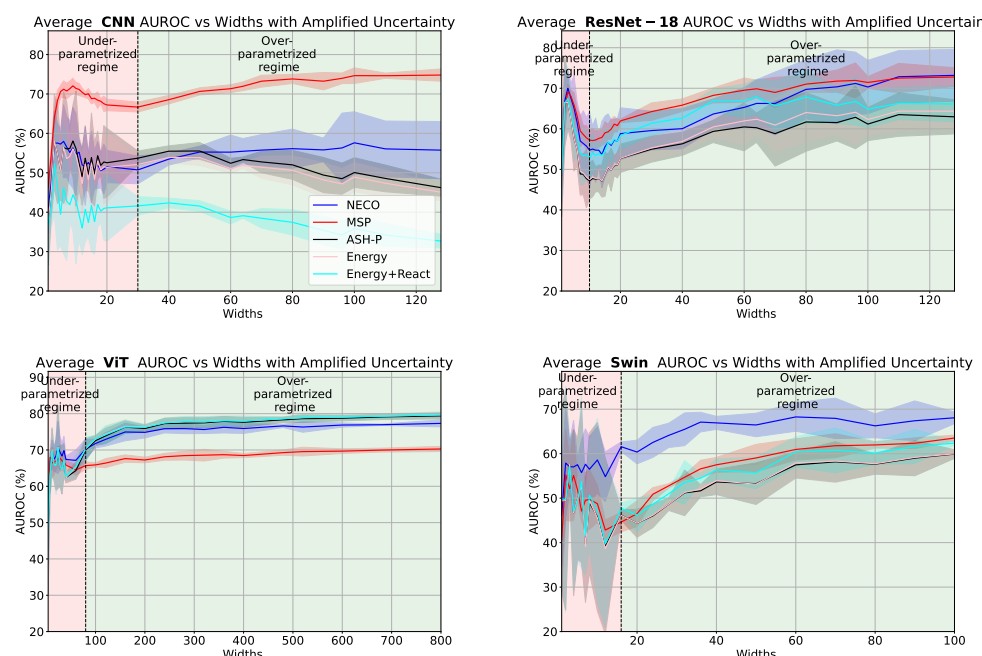

Figure D.2: OOD detection evolution curve w.r.t model's width (x-axis) in terms AUC. With CIFAR10 as ID and Textures as OOD, for (from top-left to bottom-right) CNN, ResNet-18, ViT and Swin.

the learning task easier, as all models' performance metrics have increased, such as ResNet-18 max accuracy rising from 83.40% to 94.48%, with a similar rise in AUC.

# E  MODEL REPRESENTATIONS AND NEURAL COLLAPSE ANALYSIS

## E.1  STRUCTURE OF THE MODEL REPRESENTATION

Convergence towards Neural Collapse is an indicator of improved model representation, as defined by the ETF structure. As such, the model eigenvalues distribution describes how much the model representation aligns with this structured manifold. The properties of the ETF implies that the top $c$ eigenvalues are equally prominent, while the remaining eigenvalues are less influential. Figure E.1 shows the distribution of each model eigenvalues at the overparametrized regime. It is worth noting that all overparametrized model widths show similar curves per-architecture. ResNet-18 and Swin models follow the expected NC pattern, with a steep drop in importance at the $c^{th}$ eigenvalue indicating the limit of the ETF. However, ViT and CNN show a slowly decaying curve, which indicates a lack of clear separation in the model representation between highly important ID information and noisy features. This lack of a global structure in their representation, results in both models failing to reliably outperform their underpametrized minima.

This highlights the importance of the ETF structure and NC convergence in enhancing representation stability, which can be useful for improving ID classification and OOD detection tasks.

## E.2  EVOLUTION OF NC1

In this section, we further analyse the evolution of NC1 and the model's learned representation with overparametrization, to further show their correlation. Our analysis will focus primarily on the ResNet and CNN models, due to their similarities. We will not address the transformer-based models whose performance, especially for generalization, were lower than those of ResNet or CNN. This is because transformers typically require extensive pre-training, particularly for small datasets, and this was not the case for our experiments.

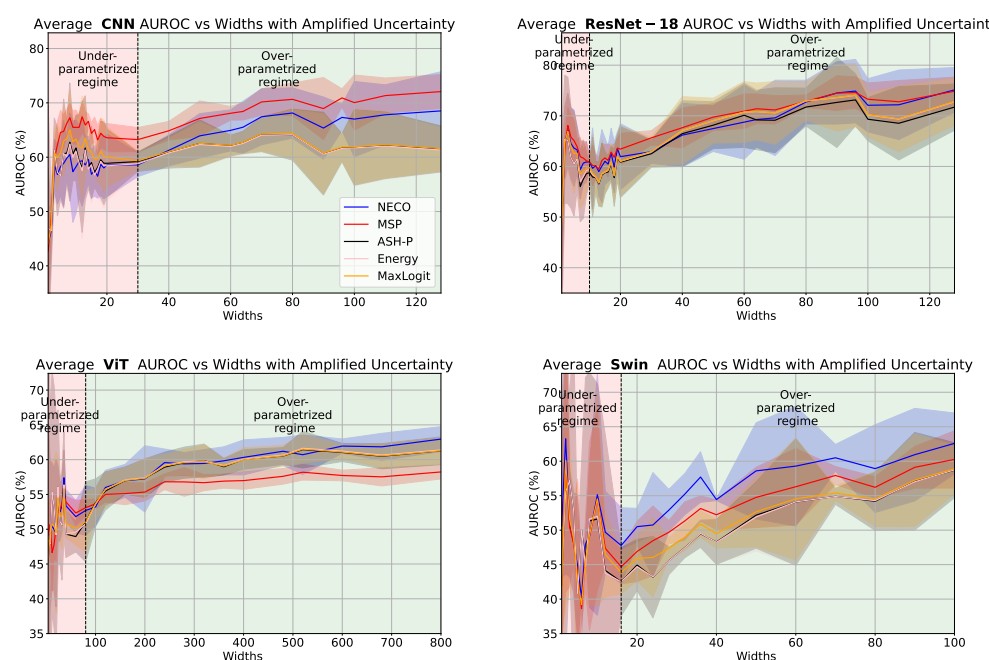

Figure D.3:    OOD detection evolution curve w.r.t model's width (x-axis) in terms `AUC`. With CIFAR10 as ID and iNaturalist as OOD, for (from top-left to bottom-right) CNN, ResNet-18, ViT and Swin.

In order to visualize the variability collapse predicted by NC1, Figure E.2 shows the last-layer activations for both models at their optimal underparameterized and overparameterized widths. In ResNet, transitioning to overparameterized models leads to significant improvements in the compactness and separation of ID clusters, as well as enhanced orthogonality with OOD points. In contrast, the CNN model does not show clear improvements in ID compactness or OOD separation, making it difficult to determine which representation is better.

The same phenomenon is shown in Figure E.2, where the NC1 metric is shown against the model widths for both ResNet and CNN. While both models exhibit a double descent pattern, CNN barely matches its underparameterized metric value, whereas ResNet continuously improves with added complexity. This discrepancy in NC convergence with overparametrization explains why only ResNet benefits from increased complexity, suggesting that without improvement in NC, increasing model complexity provides no benefits for the learned representations.

### E.3    MAHALANOBIS AND RESIDUAL JOINT PERFORMANCE

We noticed that for all architectures, and on all OOD datasets, Mahalanobis and Residual follow the same evolution curve, with usually slightly higher *AUC* in favor of Mahalanobis. This behaviour is intriguing, due to the fact that each method relies on different types of information. While Mahalanobis models the ID distribution, *i.e.*, the principal space, Residual relies on computing the null space norm, which is orthogonal to the principal space.

We associate this behavior with the noise isolation in each architecture, which is specific to the double descent training paradigm. Indeed, in order for models to be able to perfectly interpolate all the training data and achieve (almost) zero training error, noisy samples must be represented closer to their assigned (noisy) label, rather than to their true label. This will cause the train class clusters (using the true labels) to be less compact and separable, making their high-likelihood region to span almost the entire principal space, in which the ID data is represented. Hence, to separate ID from OOD, learning the Mahalanobis GMM (fitted on the train data) becomes equivalent to separating the principal and null space, which is the same reasoning behind the Residual score.

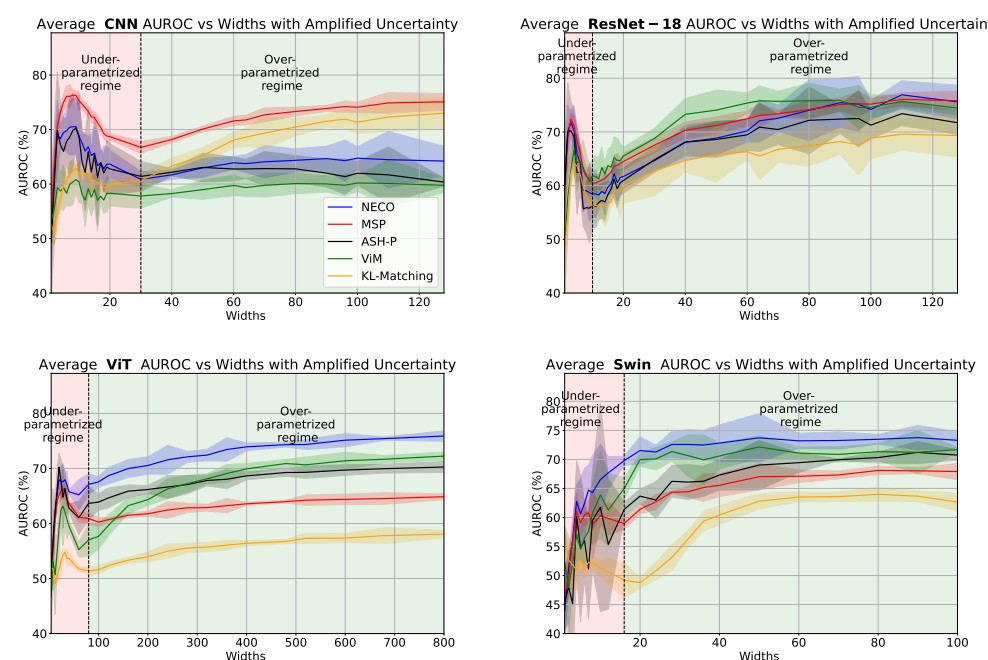

Figure D.4: OOD detection evolution curve w.r.t model's width (x-axis) in terms AUC. With CI-FAR10 as ID and SUN as OOD, for (from top-left to bottom-right) CNN, ResNet-18, ViT, and Swin.

This overfitting occurs at the interpolation threshold, which causes the learned distributions by Mahalanobis to be sparse and not robust to OOD data, impeding its improvement as we transition towards overparametrization. It is important to note also that both of these methods are usually below, or struggle to surpass the random choice threshold of $0.5$ *AUC* in the overparametrized regime (with the exception of texture dataset on ResNet-18 case).

Interestingly, both of these methods suffer much less from this behaviour under the Transformer based architecture, and even exhibit a double descent curve on most datasets. This can be explained by the fact that even the most overparametrized Transformer variant have an error higher than 4%, considerably higher than the training error lower than $0.01\%$ that convolutional models consistently achieve. Hence, Transformers suffer less from this effect because they have not interpolated the noise in the training data perfectly. It is worth noting that interpolating the noise is desirable, as it is necessary for generalisation in this setup (Bartlett et al., 2020). Transformer-based architectures require extensive pre-training to generalise well, especially for small scale dataset, which was not performed in our experiments. This inability of transformers to perfectly interpolate the training data contributes to their lower performance in terms of generalisation in the overparametrized regime, especially in the ViT case.

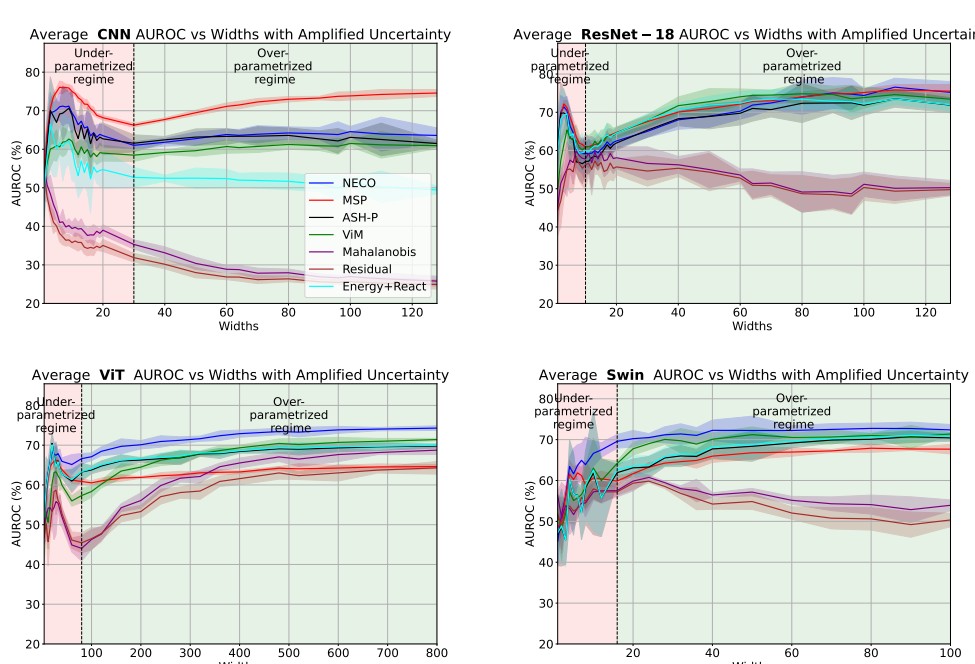

Figure D.5: OOD detection evolution curve w.r.t model's width (x-axis) in terms `AUC`. With CIFAR10 as ID and places365 as OOD, for (from top-left to bottom-right) CNN, ResNet-18, ViT and Swin.

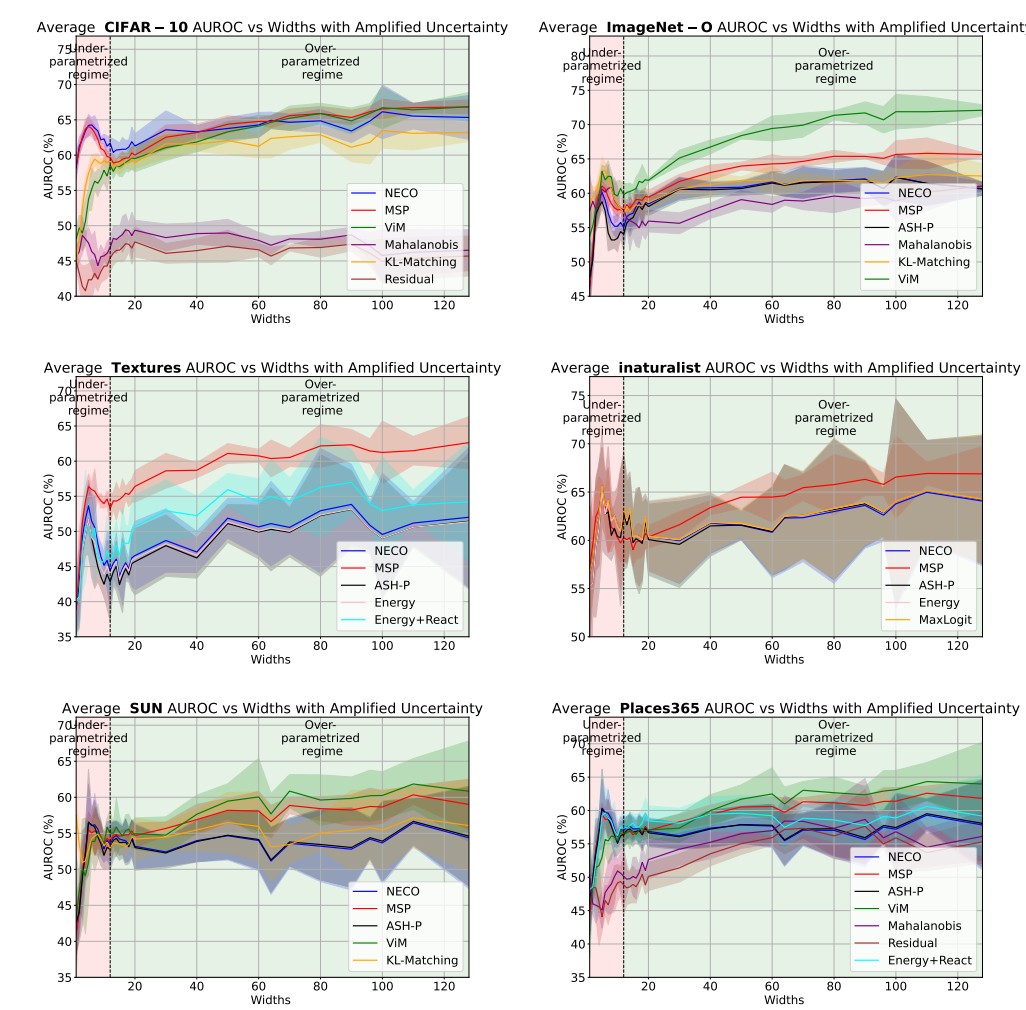

Figure D.6: OOD detection evolution curve w.r.t model's width (x-axis) in terms AUC. For a ResNet-18 model with CIFAR-100 as ID and (from top-left to bottom-right) CIFAR-10, ImageNet-O, Texture, iNaturalist, SUN and places365 as OOD.

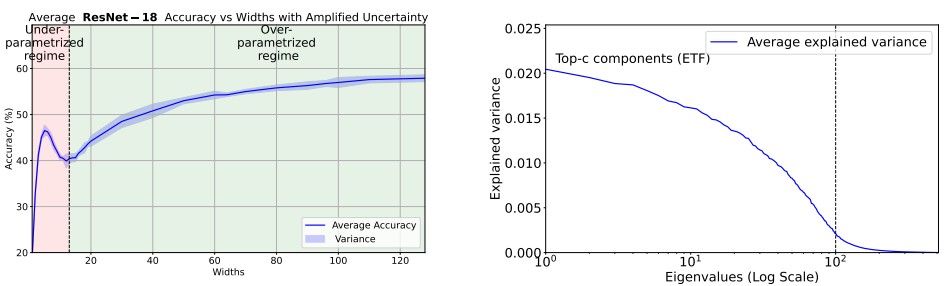

Figure D.7: Generalisation evolution curve (left) w.r.t model's width (x-axis), for a ResNet-18 model with CIFAR-100 as ID. Eigenvalues explained variance distribution (right) in the over-parametrized regime for ResNet-18 (width 64), with CIFAR-100 as ID. Black line represents the $100^{th}$ eigenvalue.

Table 2: Models performance in terms of AUC in the underparametrized local minima ($\text{AUC}_u$) and the overparametrized maximum width ($\text{AUC}_o$), *w.r.t* $NC1_{u/o}$ value. Best is highlighted in green when $\text{AUC}_u$ is higher, red when $\text{AUC}_u$ is higher and blue if both AUC are within standard deviation range. The highest AUC value per-dataset and per-architecture is highlighted in **bold**.

| Model | $NC1_{u/o}$ | Method | ImageNet-O | | Textures | | iNaturalist | |
|---|---|---|---|---|---|---|---|---|
| | | | $\text{AUC}_u\uparrow$ | $\text{AUC}_o\uparrow$ | $\text{AUC}_u\uparrow$ | $\text{AUC}_o\uparrow$ | $\text{AUC}_u\uparrow$ | $\text{AUC}_o\uparrow$ |
| CNN | 0.88 | Softmax score | 71.60+0.37 | **71.78±0.57** | 70.76±0.99 | **74.80±0.78** | 66.19±1.27 | **72.07±1.56** |
| | | MaxLogit | 68.91±0.96 | 65.94±0.51 | 62.01±3.75 | 45.56±1.06 | 63.38±1.89 | 61.58±2.07 |
| | | Energy | 65.06±1.57 | 65.76 ±0.51 | 53.62±5.87 | 45.07±1.04 | 60.11±2.32 | 61.32±2.05 |
| | | Energy+ReAct | 59.06±2.60 | 58.08±0.80 | 42.80±7.54 | 32.67±0.86 | 51.17±2.82 | 49.41±2.47 |
| | | NECO | 67.45±1.14 | 68.89±0.35 | 56.54±3.83 | 55.75±3.63 | 59.79±1.96 | 68.52±3.60 |
| | | ViM | 67.13±1.61 | 65.76±0.51 | 49.84±3.59 | 45.09±0.79 | 51.58±2.82 | 61.37±1.96 |
| | | ASH-P | 66.20±1.48 | 66.26±0.57 | 56.07±5.94 | 46.23±1.07 | 60.73±2.83 | 61.52±2.11 |
| ResNet | 1.96 | Softmax score | 70.90±0.52 | 75.91±0.49 | 68.03±0.75 | 72.78 ±1.14 | 65.79±2.16 | 74.85±1.39 |
| | | MaxLogit | 69.45±0.88 | 72.50±0.88 | 65.06±1.80 | 64.43±2.04 | 63.88±3.47 | 72.82±2.32 |
| | | Energy | 67.36±1.26 | 72.44±0.89 | 62.28±2.55 | 64.35±1.84 | 61.73±4.97 | 72.77±2.33 |
| | | Energy+ReAct | 68.02±1.41 | 72.00±0.81 | 64.15±2.68 | 66.13±1.20 | 61.68±6.83 | 71.85±2.60 |
| | | NECO | 70.42±0.83 | **76.11±1.42** | 67.56±1.84 | 73.18±3.20 | 64.50±3.51 | **75.12±2.21** |
| | | ViM | 70.94±1.54 | 75.03±0.62 | 77.88±1.66 | **81.02±1.42** | 62.56±4.91 | 67.12±2.44 |
| | | ASH-P | 67.36±1.26 | 71.57±0.94 | 62.28±2.55 | 62.96±2.11 | 61.73±4.97 | 71.72±2.28 |
| Swin | 1.70 | Softmax score | 56.62±3.15 | 64.78±1.48 | 49.71±3.38 | 63.51±0.33 | 49.10±6.81 | 60.26±2.08 |
| | | MaxLogit | 55.86±3.95 | 64.70±2.07 | 49.47±4.18 | 60.29±0.38 | 49.22±5.07 | 58.91±1.81 |
| | | Energy | 49.58±5.48 | 64.56±2.05 | 49.49±6.73 | 59.96±0.46 | 51.61±7.96 | 58.73±1.75 |
| | | Energy+ReAct | 49.81±5.51 | 65.43±2.09 | 50.66±6.07 | 62.38±0.24 | 51.87±7.51 | 59.39±1.83 |
| | | NECO | 57.63±4.14 | 68.19±2.71 | 56.50±3.64 | 68.06±0.66 | 49.09±5.14 | 62.58±2.21 |
| | | ViM | 65.18±1.83 | **73.45±2.25** | **84.47±1.46** | 78.67±1.65 | **67.86±2.44** | 63.83±2.63 |
| | | ASH-P | 49.46±5.66 | 64.48±2.11 | 48.96±7.85 | 59.90±0.48 | 51.33±10.79 | 58.69±1.99 |
| ViT | 2.32 | Softmax score | 64.17±0.64 | 63.64±0.80 | 67.73±1.82 | 70.27±0.36 | 52.79±0.77 | 58.23±0.51 |
| | | MaxLogit | 63.15±0.88 | 68.87±0.52 | 67.22±2.25 | 79.25±0.38 | 51.90±1.95 | 61.28±0.94 |
| | | Energy | 61.24±1.08 | 69.10±0.50 | 65.59±2.78 | 79.68±0.38 | 51.11±3.04 | 61.40±0.97 |
| | | Energy+ReAct | 61.30±1.21 | 69.09±0.49 | 65.64±2.86 | **79.68±0.38** | 51.63±4.61 | 61.39±0.98 |
| | | NECO | 65.83±1.35 | **69.95±0.37** | 69.41±2.16 | 77.32±0.49 | 52.50±2.05 | **62.96±0.89** |
| | | ViM | 68.76±1.52 | 67.82±0.55 | 65.58±2.69 | 74.13±0.38 | 56.41±3.81 | 60.42±0.90 |
| | | ASH-P | 61.24±1.08 | 68.96±0.48 | 65.59±2.78 | 79.39±0.42 | 51.11±3.04 | 61.28±0.99 |

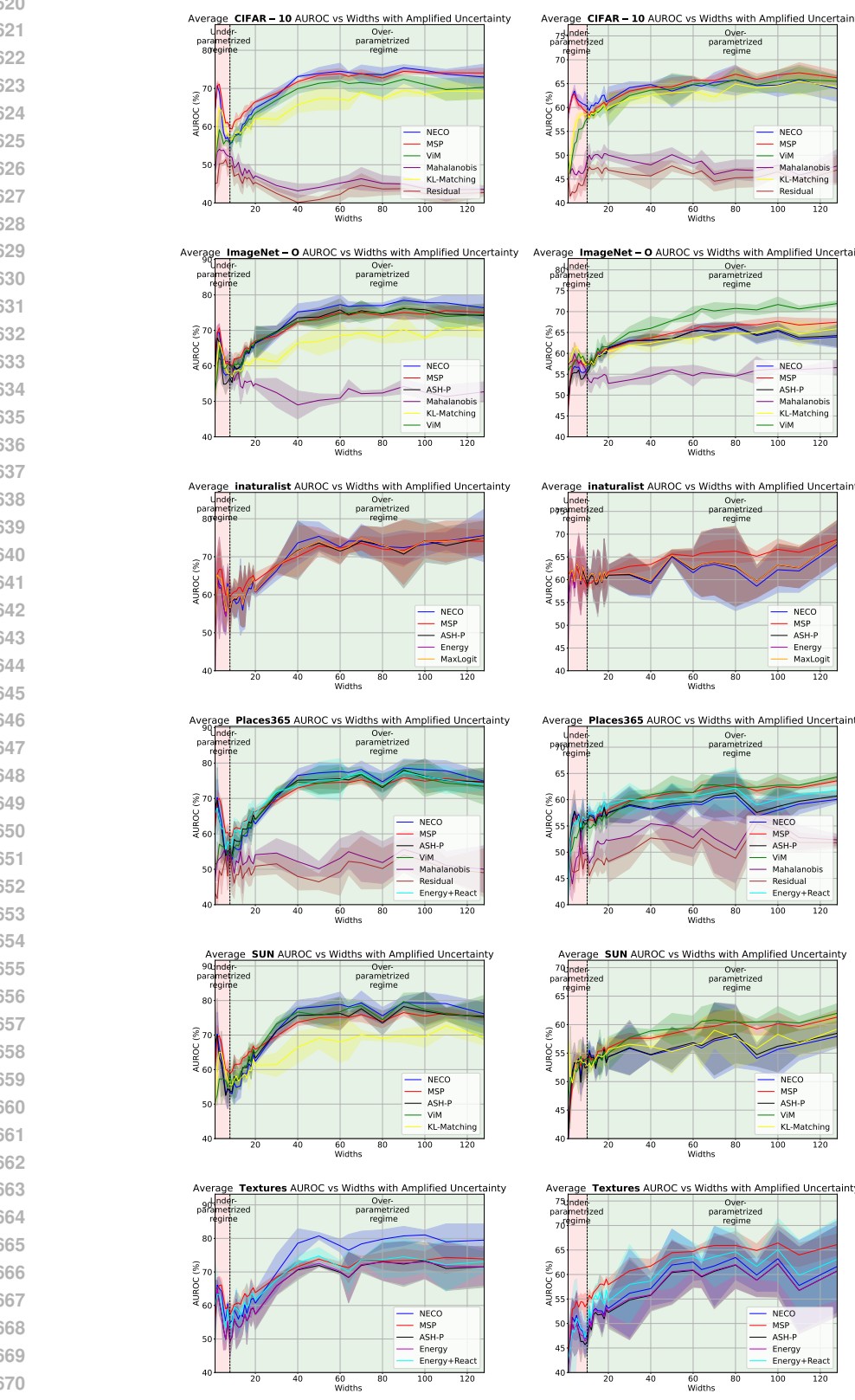

Figure D.8: OOD detection evolution curve w.r.t model's width (x-axis) in terms AUC. For a ResNet-34 model with CIFAR-10 as ID (left) and CIFAR-100 as ID (right). OOD datasets are CIFAR-10, ImageNet-O, iNaturalist, SUN, places365 and Textures.

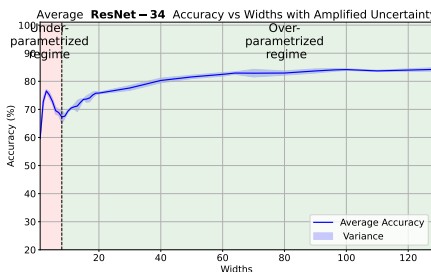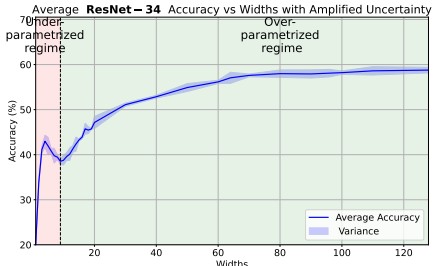

Figure D.9: Generalization evolution curve w.r.t model's width (x-axis), for a ResNet-34 model with CIFAR-10 as ID (left) and CIFAR-100 as ID (right).

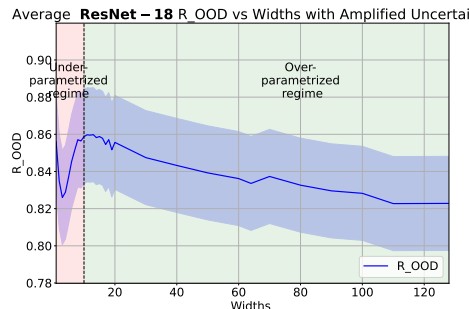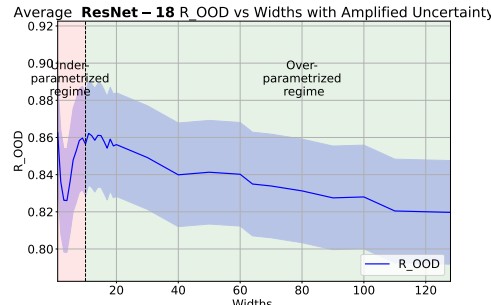

Figure D.10: OOD risk evolution curve w.r.t model's width (x-axis). For a ResNet-18 model with CIFAR-10 as ID and CIFAR-100 as OOD (left) and ImageNet-O as OOD (right).

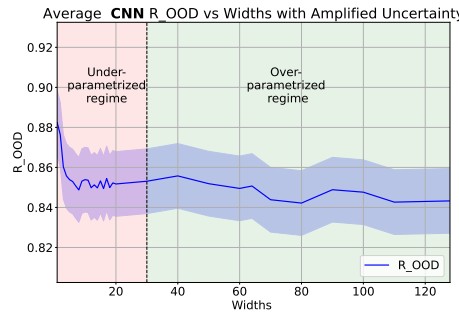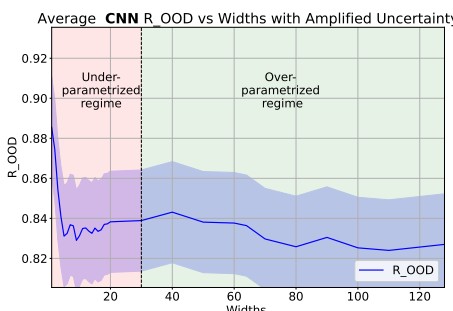

Figure D.11: OOD risk evolution curve w.r.t model's width (x-axis). For a CNN model with CIFAR-10 as ID and iNaturalist as OOD (left) and Textures as OOD (right).

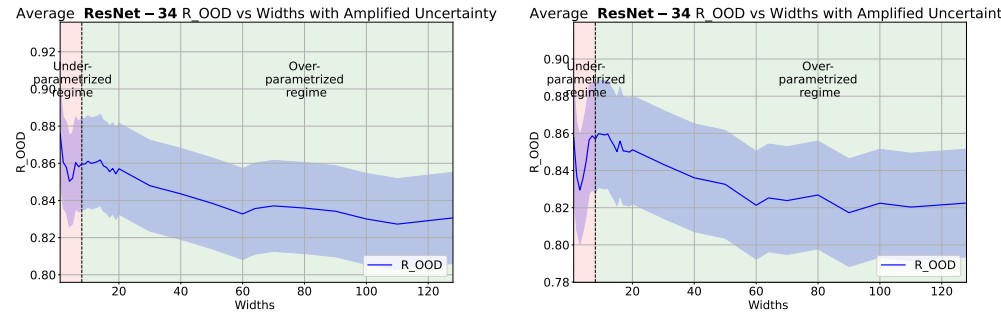

Figure D.12: OOD risk evolution curve w.r.t model's width (x-axis). For a ResNet-34 model with CIFAR-10 as ID and iNaturalist as OOD (left) and ImageNet-O as OOD (right).

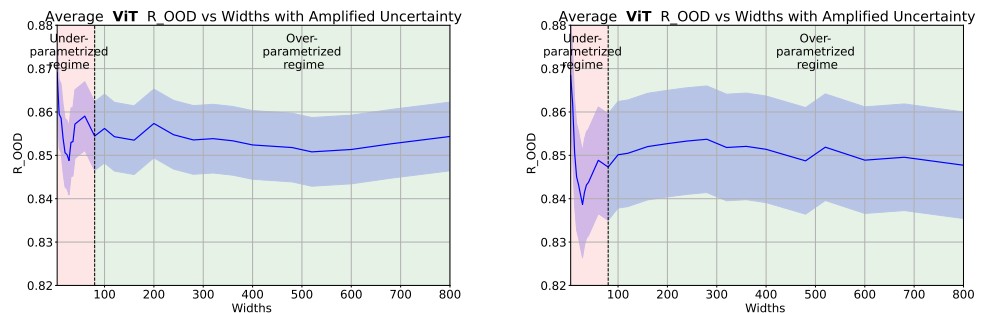

Figure D.13: OOD risk evolution curve w.r.t model's width (x-axis). For a ViT model with Imagenet-o as ID and Places365 as OOD (left) and ImageNet-O as OOD (right).

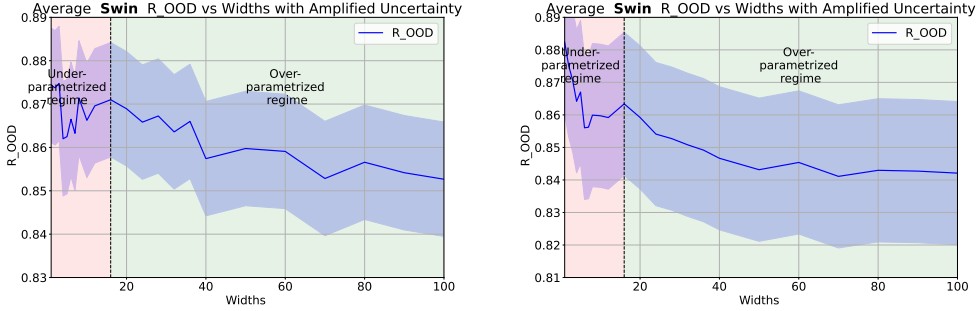

Figure D.14: OOD risk evolution curve w.r.t model's width (x-axis). For a Swin model with ImageNet-O as ID and SUN as OOD (left) and ImageNet-O as OOD (right).

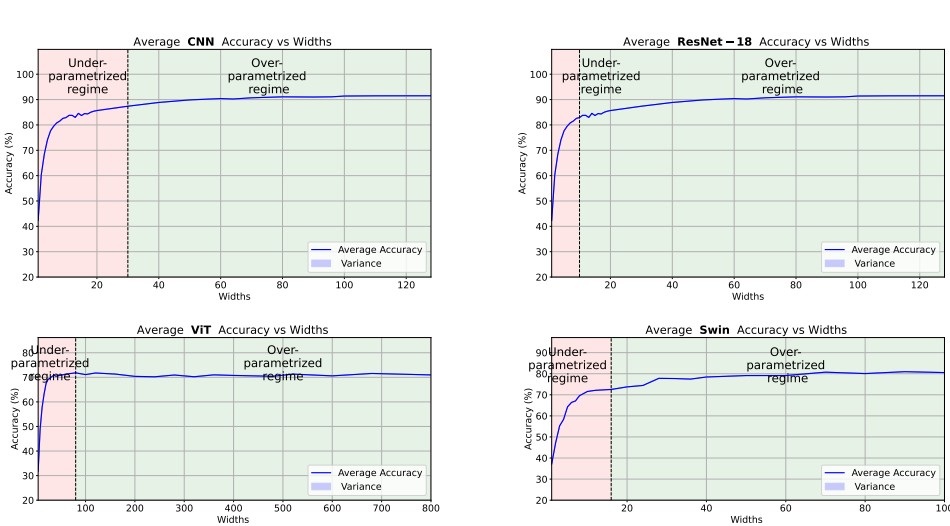

Figure D.15: Generalization evolution curve w.r.t model's width (x-axis), for a ResNet-34 model with CIFAR-10 as ID (left) and CIFAR-100 as ID (right).

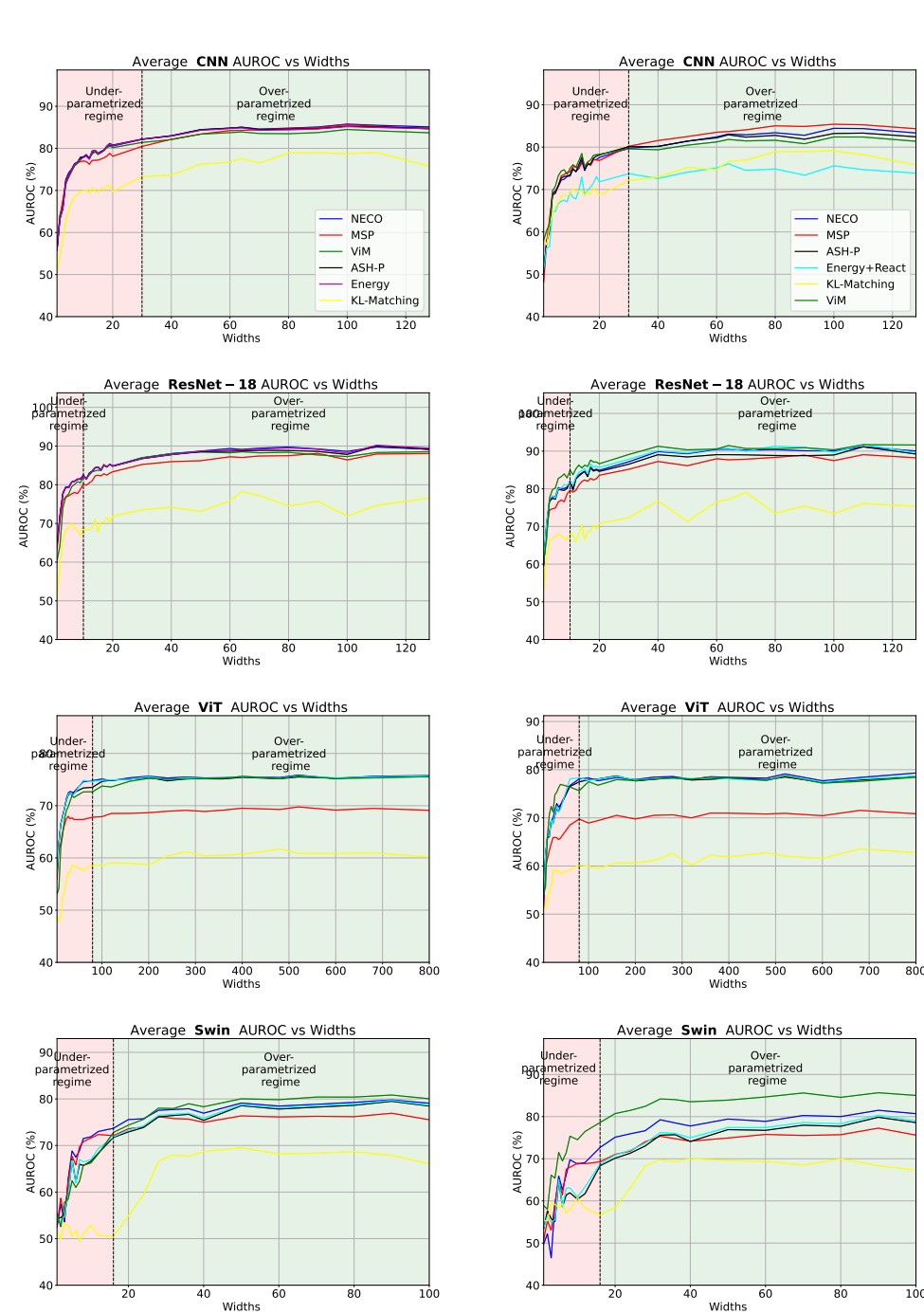

Figure D.16: OOD detection evolution curve w.r.t model's width (x-axis) in terms AUC, on the noise-less training case. with CIFAR-10 as ID and CIFAR-100 as OOD (left) and ImageNet-O as OOD (right). Used models are, from top to bottom, CNN, ResNet-18, Vit, Swin.

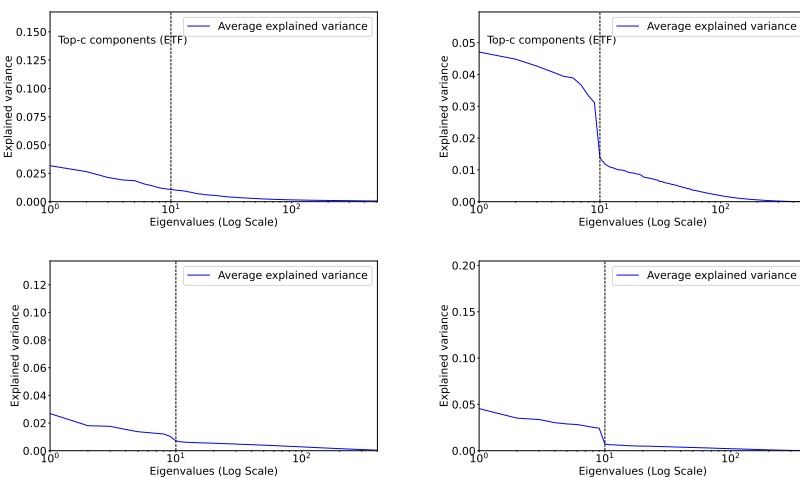

Figure E.1: Eigenvalues explained variance distribution in the overparametrized region for ((from top-left to bottom-right)) CNN (width 64), ResNet-18 (width 64), ViT (width 400) and Swin (width 50) from left to right respectively, all with CIFAR-10 as ID. Black line represents the $10^{th}$ eigenvalue.

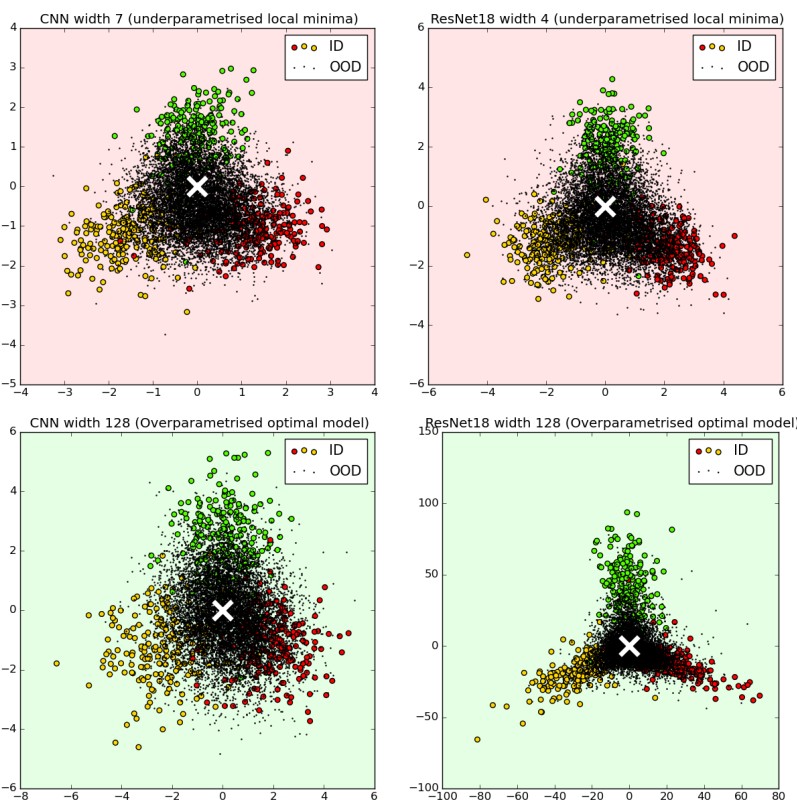

Figure E.2: Visualization of the last-layer activations on the test set for ResNet and CNN in the underparametrized local minima and the overparametrized width 128 model, with cifar10 as ID and cifar100 as OOD dataset. ID point are shown in colors and OOD in black. ResNet underparameterized (left), ResNet overparametrized (middle left), CNN underparametrized (middle right) CNN overparametrized (right).

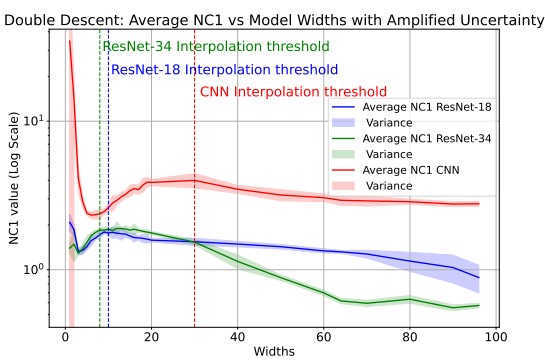

Figure E.3: NC1 metric evolution (Log scale), w.r.t model width increase. ResNet-18 is shown in blue, ResNet-34 in green CNN in red. Dashed lines represent the interpolation threshold for each model with matching color.

