# OpenReview forum: "Double Descent Meets Out-of-Distribution Detection: Theoretical Insights and Empirical Analysis of the role of model complexity"
_ICLR.cc/2025/Conference — Submitted to ICLR 2025_

### Official Review · Reviewer_ZoGQ · 2024-10-28

**Soundness:** 2
**Presentation:** 3
**Contribution:** 2
**Rating:** 6
**Confidence:** 3

**Summary:**

This work focuses on exploring the Double Descent phenomenon in Out-of-Distribution (OOD) Detection. Theoretically, it proposes an expected OOD risk metric and, by leveraging Random Matrix Theory, establishes the relationship between OOD detection and the Double Descent phenomenon within a linear binary classification framework. Experimentally, the study conducts a comprehensive empirical investigation across various datasets and models. In the experiments, model complexity is indicated by different filter sizes, allowing for the observation of the Double Descent phenomenon in OOD detection under varying filter configurations. The conclusion is valid; however, the novelty of the work is not particularly significant.

**Strengths:**

1.	This work focuses on two recent cutting-edge concepts: OOD detection and the Double Descent phenomenon. Both concepts have significant implications for deep learning applications and the understanding of deep learning optimization.
2.	The paper provides clear and rigorous results for OOD detection and the Double Descent phenomenon.
3.	Both the theoretical and experimental results appear to be valid.

**Weaknesses:**

1.	Although the issues considered are cutting-edge concepts, I believe that the motivation presented is weak. The paper does not explore in depth the connections between these two concepts; it primarily provides derivations and experiments based on existing results. Maybe, the authors could discuss potential implications of their findings for model selection in OOD detection tasks, or to analyze how the double descent phenomenon might inform the design of more robust OOD detection methods.
2.	The motivation for expected OOD risk metric is unclear to me; I would appreciate a more detailed explanation. The authors could provide examples of how this metric might be used in practice, or to compare it with existing OOD detection metrics to highlight its unique contributions.
3.	Experimentally, the use of different filter sizes to represent model complexity is appropriate, but it also introduces certain limitations in the experimental results.

**Questions:**

1.	In Section 5.2, the article states, “We report NC metrics, where lower values imply better performance in terms of both generalization and Out-of-Distribution (OOD) detection.” I find this statement insufficiently rigorous. While NC serves as a paradigm to illustrate the optimization process in deep learning, its lower values do not necessarily imply better performance regarding generalization. However, the authors carefully demonstrate the relationship between NC and generalization. Consequently, I remain skeptical of the authors' position and believe that this statement lacks the necessary rigor.
2.	I have concerns about the conclusion of Remark 4.1 of expected OOD risk metric. If the proposed expression (5) yields a lower value, it should not be possible to simultaneously derive conclusions (1) and (2). A more detailed explanation would be appreciated. I would appreciate a more detailed explanation.
3.	Since the Double Descent phenomenon is also based on a linear model, could the depth of the model be related to this phenomenon? Or does the model complexity mentioned in the context of Double Descent pertain solely to the model's width? Additionally, what is the relationship between the Double Descent phenomenon and the depth of the model?
4.	Please refer to “Weaknesses”.

---

> ### Author Response · Authors · 2024-11-18
>
> We thank Reviewer ZoGQ for their insightful comments and suggestions. Below, we address the concerns raised, organized into five key points, presented separately in two parts due to character limit restrictions:
>
> ### **A1.** Motivation for the Paper (W1):
> OOD detection is a rapidly growing field in deep learning, with most algorithms relying heavily on deep neural networks (DNNs). While the double descent phenomenon has been extensively studied for classification and regression tasks, its implications for OOD detection remain largely unexplored.
>
> OOD detection differs fundamentally from classification, particularly in unsupervised settings, where it can be linked to the epistemic uncertainty of DNNs. Given that OOD detection primarily relies on classification-based DNNs, it is already indirectly influenced by the model’s generalization performance. This work aims to establish a more explicit link between OOD detection and the double descent phenomenon, filling a gap in the literature.
>
> Our paper not only provides a theoretical framework to demonstrate this phenomenon but also empirically validates it across multiple datasets and DNN architectures, offering robust evidence of its significance in OOD detection.
>
> ### **A2.** Motivation for the Expected OOD Risk Metric (W2):
>
> To the best of our knowledge, no prior work introduces an "Expected OOD Risk Metric." Traditional OOD approaches typically focus on metrics like AUROC, AUPR, and FPR, optimizing OOD detection through various thresholding. Our proposed OOD risk metric,
> $R_\text{OOD}(\hat f)$ (5) (line 208), is linked to confidence-based OOD detection methods, such as Maximum Softmax Probability (MSP).
>
>
> Concretely, $R_\text{OOD}(\hat{f})$ is minimized when the binary classifier $\hat{f}$ is confident for in-distribution (ID) samples ($\hat{f}(\mathbf{x}) \in \{1, 0\}, \mathbf{x} \in P_{\mathcal{X}, \mathcal{Y}}$) and uncertain for OOD samples ($\hat{f}(\mathbf{x}) = 0.5, \mathbf{x} \in P_{\mathcal{X}, \mathcal{Y}}^\text{OOD}$). Deviations from this behavior, such as uncertainty on ID samples ($\hat{f}(\mathbf{x}) \approx 0.5$) or confidence on OOD samples ($\hat{f}(\mathbf{x}) \neq 0.5$), lead to higher $R_\text{OOD}(\hat{f})$ values.
> This metric naturally extends to the multi-class case (Remark 4.2) using the infinity norm. In this context, $R_\text{OOD}(\hat{f})$ is minimized when logits are maximally confident on ID samples (only one logit is non-zero) and uniformly distributed on OOD samples. This setup describes the behavior of a perfectly calibrated classifier using the MSP method.
>
> Regarding conclusions (1) and (2), these follow from the behavior of $z$, which is a function of $\mathbf{x}$. If $\hat{f}$ approximates $f^*$ well, $z$ behaves as:
> $$
>    \\begin{align}
>         z(x)& \\approx
>         \\begin{cases} \epsilon',  & \\text{if } x \\in P_{\\mathcal{X}, \\mathcal{Y}}^\text{OOD}, \\\\
>            \\pm1+ \\epsilon', & \\text{if } x \\in P_{\\mathcal{X}, \\mathcal{Y}},
>         \\end{cases}
>     \\end{align}
> $$
> where $\epsilon'$ is a small perturbation. Thus, $R_\text{OOD}(\hat{f})$ can simultaneously minimize both terms under the assumption that $\hat{f}$ is a good approximation of $f^*$.
>
> ### **A3.** Experimental Limitations with the Width (W3 + Q3):
>
> We are unsure of the exact limitation referenced by the reviewer. Double descent links model capacity/complexity to generalization performance. By definition, model depth relates to the model complexity, as deeper models are inherently more complex (at equal widths). Therefore, as the model complexity depends on the depth of the neural network, we can assume that the depth of the neural network also plays a role in the double descent phenomenon. To the best of our knowledge, we do not know works that considered varying the depth to study the double descent phenomenon. Our findings support the hypothesis that deeper models may achieve the interpolation threshold at lower widths, providing an avenue for future exploration.

---

> ### Author Response · Authors · 2024-11-18
>
> ### **A4.** Concerns About NC Metrics and Generalization (Q1):
>
>
> The link between the quality of OOD detection and Neural Collapse has been established in [1,2,3,4,5]; however, its connection to generalization quality remains unproven. We will revise the sentence, which was intended only to clarify the metric used and did not draw any conclusions. We acknowledge this oversight and greatly appreciate the reviewer bringing it to our attention. We encourage further feedback to help us address any similar issues.
>
> ### **A5.** Clarification Needed for Remark 4.1 (Q2):
>
> Thank you for highlighting the lack of clarity on that point. A lower value for the expected Out-of-Distribution (OOD) risk does not necessarily imply both better confidence on in-distribution (ID) data and lower confidence on OOD data simultaneously. However, it does indicate either better confidence on ID data or lower confidence on OOD data, or a combination of both.
>
> In particular, the expected OOD risk is minimized when the classifier $\hat{f}(\cdot)$ is confident in its predictions over the in-distribution (ID) data and  uncertain  in its predictions over out-of-distribution (OOD) samples. Indeed, ideally,  $\\hat{f}(\\mathbf{x})$ should be close to $1$ or $0$ for $\mathbf{x} \in P_{\mathcal{X},\mathcal{Y}}$ and  $\hat{f}(\mathbf{x})$ should be close to  $0.5$ for $\mathbf{x} \in P_{\mathcal{X},\mathcal{Y}}^\text{OOD}$. The more the classifier deviates from this ideal setup, the higher the OOD risk. Specifically, if the classifier becomes more uncertain on ID samples (with $\hat{f}(\mathbf{x})$ approaching $0.5$  for $\mathbf{x} \in P_{\mathcal{X},\mathcal{Y}}$) and more confident on OOD samples (with $\hat{f}(\mathbf{x})$ moving further from $0.5$  $\mathbf{x} \in P_{\mathcal{X},\mathcal{Y}}^\text{OOD}$), the OOD risk increases. This behavior is akin to confidence-based OOD detection methods, such as the Maximum softmax probability (MSP) or the MaxLogit metrics, which measure the confidence of the classifier's predictions.
>
>
> ### References
>
> [1] Liu, Litian, and Yao Qin. "Detecting out-of-distribution through the lens of neural collapse." arXiv preprint arXiv:2311.01479 (2023).
>
> [2] Ben Ammar, M., Belkhir, N., Popescu, S., Manzanera, A., & Franchi, G. (2023). NECO: Neural Collapse Based Out-of-distribution Detection. ICLR 2024.
>
> [3] Haas, Jarrod, William Yolland, and Bernhard Rabus. "Linking neural collapse and l2 normalization with improved out-of-distribution detection in deep neural networks." arXiv preprint arXiv:2209.08378 (2022).
>
> [4] Zhang, Jiawei, et al. "EPA: Neural Collapse Inspired Robust Out-of-distribution Detector." ICASSP 2024-2024 IEEE International Conference on Acoustics, Speech and Signal Processing (ICASSP). IEEE, 2024.
>
> [5] Wu, Y., Yu, R., Cheng, X., He, Z., & Huang, X. (2024). Pursuing Feature Separation based on Neural Collapse for Out-of-Distribution Detection. arXiv preprint arXiv:2405.17816.

---

> ### Comment · Reviewer_ZoGQ · 2024-11-22
> **Response to Rebuttal**
>
> Thanks for the authors' rebuttal. Based on their clarifications, I am inclined to raise my score. However, I still have one major question.
>
> Neural Collapse (NC) is a phenomenon observed in model optimization, but does a lower NC value necessarily imply better performance? While some literature may suggest this conclusion, I remain skeptical. When introducing Neural Collapse, the authors are advised to avoid directly associating lower NC values with better performance. Instead, NC could be presented as a metric, which would provide a more neutral and rigorous framing than the current description.
>
> For example, the statement in lines 327–329,"Neural Collapse: we report NC metrics, where lower values imply better performance in terms of OOD detection," might be reconsidered for greater clarity and precision.

---

> > ### Author Response · Authors · 2024-11-22
> > **Thank you for the reply**
> >
> > Thank you for your reply. We appreciate the reviewer's feedback and understand the concern raised. The concept of Neural Collapse (NC), particularly NC1 and NC2, relates to the idea that the penultimate layer embeddings of data points converge towards their respective class centroids, while the centroids of different classes become more distinct and move farther apart.
> >  This principle has been utilized in [1], and subsequent works building on [2,3,4,5] have demonstrated that NC may enhance OOD detection.
> >
> > However, we fully understand the reviewer's caution regarding the interpretation of these findings, and we have taken this feedback into account. To address this, we have revised our statement to ensure it is more cautious and precise.
> >
> > Here is the updated statement: "We report NC metrics, which, as noted in [2,3,4,5], are associated with certain aspects of OOD detection."
> >
> > We hope this updated version aligns with the reviewer's expectations and would welcome any further feedback.
> >
> >
> >
> > ### References
> >
> > [1] Ming, Y., Bai, H., Katz-Samuels, J., & Li, Y. (2024). Hypo: Hyperspherical out-of-distribution generalization. arXiv preprint arXiv:2402.07785.
> >
> > [2] Zhang, J., Chen, Y., Jin, C., Zhu, L., & Gu, Y. (2024, April). EPA: Neural Collapse Inspired Robust Out-of-distribution Detector. In ICASSP 2024-2024 IEEE International Conference on Acoustics, Speech and Signal Processing (ICASSP) (pp. 6515-6519). IEEE.
> >
> > [3] Ammar, M. B., Belkhir, N., Popescu, S., Manzanera, A., & Franchi, G. (2023). NECO: NEural Collapse Based Out-of-distribution detection. arXiv preprint arXiv:2310.06823.
> >
> > [4] Haas, Jarrod, William Yolland, and Bernhard T. Rabus. "Linking Neural Collapse and L2 Normalization with Improved Out-of-Distribution Detection in Deep Neural Networks." Transactions on Machine Learning Research.
> >
> > [5] Zhao, Zhilin, and Longbing Cao. "Dual representation learning for out-of-distribution detection." Transactions on Machine Learning Research (2023).

---

### Official Review · Reviewer_eoHk · 2024-11-03

**Soundness:** 2
**Presentation:** 3
**Contribution:** 2
**Rating:** 5
**Confidence:** 2

**Summary:**

This paper investigates the influence of model complexity on the out-of-distribution (OOD) detection problem by examining its connection to the double descent phenomenon. Specifically, This paper proposes a novel OOD risk metric and establishes bounds for the excess risk of both test samples and OOD samples, assuming Gaussian data. Leveraging these theoretical findings, this paper demonstrates that the model achieves optimal performance when the number of parameters matches the number of samples. Extensive experiments have been conducted across multiple neural network architectures to validate their claims.

**Strengths:**

Overall, this paper presents a robust theoretical framework for analyzing the OOD detection problem and introduces a valid method supported by theoretical derivations and extensive numerical studies.

**Weaknesses:**

The main weakness lies in the strong assumptions on both the data and the model.

1. The model is a one-hidden-layer network with a single neuron.

2. The data is assumed to be Gaussian and isotropic. First, the Gaussian distribution may not reflect the real data application.

3. From lines 830 to 835, the nonlinear function is approximated by a linear function. However, the validity of this approximation is unclear—specifically, the conditions under which this approximation is valid and the rationale behind deriving it. Furthermore, this simplification may reduce the analysis to a purely linear case, potentially limiting the paper's contribution.

===============================

(comments after rebuttal) I changed my score from 6 to 5 (borderline reject) because I believe this paper is not yet ready for publication.

1. The Gaussian input assumption.
2. The simplicity of the model, which involves only one hidden layer with a single neuron.
3. A new concern was raised during discussions with the authors: the limited technical contributions compared to existing works, which also focus on linear models.
4. The current version contains multiple mistakes in the proof.

**Questions:**

(1) I feel that Eqn. (5) is unclear, as $z$ depends on $x$. The current format gives the impression that $z$ is fixed in both terms.

(2) Why Gaussian distribution is necessary for the proof?

(3) Could you provide more details for understanding remark 4.6? Also, how Theorems 1 & 2 are related to the OOD detection method?

---

> ### Author Response · Authors · 2024-11-18
>
> We would like to express our gratitude to reviewer eoHk for their insightful comments and suggestions. Below, we address the concerns raised in the following five points,presented separately in two parts due to character limit restrictions:
>
> ### **A1.** Assumptions on the Model (W1):
>
>  We acknowledge the reviewer's concern regarding the model's simplicity. Theoretically demonstrating the existence of the double descent phenomenon remains a highly active and challenging area of research in machine learning. Most theoretical studies on the double descent phenomenon consider  simple models, such as single hidden layer models, to operate within more mathematically tractable frameworks. In particular, this approach enables researchers to leverage powerful mathematical tools, including Random Matrix Theory, to gain better insights into the double descent phenomenon. In particular, [1,2,3,4,5] considered simplified models like linear regression or random features to obtain mathematical insights into the double descent phenomenon. To obtain similar theoretical insights into the OOD metrics and the double descent phenomenon, we have adopted a similar approach.
>
> ###  **A2.** Assumptions on the Data (W2+Q2):
>
> The assumption of Gaussian-distributed data is common and often used in theoretical works, especially when studying phenomena like double descent and Random Matrix Theory [2,3,4,5]. This assumption allows for more tractable and accurate results, particularly in deriving the Wishart distribution for the Gram matrix of the data. While this assumption might not perfectly match real-world data, it enables us to derive meaningful bounds that depend solely on model complexity. This helps illuminate the double descent phenomenon.
>
> Furthermore, in high-dimensional random projections, data often becomes isotropically distributed, even if the original data is not perfectly Gaussian. The Central Limit Theorem also suggests that sums of independent random variables will approximate a normal distribution as the number of terms increases. Therefore, high-dimensional aggregated data often closely approximates a Gaussian distribution. Empirical studies in deep learning also show that deep neural network embeddings often approximate isotropic Gaussian distributions, particularly in high-dimensional feature spaces.

---

> > ### Comment · Reviewer_eoHk · 2024-11-21
> >
> > Thank you for the rebuttal. Could the authors provide additional clarification and supporting evidence for the statement: "Empirical studies in deep learning also show that deep neural network embeddings often approximate isotropic Gaussian distributions, particularly in high-dimensional feature spaces"?

---

> > > ### Author Response · Authors · 2024-11-21
> > > **Thank you for the reply**
> > >
> > > We appreciate the reviewer’s response and its valuable feedback. The Gaussian hypothesis is indeed well-established in the deep learning literature. Numerous studies [1, 2, 3, 4, 5, 6, 7] have shown that as the width of a finite-depth neural network with random initialization increases, its behavior increasingly resembles that of a Gaussian process. Additionally, works in the Neural Tangent Kernel (NTK) community [8, 9, 10] have demonstrated that gradient descent trajectories can be characterized by a kernel matrix computed using the NTK, effectively linking neural networks to Gaussian processes.
> > >
> > > In the double descent literature [11, 12, 13], it is commonly (not all the works) assumed that data follow an isotropic Gaussian distribution. While [14] extends beyond this assumption, other works such as [15] utilize a linear model with random features to avoid making strong assumptions about the dataset. The use of random features is also prominent in theoretical Kernel Ridge Regression (KRR) studies [16, 17, 18], where the Gaussian assumption is employed to interpret KRR as a linear model with Gaussian random features. In spectral bias research, [19] assumes that the data distribution is uniform on the sphere.
> > >
> > > Our central claim is that assuming an isotropic Gaussian nature of the data, although not exact, provides a foundation for deriving equations and establishing a mathematical framework for double descent. We acknowledge that this formulation relies on strong hypotheses, which are open to criticism. However, we chose this model as it enabled us to extend the analysis to out-of-distribution (OOD) scenarios. In future work, we plan to explore alternative approaches, such as leveraging random projection methods, to reduce reliance on these assumptions. We plan to add a discussion about this point.
> > >
> > > We sincerely thank the reviewer once again for your answer and would greatly appreciate further clarification if we have misunderstood any aspect of the points raised.
> > > Please feel free to reach out if you have additional questions or require further explanation regarding our response.
> > >
> > >  ### References
> > > [1] Radford M Neal. Bayesian learning for neural networks, volume 118. Springer Science & Business Media, 2012.
> > >
> > > [2] Jaehoon Lee, Yasaman Bahri, Roman Novak, Samuel S Schoenholz, Jeffrey Pennington, and Jascha Sohl-Dickstein. Deep neural networks as gaussian processes. In International Conference on Learning Representations.
> > >
> > > [3] Alexander G de G Matthews, Jiri Hron, Mark Rowland, Richard E Turner, and Zoubin Ghahramani. Gaussian process behaviour in wide deep neural networks. In International Conference on Learning Representations.
> > >
> > > [4] Gao, T., Huo, X., Liu, H., & Gao, H. (2023). Wide neural networks as gaussian processes: Lessons from deep equilibrium models. Advances in Neural Information Processing Systems, 36, 54918-54951.
> > >
> > > [5] Roman Novak, Lechao Xiao, Yasaman Bahri, Jaehoon Lee, Greg Yang, Jiri Hron, Daniel A Abolafia, Jeffrey Pennington, and Jascha Sohl-dickstein. Bayesian deep convolutional networks with many channels are gaussian processes. In International Conference on Learning Representations.
> > >
> > > [6] Greg Yang. Wide feedforward or recurrent neural networks of any architecture are gaussian processes. Advances in Neural Information Processing Systems, 32, 2019.
> > >
> > > [7]  Stefano Peluchetti and Stefano Favaro. Infinitely deep neural networks as diffusion processes. In International Conference on Artificial Intelligence and Statistics, pages 1126–1136. PMLR, 2020.
> > >
> > > [8] Arthur Jacot, Franck Gabriel, and Clément Hongler. Neural tangent kernel: Convergence and
> > > generalization in neural networks. Advances in neural information processing systems, 31,
> > > 2018.
> > >
> > > [9] Arora, S., Du, S. S., Hu, W., Li, Z., Salakhutdinov, R. R., & Wang, R. (2019). On exact computation with an infinitely wide neural net. Advances in neural information processing systems, 32.
> > >
> > > [10] Lee, J., Xiao, L., Schoenholz, S., Bahri, Y., Novak, R., Sohl-Dickstein, J., & Pennington, J. (2019). Wide neural networks of any depth evolve as linear models under gradient descent. Advances in neural information processing systems, 32.
> > >
> > > [11] Nakkiran, P., Venkat, P., Kakade, S. M., & Ma, T. Optimal Regularization can Mitigate Double Descent. In International Conference on Learning Representations.
> > >
> > > [12] Derezinski, Michal, Feynman T. Liang, and Michael W. Mahoney. "Exact expressions for double descent and implicit regularization via surrogate random design." Advances in neural information processing systems 33 (2020): 5152-5164.
> > >
> > > [13] Belkin, Mikhail, Daniel Hsu, and Ji Xu. "Two models of double descent for weak features." SIAM Journal on Mathematics of Data Science 2.4 (2020): 1167-1180.
> > >
> > > [14] Bach, F. (2024). High-dimensional analysis of double descent for linear regression with random projections. SIAM Journal on Mathematics of Data Science, 6(1), 26-50.

---

> > > > ### Author Response · Authors · 2024-11-21
> > > > **the other references**
> > > >
> > > > ### References
> > > >
> > > > [15] Liao, Zhenyu, Romain Couillet, and Michael W. Mahoney. "A random matrix analysis of random fourier features: beyond the gaussian kernel, a precise phase transition, and the corresponding double descent." Advances in Neural Information Processing Systems 33 (2020): 13939-13950.
> > > >
> > > > [16] Jacot, A., Simsek, B., Spadaro, F., Hongler, C., & Gabriel, F. (2020). Kernel alignment risk estimator: Risk prediction from training data. Advances in neural information processing systems, 33, 15568-15578.
> > > >
> > > > [17] Canatar, A., Bordelon, B., & Pehlevan, C. (2021). Spectral bias and task-model alignment explain generalization in kernel regression and infinitely wide neural networks. Nature communications, 12(1), 2914.
> > > >
> > > > [18] Simon, J. B., Dickens, M., Karkada, D., & Deweese, M. (2023). The eigenlearning framework: A conservation law perspective on kernel ridge regression and wide neural networks. Transactions on Machine Learning Research.
> > > >
> > > > [19] Bietti, A., & Mairal, J. (2019). On the inductive bias of neural tangent kernels. Advances in Neural Information Processing Systems, 32.

---

> ### Author Response · Authors · 2024-11-18
>
> ### **A3.** Validity of Nonlinear Approximation (W3):
>
> We acknowledge the reviewer's observation about the lack of an important assumption in the paper regarding the nonlinear approximation. Specifically, we forgot to state that the optimal weight $\\hat{\\mathbf{w}}$ should not differ significantly from the optimized weight $\\mathbf{w}^*$. This assumption, while challenging to justify rigorously, is supported empirically in our study, where the difference between the two is found to be small. We will make this assumption explicit in the revised manuscript.
>
>
> ### **A4.** Clarification of Equation 5 (Q1):
>
>  We appreciate the reviewer pointing out the ambiguity in Equation (5). The dependency of $z$ on $x$ was not sufficiently clear, and we have revised the manuscript to clarify this. Specifically, $z$ is dependent on whether $x$ is from the in-distribution (ID) or out-of-distribution (OOD) set. We now explicitly define $z$ as follows:
>  $$
>    \\begin{align}
>         z(x)& \\approx
>         \\begin{cases} \epsilon',  & \\text{if } x \\in P_{\\mathcal{X}, \\mathcal{Y}}^\text{OOD}, \\\\
>            \\pm1+ \\epsilon', & \\text{if } x \\in P_{\\mathcal{X}, \\mathcal{Y}},
>         \\end{cases}
>     \\end{align}
> $$
>
>  This definition ensures that the prediction function $f^*$ approaches 0.5 for OOD data and 1 for in-distribution data, which helps minimize the OOD risk when plugged into the risk function $R_\\text{OOD}(\\hat f)$.
>
>
>  ### **A5.** Further Details on Remark 4.6 and Relation to OOD Detection (Q3):
>
>
> Our objective in both Theorem 1 and Theorem 2 is to show that the expected risk and the expected OOD risk exhibit a double descent descent phenomenon that we associate with an infinite peak when the number of samples $n$ is close  to the number of features $p$ as noted in Remark 4.5 and Remark 4.6.
>
> Specifically, we aim to derive lower and upper bounds that depend solely on the model complexity (which is proportional to the number of samples n and the number of features p) and the level of noise $\\sigma$, through the function  $c(n, p, \sigma)$. As $c(n, p, \\sigma)=\\infty$ when $n-1 \\leq p \\leq n+1$, we can deduce from the squeeze theorem that $E_X[R_\\text{OOD}(\\hat f)]= \\infty$ when $n-1 \\leq p \\leq n+1$  in this regime.
>
> We associate this infinite peak in the expected OOD risk with the double descent phenomenon. Since we have introduced the expected OOD risk as an OOD detection metric, to the best of our knowledge, this work is the first to mathematically describe a double descent phenomenon for an OOD metric.
>
>  ### References
>
> [1] Francis Bach (2023).High-dimensional analysis of double descent for linear regression with random projections. SIAM Journal on Mathematics of Data Science
>
> [2] Belkin, M., Hsu, D., Ma, S., & Mandal, S. (2019). Reconciling modern machine-learning practice and the classical bias–variance trade-off. Proceedings of the National Academy of Sciences, 116(32), 15849-15854.
>
> [3] Louart, C., Liao, Z., & Couillet, R. (2018). A random matrix approach to neural networks. The Annals of Applied Probability, 28(2), 1190-1248.
>
> [4] Liao, Z., Couillet, R., & Mahoney, M. W. (2020). A random matrix analysis of random fourier features: beyond the gaussian kernel, a precise phase transition, and the corresponding double descent. Advances in Neural Information Processing Systems, 33, 13939-13950.
>
> [5] Belkin, M., Hsu, D., & Xu, J. (2020). Two models of double descent for weak features. SIAM Journal on Mathematics of Data Science, 2(4), 1167-1180.

---

> ### Comment · Reviewer_eoHk · 2024-11-21
> **Require a Further Clarification on Gaussian Process**
>
> I believe the Neural Tangent Kernel (NTK) does not assume that the data follows a Gaussian distribution. The connection between NTK training and Gaussian processes arises from the **initialization of neurons using a Gaussian distribution**, along with certain additional assumptions.
>
> Also, NTK has been found to lack practical applicability to neural network training due to its strong assumptions. Specifically, it links neural network training to Gaussian processes (random feature learning), whereas practical training involves neurons learning features from input data. For example, see [1]. Could the author share your thoughts on this point?
>
> [1] Feature Purification: How Adversarial Training Performs Robust Deep Learning

---

> > ### Author Response · Authors · 2024-11-22
> > **Thank you for the reply**
> >
> > First, we would like to sincerely thank the reviewer for initiating a discussion with us and for sharing the referenced paper, which we have found very interesting. The notions discussed in the paper are indeed related to ours, as it focuses on adversarial attacks and introduces the concept of feature purification to theoretically understand their causes. While we deeply appreciate the insights from this paper, we believe its primary focus on adversarial attacks is not entirely aligned with our work.
> >
> > Adversarial attack has long been recognized as distinct from OOD detection. For instance, [1] discusses their differences while acknowledging some overlaps, and several surveys [2,3] focus on OOD detection without really addressing adversarial attacks. If the reviewer agrees, we propose adding a sentence to our paper to clarify its scope and explicitly state that we do not address adversarial attacks such as the reference mentioned.
> >
> > Regarding the Neural Tangent Kernel (NTK), we apologize if our phrasing caused confusion. We did not imply that NTK inherently requires data to follow a Gaussian distribution. Instead, we aimed to highlight that DNNs are connected to Gaussian processes. Our discussion on NTK is orthogonal to the core focus of our paper and was included to illustrate the presence of other gaussian hypotheses in prior works.
> >
> > Concerning the assumptions on the data, we acknowledge that such hypotheses, although strong, are commonly used to facilitate mathematical derivations. For example, some works assume data is distributed on a unit circle [3,4,5], from sub-gaussian/gaussian distributions [6, 7, 8, 9].  Most of the time, idealized assumptions are necessary to establish formal results. Nonetheless, those theoretical results have provided insight into the effects of overparameterization and regularization with practical implications, e.g., on the choice of hyperparameters. We expect that further theoretical studies will be conducted to establish a formal link between double descent in OOD and neural networks. Finally, we would like to kindly remind you that our theoretical study is mainly focused on providing theoretical “insights” on simple models, rather than proving a formal double descent on more realistic models. We hope that this focus is reflected in the paper's title.
> >
> > Once again, thank you for bringing this paper to our attention. While it is not directly related to our specific objectives, we greatly appreciate its relevance to the broader field. We hope this response clarifies our position, and we look forward to continuing this constructive discussion.
> >
> >
> > ### References
> >
> > [1] Karunanayake, N., Gunawardena, R., Seneviratne, S., & Chawla, S. (2024). Out-of-Distribution Data: An Acquaintance of Adversarial Examples--A Survey. arXiv preprint arXiv:2404.05219.
> >
> > [2] Miyai, A., Yang, J., Zhang, J., Ming, Y., Lin, Y., Yu, Q., ... & Aizawa, K. (2024). Generalized out-of-distribution detection and beyond in vision language model era: A survey. arXiv preprint arXiv:2407.21794.
> >
> > [3] Liu, J., Shen, Z., He, Y., Zhang, X., Xu, R., Yu, H., & Cui, P. (2021). Towards out-of-distribution generalization: A survey. arXiv preprint arXiv:2108.13624.
> >
> > [4] Bietti, A., & Mairal, J. (2019). On the inductive bias of neural tangent kernels. Advances in Neural Information Processing Systems, 32.
> >
> > [5] Mei, S., & Montanari, A. (2022). The generalization error of random features regression: Precise asymptotics and the double descent curve. Communications on Pure and Applied Mathematics, 75(4), 667-766.
> >
> > [6] Bach, F. (2017). Breaking the curse of dimensionality with convex neural networks. Journal of Machine Learning Research, 18(19), 1-53.
> >
> > [7] Bach, F. (2024). High-dimensional analysis of double descent for linear regression with random projections. SIAM Journal on Mathematics of Data Science, 6(1), 26-50.
> >
> > [8] d’Ascoli, S., Refinetti, M., Biroli, G., & Krzakala, F. (2020, November). Double trouble in double descent: Bias and variance (s) in the lazy regime. In International Conference on Machine Learning (pp. 2280-2290). PMLR.
> >
> > [9] Belkin, M., Hsu, D., & Xu, J. (2020). Two models of double descent for weak features. SIAM Journal on Mathematics of Data Science, 2(4), 1167-1180.

---

> ### Comment · Reviewer_eoHk · 2024-11-22
>
> Let me clarify my second point. I am not asking you to compare your paper with the specific content in [1]. My point is that [1] argues that using Gaussian processes to analyze DNNs does not align with practical settings, whereas the framework proposed in [1] claims to address this issue. Could you share your thoughts on this?

---

> ### Author Response · Authors · 2024-11-23
> **Thank you for the reply**
>
> Thank you for your clarification. We recognize that there may have been a misunderstanding arising from our initial response, in which we wrote: "Empirical studies in deep learning also show that deep neural network embeddings often approximate isotropic Gaussian distributions, particularly in high-dimensional feature spaces."
> We would like to correct this statement: we did not intend to imply that DNNs follow an isotropic Gaussian distribution or are Gaussian processes. This was a misstatement and we formally retract that point in our reply.
> To clarify, our paper does not rely on the assumption that DNNs follow a Gaussian distribution or a Gaussian Process. Instead, we assume that the input data follows an isotropic Gaussian distribution. As highlighted in our previous replies, such assumptions are common in the literature [1,2,3,4] and are necessary for deriving formal proofs/providing insights.
> We acknowledge the argument developed in [5] that using Gaussian processes to analyze DNNs may not align with practical settings, and we do not disagree with this point. However, we emphasize that our work does not depend on such assumptions regarding the DNNs themselves.
> We hope this response resolves any remaining confusion and are happy to continue this discussion further.
> ### References
> [1] Bach, F. (2017). Breaking the curse of dimensionality with convex neural networks. Journal of Machine Learning Research, 18(19), 1-53.
>
> [2] Bach, F. (2024). High-dimensional analysis of double descent for linear regression with random projections. SIAM Journal on Mathematics of Data Science, 6(1), 26-50.
>
> [3] d’Ascoli, S., Refinetti, M., Biroli, G., & Krzakala, F. (2020, November). Double trouble in double descent: Bias and variance (s) in the lazy regime. In International Conference on Machine Learning (pp. 2280-2290). PMLR.
>
> [4] Belkin, M., Hsu, D., & Xu, J. (2020). Two models of double descent for weak features. SIAM Journal on Mathematics of Data Science, 2(4), 1167-1180.
>
> [5] Allen-Zhu, Zeyuan, and Yuanzhi Li. "Feature purification: How adversarial training performs robust deep learning." 2021 IEEE 62nd Annual Symposium on Foundations of Computer Science (FOCS). IEEE, 2022.

---

> > ### Comment · Reviewer_eoHk · 2024-11-25
> > **Require Additional Clarfication**
> >
> > If my understanding is correct, the analysis relies on (i) the Gaussian distribution and (ii) the approximation of a nonlinear function by a linear function, which cannot be rigorously proven. Could the author elaborate on the technical contributions compared to existing works that consider a linear model with a Gaussian distribution?

---

> > > ### Author Response · Authors · 2024-11-25
> > > **Thank you  for your comment**
> > >
> > > The theoretical analysis in our work relies only on the assumption (i) that the in-distribution data follows a Gaussian distribution. This assumption is widely used in the literature to derive formal results and gain insights into various phenomena.
> > >
> > >
> > > To clarify, our proofs do not rely on assumption (ii) regarding the closeness of $\hat w$ to $w^*$. We have removed this assumption and instead used the mean-value theorem to approximate the nonlinear binary classifier $\hat f(\cdot)$ with a linear function. No assumptions on $\hat w$ and $w^*$ are thus necessary.  Details of this linear approximation can be found in the Appendix, specifically at line 1030.
> > > As highlighted in our previous replies, assuming a Gaussian distribution for the data is common in the literature. For instance, [1] uses a similar assumption to demonstrate a double descent phenomenon in the generalization error of simple models in regression. Similarly, [2] studied a linear model with random features and Gaussian data to obtain precise asymptotic expressions for the bias-variance decomposition of the test error, showing a phase transition at the interpolation threshold and a double descent phenomenon. Other works have assumed linear models with random features and sub-Gaussian data in regression to exhibit a similar double descent phenomenon in the test error [3]. Other works like [4] have assumed linear models with random features and data distribution on a unit circle to study the double descent phenomenon in the regression case. This assumption coupled with the NTK was also used to study the spectral bias in neural networks like in [5]. These assumptions, although not necessarily satisfied in practice, are crucial for obtaining formal results and understanding the underlying mechanisms.
> > > To the best of our knowledge, **the double descent phenomenon has not been previously observed in the context of out-of-distribution (OOD) detection.** Our main contributions can be summarized as:
> > >
> > > 1. We propose an expected OOD risk metric, $R_\text{OOD}(\hat f)$ (equation 5, line 208), to evaluate classifiers' confidence on both training and OOD samples. This metric is linked to confidence-based OOD detection methods, such as Maximum Softmax Probability (MSP).
> > > 2. Using Random Matrix Theory, we derive bounds for both the expected risk and OOD risk of binary least-squares classifiers applied to Gaussian data with respect to the model complexity. We show that both risks exhibit an infinite peak for this simple model when the number of parameters is equal to the number of samples, which we associate with the double descent phenomenon.
> > > 3. We empirically observe a double descent phenomenon curve in various OOD detection methods and across multiple neural architectures, including transformer-based (ViT, Swin) and convolutional-based models (ResNet, CNN). 4. We also observe that OOD detection in the overparametrized regime is not guaranteed to be better than in the underparametrized regime. Using the Neural Collapse framework Papyan et al. (2020), we propose to better explain this architecture-dependant improvement on OOD detection with overparametrization.
> > >
> > > **Above all, our experimental study aims to extend the theoretical insights found for Gaussian covariate models to true Deep Neural Network architectures and datasets,** providing a more comprehensive understanding of the influence of model complexity on OOD detection. Note that our theoretical study extends [1] for the binary classification task and the OOD detection through the study of the expected OOD risk. Finally, our objective in the theoretical study is to provide insights into how the model complexity behaves on the OOD detection and that the observed phenomena in the experimental study are not artifacts of some specific experimental protocols.
> > > We trust these clarifications address your concerns and  clarify the objectives of the paper. Please let us know if further elaboration is needed.
> > >
> > > ### References
> > >
> > > [1] Belkin, M., Hsu, D., & Xu, J. (2020). Two models of double descent for weak features. SIAM Journal on Mathematics of Data Science, 2(4), 1167-1180.
> > >
> > > [2] d’Ascoli, S., Refinetti, M., Biroli, G., & Krzakala, F. (2020, November). Double trouble in double descent: Bias and variance (s) in the lazy regime. In International Conference on Machine Learning (pp. 2280-2290). PMLR.
> > >
> > > [3] Bach, F. (2024). High-dimensional analysis of double descent for linear regression with random projections. SIAM Journal on Mathematics of Data Science, 6(1), 26-50.
> > >
> > > [4] Mei, S., & Montanari, A. (2022). The generalization error of random features regression: Precise asymptotics and the double descent curve. Communications on Pure and Applied Mathematics, 75(4), 667-766.
> > >
> > > [5] Bietti, A., & Mairal, J. (2019). On the inductive bias of neural tangent kernels. Advances in Neural Information Processing Systems, 32.

---

> > > > ### Comment · Reviewer_eoHk · 2024-11-26
> > > >
> > > > Thank you for the detailed rebuttal. At first glance, the proof of Lemma 4.1 does not appear to be rigorously correct. A monotonically non-decreasing function means that its gradient is **greater than or equal to zero**. I will look closely into other parts when I am available later this week.  Also, a quick question: what are the challenges in analyzing the case of multiple neurons?
> > > >
> > > > In addition, without explicitly characterizing the magnitude of $\lambda_{\min}$, how could we translate the bound from population risk to empirical risk? For example, $\lambda_{\min}$ may depend on $d$ or some other parameter, leading to a considerably large sample complexity (suppose we will use some concentration inequality).

---

> ### Author Response · Authors · 2024-11-26
>
> Thank you for your response and for appreciating our answers.
>
> We appreciate you pointing out the typo. Indeed, we intended to state "**strictly** monotonically increasing" rather than "monotonically increasing" in the assumption. The case where the activation function $\Phi(\cdot)$ is constant is not relevant to our problem. To the best of our knowledge, there is no constant activation function used in classification. We have updated the paper to correct this error.
>
> We are unclear about your specific question regarding the difficulty in analyzing the case of multiple neurons. Could you please clarify whether you are referring to multiple classes with a single fully connected layer, or multiple fully connected layers? This clarification will help us address your concern more accurately.
>
> Regarding your question about the empirical risk, we would like to clarify that our focus is solely on the population risk. Classically, in Machine Learning, the true objective is to minimize the population risk, while the empirical risk serves as an estimate using a finite number of samples [1]. If you are interested in connecting the empirical risk to the population risk, which is not the primary focus of this paper, concentration inequalities can be employed. In the case you also want to increase d with n the situation becomes more complex as highlighted for example in [2,3,4] or with the Marchenko-Pastur distribution. Nonetheless, it is still possible to obtain concentration inequalities and establish connections between the empirical and population risks [4,5].
>
> ### References
>
> [1] Bach, Francis. Learning theory from first principles. MIT press, 2024.
>
> [2] Couillet, Romain, and Zhenyu Liao. Random matrix methods for machine learning. Cambridge University Press, 2022.
>
> [3] Liao, Zhenyu, Romain Couillet, and Michael W. Mahoney. "A random matrix analysis of random fourier features: beyond the gaussian kernel, a precise phase transition, and the corresponding double descent." Advances in Neural Information Processing Systems 33 (2020): 13939-13950.
>
> [4] Louart, C., Liao, Z., & Couillet, R. (2018). A random matrix approach to neural networks. The Annals of Applied Probability, 28(2), 1190-1248.
>
> [5] Brellmann, David, et al. "On Double Descent in Reinforcement Learning with LSTD and Random Features." The Twelfth International Conference on Learning Representations.

---

> > ### Comment · Reviewer_eoHk · 2024-11-26
> >
> > 1. I believe the ReLU function does not satisfy the condition "strictly monotonically increasing".
> >
> > 2. I am referring to a two-layer neural network with a fixed output layer, like $f_\theta(x)=\sum_{i=1}^m v_i\phi(w_i^\top x)$, which is the typical setup in analyzing the neural network.
> >
> > 3. I have this question based on the following considerations. For me, it is expected that the results for the empirical risk function would be terrible if, for example,
> > $\lambda_\min = c^{-d}$
> > causes the sampling complexity or other conclusions to scale in the order of
> > $c^{d}$. However, your answers help me better understand that the focus of your paper is on asymptotic analysis.

---

> > > ### Author Response · Authors · 2024-11-27
> > >
> > > Dear Reviewer,
> > >
> > > 1. We appreciate your point regarding the ReLU function and its lack of strict monotonicity. We didn’t consider of ReLU because it’s uncommon to use it for mapping to probabilities in classification tasks. Our focus was primarily on activation functions, denoted as $\Phi(\cdot)$, that are typically used to map the outputs of a linear regression model to the probability space of labels as depicted in l.179-l.184.  The ReLU function is thus not relevant for this purpose because it does not output probabilities and is typically not considered in classification problems on the last linear layer. Instead, activation functions like the sigmoid functions are commonly used for classification tasks, as they map inputs to outputs between 0 and 1, making them suitable for binary classification. Note that our assumption holds on such activation functions (e.g the sigmoid function).
> > >
> > > 2.  We want to thank you for your comment. But, we choose not to consider a two-layer neural network  to avoid complicating the problem. Instead, we opted for a least-squares solution to simplify the analysis and avoid the difficulties associated with gradient descent. This approach is also consistent with the double descent literature, where least-squares solutions are often used to derive formal results [1,2,3,4,5,6]. If we were to consider a model similar to a two-layer neural network using least-squares solutions, we would need to leverage the lazy training regime as in [1,2,3,4]. This involves approximating the features returned by the hidden layer with random features. Deriving results for empirical risks would require assumptions about the testing dataset being close to the training dataset, which is less realistic for out-of-distribution (OOD) cases. To study the population risk, we would need to find the expected value of the random matrix $[\\sigma(WX)^T\\sigma(WX)]^{-1}$, where $\\sigma(\\cdot)$ is the activation function of the hidden layer and W are the weights of the hidden layer. This expected value should depend solely on the number of features and the number of samples. However, due to the non-linearity introduced by the activation function, calculating such expected values is highly challenging. For example, finding expected versions with respect to W for similar matrices can be seen in [3, Table 1], but extending this to include expectations with respect to the distribution on $X$ is extremely difficult and beyond our current capabilities.
> > >
> > >
> > > ### References
> > >
> > > [1] Couillet, Romain, and Zhenyu Liao. Random matrix methods for machine learning. Cambridge University Press, 2022.
> > >
> > > [2] Liao, Zhenyu, Romain Couillet, and Michael W. Mahoney. "A random matrix analysis of random fourier features: beyond the gaussian kernel, a precise phase transition, and the corresponding double descent." Advances in Neural Information Processing Systems 33 (2020): 13939-13950.
> > >
> > > [3] Louart, C., Liao, Z., & Couillet, R. (2018). A random matrix approach to neural networks. The Annals of Applied Probability, 28(2), 1190-1248.
> > >
> > > [4] Brellmann, David, et al. "On Double Descent in Reinforcement Learning with LSTD and Random Features." The Twelfth International Conference on Learning Representations.
> > >
> > > [5] Belkin, M., Hsu, D., & Xu, J. (2020). Two models of double descent for weak features. SIAM Journal on Mathematics of Data Science, 2(4), 1167-1180.
> > >
> > > [6] Bach, F. (2024). High-dimensional analysis of double descent for linear regression with random projections. SIAM Journal on Mathematics of Data Science, 6(1), 26-50.

---

> ### Comment · Reviewer_eoHk · 2024-11-27
> **Editted comments**
>
> Thanks for the authors' rebuttal. I will take a closer look into the corresponding proof later this week with your clarification of point two.
>
> A quick comment on point 1 is that, since the data follows a Gaussian distribution (unbounded), for an activation function like the sigmoid, such an epsilon (>0) does not exist.
>
> Edit: I believe this issue could be resolved using advanced tools. The key point is likely that the results are based on a stronger assumption but yield much weaker outcomes compared to the original conclusion. I need some time to read the paper again.
> In the meantime, I suggest that the author review the revised proof again, as it appears rushed and requires more time for refinement.

---

> > ### Author Response · Authors · 2024-11-28
> >
> > Thank you for pointing out the issue with the epsilon part in the proof of Lemma 4.1 involving the sigmoid function.To address this, we have refined the proof by replacing the epsilon part with the following:
> >
> >
> > *Since the derivative* $\\Phi'(\\cdot)$ *is strictly positive, for any* $a \in \\mathbb{R}^d \backslash \\{0\\}$, *we have*
> >
> > $a^T \\Sigma a = a^T E_{(x, \\cdot) \\sim P_{\\mathcal{X}, \\mathcal{Y}}} \\bigl[\\Phi'\bigl(c(x, \\hat w, w^*)\\bigr)^2 x x^T \\bigr] a = \\int_{\\mathbb{R}^d} a^T \phi'\bigl(c(x, \hat w, w^*)\bigr)^2 x x^T a \mu(x)  dx = \int_{\\mathbb{R}^d}  \phi'\bigl(c(x, \hat w, w^*)\bigr)^2 \lVert x^T a \rVert_2^2 \mu(x)   dx > 0$,
> >
> > *where* $\\mu(\\cdot)$ *is the probability density function of* $P_{\\mathcal{X}, \\mathcal{Y}}.$
> >
> > Lemma 4.1 is thus true and holds for the sigmoid function.
> > Regarding your remarks on our claims, we would like to clarify that previously we modified and relaxed our assumptions based on the feedback from all reviewers. In particular, we revised Assumption 4.1 to enhance clarity and readability. The former version of Assumption 4.1, which concerned the nonsingular property of the matrix Sigma, has been restricted to strictly monotonically increasing activation functions. This revision is motivated by Lemma 4.1, which shows the nonsingular property of the matrix Sigma.
> >
> > We agree with you that the proof of Lemma 4.1 required a refinement, which we have now implemented (and written above).  Note that we will modify the paper after the rebuttal period. We thank the reviewer for highlighting the issue with the epsilon part for the sigmoid function. Although the assumption was modified, there are no consequences on the results obtained with Theorems 1 and 2. Please note that this assumption is not used in the proofs of these theorems but is referenced in Remarks 4.5 and 4.6. In addition to the above changes, we also modified the proofs of Theorems 1 and 2 by using the mean-value theorem instead of the linear Taylor approximation. This change eliminates the need for assumptions about the closeness of $\hat w$ with $w^*$.
> >
> > In summary, we have addressed the concerns raised by refining the proof of Lemma 4.1 and modifying Assumption 4.1. We have also improved the proofs of Theorems 1 and 2 by employing the mean-value theorem. We believe these changes have enhanced the clarity and robustness of our arguments without affecting the main results.

---

### Official Review · Reviewer_UnVj · 2024-11-04

**Soundness:** 3
**Presentation:** 3
**Contribution:** 2
**Rating:** 6
**Confidence:** 4

**Summary:**

This paper examines the impact of model complexity on (supervised) out-of-distribution (OOD) detection, showing that, similar to in-distribution (ID) generalization, OOD detection exhibits a double descent behavior as models become overparameterized.

Theoretically, the authors show that, in a setup similar to Belkin et al. (2020), both ID and OOD detection risks follow a double descent pattern in linear least-squares regression. Empirically, they explore various datasets, architectures, and OOD detection techniques, showing that logit-based metrics produce a smoother double descent curve compared to feature-based ones. They also relate the improvement seen with overparameterization to improvements in neural collapse (NC) within the feature space.

**Strengths:**

The paper is written clearly. The authors discuss connections to previous work well and provide extensive experiments across various metrics, architectures, and datasets to support their claims.

**Weaknesses:**

(see below)

**Questions:**

1.  Can you comment on the significance of your theoretical results? It seems that theorem 2 is a simple extension of theorem 1 applied to both ID and OOD distributions, and theorem 1 closely resembles theorem 1 in Belkin et al. (2020), with the addition of the activation function/non-linearity $\phi(\cdot)$ (as also acknowledged in the paper). But do you use $\phi(\cdot)$ to link OOD detection and overparameterization for your overall conclusion? In other words, without $\phi(\cdot)$, what is the technical challenge for extending theorem 1 in Belkin et al. (2020) to the OOD risk?

2.  Your theorem doesn’t specify assumptions on the OOD distribution. Would your theory provide any insights on how the expected behavior of the OOD risk might change with varying degrees of distributional shift between ID and OOD distributions?

3.  The trace operation in the NC1 metric is applied to $d\times d$ matrices ($d$ being the dimension of last-layer features). Thus, I suspect that the value of NC1 would be sensitive to changes in the value of $d$. Is the last-layer dimension in your experiments in both the under- and over-parameterized settings similar?  Or do you vary them as you increase the widths of the intermediate layers of the network? In the latter case, I am wondering if comparing the NC1 values when $d$ is different is fair.

4.  The paper makes heuristic connections between how much over-parameterization improves OOD detection and how well the features align with ETF. ETF is only predictive of the last-layer's features' geometry if the training data is balanced (e.g., see [1]). Are all your training datasets balanced across classes?  If yes, would OOD detection under overparameterization behave differently if the training data were imbalanced?

[1] Thrampoulidis et al. "Imbalance trouble: Revisiting neural-collapse geometry." Advances in Neural Information Processing Systems 35 (2022)

---

> ### Author Response · Authors · 2024-11-18
>
> We would like to thank reviewer UnVj for their insightful comments and suggestions. We address the concerns raised in the following four points, presented separately in two parts due to character limit restrictions:
>
> ### **A1.** Significance of Theoretical Results (Q1):
>
> We understand the reviewer’s concern regarding the significance of Theorem 2. However, we would like to emphasize that, in Belkin et al. (2019), the authors focused on a regression problem where the outputs were noisy responses of a linear function. In contrast, our work extends these results to binary classification and out-of-distribution (OOD) detection problems. To adapt the regression problem to binary classification, we introduce an activation function $\Phi(\cdot)$, which maps the outputs of a linear regression model to the probability space of the labels. This is typically achieved using functions like softmax or sigmoid for $\Phi(\cdot)$. The challenge lies in incorporating this non-linear function, which leads to inequalities in our results (Theorems 1 and 2) under Assumption 4.1, as opposed to the equalities presented in Belkin et al.'s theorem, which did not include this assumption. We hope this explanation clarifies the theoretical contribution of our work.
>
> ### **A2.** Assumptions on OOD Distribution (Q2):
>
> Our objective in both Theorem 1 and Theorem 2 is to show that the expected risk and the expected OOD risk exhibit a double descent descent phenomenon that we associate with an infinite peak when the number of samples $n$ is close  to the number of features $p$ as noted in Remark 4.5 and Remark 4.6.
>
> To this end, Assumption 4.1  was introduced in Theorem 1 to ensure that the matrix $\Sigma$ is non-singular with $\lambda_{\min}(\Sigma)>0$. In particular, $\lambda_{\min}(\Sigma)>0$ guarantees that the lower bound of the expected risk is lower bounded by $c(n, p, \sigma)$. Since the expected risk is also upper bounded by $c(n, p, \sigma)$ and that $c(n, p, \sigma)=\infty$ when$n-1 \leq p \leq n+1$, we can deduce from the squeeze theorem that $E_X[R_\text{OOD}(\hat f)]=\infty$ when $n-1 \leq p \leq n+1$.
>
> In Theorem 2, we still want to obtain a lower bound for the expected OOD risk that depends on $c(n, p, \sigma)$ to show an infinite peak when $n$ is close to $p$. Assumption 4.1 is sufficient  to guarantee this result since the lower bounds exists even if $\lambda_{\min}(\Sigma_{OOD})=0$. This means that, regardless of the degree of distributional shift between the in-distribution (ID) and OOD distributions, an infinite peak occurs when $n-1 \leq p \leq n+1$.
>
> It is true that when $p < n-1$ or $p>n+1$, the degree of distributional shift between ID and OOD distributions influences how close the lower and upper bounds are to the expected OOD risk. However, our primary objective was not to provide accurate proxies for the expected risk and the expected OOD risk, but rather to highlight the existence of an infinite peak when $n$ is close to $p$ that we associate with the double descent phenomenon.

---

> ### Author Response · Authors · 2024-11-18
>
> ### **A3.** Sensitivity of NC1 Metric to Changes in Layer Dimensions (Q3):
>
> In our study, we sought to explore the correlation between Neural Collapse (NC) and model overparameterization. As part of this, we varied the width of the model to measure NC, in accordance with the double descent framework. As with the double descent tests, the size of the last-layer dimension increases dynamically alongside the intermediate layers. Altering only the intermediate layers could disrupt the architecture. While it might seem that the NC1 metric is sensitive to changes in the dimension $d$, empirical results from [1] suggest otherwise. Specifically, they showed that the overall model capacity, rather than the dimension $d$ of the last layer, is more important.
>  In Section 4.3 of [1], two models with the same capacity (ResNet-18) were trained on CIFAR-10, one with a standard last-layer size of $d=512$ and the other with a smaller $d=10$ but the same number of classes ($c=10$). Despite the difference in the size of the last layer, the NC measurement results were similar for both models, indicating that NC1 is more influenced by the learning dynamics across the entire architecture, with the model’s overall capacity being the key factor. Thus, we believe it is fair to compare NC1 values of different models in terms of their generalization, irrespective of minor differences in the last-layer size.
>
> ### **A4.** Heuristic Connections Between Overparameterization and OOD Detection (Q4):
>
> We appreciate the reviewer’s concern on this point. To clarify, all of our training datasets are balanced [2], as is common in the field, ensuring consistency with prior work [3,4,5,6,7]. This alignment with established theoretical and empirical studies facilitates a direct comparison with existing research. Furthermore, all the state-of-the-art OOD metrics we tested were developed with balanced datasets in mind, and imbalanced datasets were not considered in these methods. Although we did not address the case of imbalanced data in this study due to the substantial complexity involved, we acknowledge the relevance of this issue and will discuss it in our paper, particularly in Appendix section D.4. While we could conduct experiments on imbalanced datasets if required, we believe this would extend beyond the scope of our current work.
>
>
>
>
>
> ### References
>
> [1] Zhu, Z., Gong, Z., Wei, C., & Liu, J. (2021). "Geometric Analysis of Neural Collapse with Unconstrained Features." arXiv preprint arXiv:2105.02375.
>
> [2] Alex Krizhevsky. Learning multiple layers of features from tiny images. University of Toronto, 05 2012
>
> [3] D'Angelo, E., et al. (2021). Can Neural Nets Learn the Same Model Twice? Investigating Reproducibility and Double Descent from the Decision Boundary Perspective. NeurIPS 2021.
>
> [4] Yang, Z., et al. (2020). Rethinking Bias-Variance Trade-off for Generalization of Neural Networks. ICML 2020.
>
> [5] Unraveling the Enigma of Double Descent: An In-depth Analysis through the Lens of Learned Feature Space. ICLR 2024.
>
> [6] Belkin, M., et al. (2019). Reconciling Modern Machine Learning Practice and the Bias-Variance Tradeoff. PNAS 2019.
>
> [7] Nakkiran, P., et al. (2020). Deep Double Descent. ICLR 2020.

---

> > ### Author Response · Authors · 2024-11-22
> >
> > Dear reviewer UnVj,
> >
> > We thank you for serving as a reviewer for our paper. We would like to kindly remind you that the rebuttal period will conclude in less than a week. We would love to have your feedback to help us improve the paper, and we look forward to engaging in further discussion with you.

---

> > > ### Comment · Reviewer_UnVj · 2024-11-23
> > >
> > > Thank you for your response.
> > >
> > > **A1**: Since your theoretical analysis focuses on classification, does the paper discuss whether Assumption 4.1 holds when the logistic or softmax function is used as $\phi$? The only special case of $\phi$ I noticed being discussed is the linear function in Remark 4.3, which is the same setup as Belkin et al. (2020).
> > >
> > > **A4**: I understand that studying the impact of imbalances requires a separate investigation. I brought it up mainly because I found the connection between OOD performance and the ETF structure somewhat weak.
> > >
> > > To give a high-level argument: even if all the training features align with the ETF structure (with near-perfect collapse, NC1$\approx$0), this does not necessarily guarantee confident predictions on in-distribution data points. For confident predictions, the norm of the logits also needs to be large, as this ensures the softmax probabilities are close to zero for incorrect classes. It might be the case that in overparameterized models, both NC emergence and the increase in logits' norm co-occur, and both help show your observed correlation.
> > >
> > > Now, if we believe this connection is strong, I would expect that an imbalanced ID training set would disrupt the performance since the training features will no longer align with ETF. I was just curious about the extent to which imbalances might make OOD detection more challenging. However, I am not sure about how critical imbalances are as a topic of study in OOD detection research.
> > >
> > > In general, like reviewer ZoGQ, I believe the connection between the emergence of NC and generalization performance is still a debated topic, particularly since NC is primarily observed on the training set. While I understand the intuitive reasoning that increased data separation between classes could signal better generalization, this explanation alone may not fully capture the details.
> > >
> > > That said, I still find reporting these connections useful for future investigations, even if they are not concrete. I just believe it needs more careful phrasing.

---

> ### Author Response · Authors · 2024-11-23
>
> We want to thank the reviewer for his comment and his help. We highly appreciate being able to discuss with the reviewer.
>
> ---
>
> **A1**
>
> Thank you for pointing out the lack of clarity. **Assumption 4.1** actually holds for any strictly monotonically non-decreasing function, which includes the logistic function. To enhance the clarity and readability, we have specifically restricted **Assumption 4.1** to strictly monotonically non-decreasing activation function. The proof supporting this assumption is detailed in **Lemma 1** in the Appendix.
>
> ---
>
> **A2**
>
> We appreciate the reviewer’s comment regarding the importance of high logit values. While we agree with this observation, it is important to clarify that Neural Collapse (NC) focuses on different aspects. Specifically:
> - **NC1** states that the variability within the latent space of each class diminishes, leading to class-wise collapse.
> - **NC2** describes how the centroids of different classes move farther apart.
> - **NC4** suggests that the last fully connected layer behaves equivalently to a nearest neighbor classifier for predicting classes.
> To the best of our knowledge, there is no direct connection between these principles and logit or softmax values. However, as we mentioned, multiple papers [1,2,3,4] have empirically demonstrated that NC can be leveraged to improve OOD detection. It is worth noting, though, that there is currently no theoretical proof supporting this connection—only empirical evidence, which remains open to debate. That is why we totally agree with the reviewers (**ZoGQ**, and **UnVj** )  on that point.
> In light of this, we have carefully reviewed and rephrased all instances in our paper where Neural Collapse is discussed, ensuring that our claims are accurate and avoid overstatement.
>
> ---
>
> Thank you again for your thoughtful feedback, which has helped us improve our paper. We would be happy to address any additional comments or questions you may have.
>
> ### References
>
> [1] Zhang, J., Chen, Y., Jin, C., Zhu, L., & Gu, Y. (2024, April). EPA: Neural Collapse Inspired Robust Out-of-distribution Detector. In ICASSP 2024-2024 IEEE International Conference on Acoustics, Speech and Signal Processing (ICASSP) (pp. 6515-6519). IEEE.
>
> [2] Ammar, M. B., Belkhir, N., Popescu, S., Manzanera, A., & Franchi, G. (2023). NECO: NEural Collapse Based Out-of-distribution detection. arXiv preprint arXiv:2310.06823.
>
> [3] Haas, Jarrod, William Yolland, and Bernhard T. Rabus. "Linking Neural Collapse and L2 Normalization with Improved Out-of-Distribution Detection in Deep Neural Networks." Transactions on Machine Learning Research.
>
> [4] Zhao, Zhilin, and Longbing Cao. "Dual representation learning for out-of-distribution detection." Transactions on Machine Learning Research (2023).

---

> > ### Comment · Reviewer_UnVj · 2024-11-23
> >
> > Thanks for the additional clarification. I'll slightly increase my score.

---

### Official Review · Reviewer_wRDX · 2024-11-07

**Soundness:** 3
**Presentation:** 3
**Contribution:** 3
**Rating:** 8
**Confidence:** 4

**Summary:**

The paper investigates the influence of model complexity on Out-Of-Distribution (OOD) detection in machine learning models, with a specific focus on the double descent phenomenon. The authors propose a theoretical framework using Random Matrix Theory to explore OOD risks associated with binary least-squares classifiers on Gaussian data. The experimental analysis extends these theoretical insights across various neural network architectures, suggesting that overparameterization does not always enhance OOD detection.

**Strengths:**

- This paper provides a notable contribution by extending the concept of double descent, originally associated with generalization error, to OOD detection. This new perspective highlights a less explored aspect of model overparameterization, offering valuable insights into the robustness of overparameterized models.

- Theoretical contribution: The use of Random Matrix Theory to derive bounds for OOD risk offers valuable insights on understanding OOD detection in the over-paramterized regime.

- The experiments are quite comprehensive, particularly in the variety of OOD detection methods employed.

**Weaknesses:**

- Adding the label noise (Line 305) is a bit artificial for real-world applications, probably also leads to different phenomenon in OOD detection. Is there any result on noiseless experiments for understanding the effect of model size for OOD detection?

- Could the authors provide additional remarks on connecting the derived theoretical results and the empirical OOD detection methods? E.g., is the derived lower bound in Theorem related to one/several of the practical OOD detection methods?

**Questions:**

See Weaknesses.

(Update: increased score from 6 to 8 after author rebuttal)

---

> ### Author Response · Authors · 2024-11-18
>
> We thank reviewer WRDX for their insightful comments and suggestions. We address their concerns in the following two points, presented separately due to character limit restrictions:
>
> ###  **A1.** Label Noise and Real-World Applicability (W1):
>
> We appreciate the reviewer’s comment on label noise. It is important to note that the use of label noise is a well-established practice in the community for studying double descent [1,2,3,4,5], and we incorporated it into our experiments to align with these prior works. Label noise serves to amplify the effects we are studying and ensures our findings are comparable to existing theoretical and empirical studies.
>
> Nevertheless, we recognize the reviewer’s point about the value of noiseless experiments. In response, we have included additional experiments without label noise in Appendix Section D.5. Our observations indicate similar model behavior to previous studies, where error rates plateau around the interpolation threshold rather than peak, and we see a comparable trend in OOD detection performance. This reinforces the utility of label noise in clearly observing these phenomena.
>
> Moreover, we would like to emphasize that adding label noise aligns with real-world application scenarios, where datasets are often corrupted by noise, as demonstrated in the table below.
>
> | Dataset       | Estimated Label Noise | Source of Noise                                    |
> |---------------|-----------------------|----------------------------------------------------|
> | ImageNet [6]      | 6% - 10%              | Fine-grained classes, human annotation errors      |
> | Clothing1M [7]   | ~38%                  | Automated labels based on product descriptions     |
> | WebVision [8]  | 20% - 30%             | Keyword-based search results                       |
> | CIFAR-10/100 [9]  | Up to 5%              | Human error, ambiguous images                      |
> | Open Images [10]   | 5% - 15%              | Automated and human annotation errors              |

---

> ### Author Response · Authors · 2024-11-18
>
> ### **A2.** Connection Between Theoretical Results and Practical OOD Detection Methods (W2):
>
> We appreciate the reviewer’s suggestion to clarify the connection between our theoretical results and practical OOD detection methods. In this regard, we would like to highlight that the OOD risk expression $R_\text{OOD}(\hat f)$ (equation 5 in line 208) is closely related to confidence-based OOD detection methods, such as the Maximum softmax probability (MSP) or the MaxLogit metrics. In the binary classification scenario, $R_\text{OOD}(\hat f)$ is minimized when the binary classifier $\hat{f}(\cdot)$ is confident in its predictions over the in-distribution (ID) data and  uncertain  in its predictions over out-of-distribution (OOD) samples. Ideally,  \hat{f}(\mathbf{x}) should be close to $1$ or $0$ for $\mathbf{x} \in P_{\mathcal{X},\mathcal{Y}}$ and  $\hat{f}(\mathbf{x})$ should be close to  $0.5$ for $\mathbf{x} \in P_{\mathcal{X},\mathcal{Y}}^\text{OOD}$. The more the classifier deviates from this ideal setup, the higher the OOD risk. Specifically, if the classifier becomes more uncertain on ID samples (with $\hat{f}(\mathbf{x})$ approaching $0.5$  for $\mathbf{x} \in P_{\mathcal{X},\mathcal{Y}}$) and more confident on OOD samples (with $\hat{f}(\mathbf{x})$ moving further from $0.5$  $\mathbf{x} \in P_{\mathcal{X},\mathcal{Y}}^\text{OOD}$), the OOD risk increases. This behavior is akin to confidence-based OOD detection methods, such as the Maximum softmax probability (MSP) or the MaxLogit metrics, which measure the confidence of the classifier's predictions.
>
> This link between the OOD risk and the confidence-based OOD detection methods extends to the multi-class setting as highlighted by the remark 4.2 with use of the infinity norm ($\lVert \cdot \rVert_\infty$). Indeed, in the multi-class scenario, the OOD risk expression is minimized when all logits except the predicted class are null for ID data predictions. This corresponds to a perfectly calibrated classifier where the MSP method would yield high confidence in the correct class. For OOD data, the ideal scenario is a uniform distribution of logits, indicating maximum uncertainty.
>
> ### References
>
> [1] Nakkiran, P., et al. (2020). Deep Double Descent. ICLR 2020.
>
> [2] Belkin, M., et al. (2019). Reconciling Modern Machine Learning Practice and the Bias-Variance Tradeoff. PNAS 2019.
>
> [3] D'Angelo, E., et al. (2021). Can Neural Nets Learn the Same Model Twice? Investigating Reproducibility and Double Descent from the Decision Boundary Perspective. NeurIPS 2021.
>
> [4] Yang, Z., et al. (2020). Rethinking Bias-Variance Trade-off for Generalization of Neural Networks. ICML 2020.
>
> [5] Unraveling the Enigma of Double Descent: An In-depth Analysis through the Lens of Learned Feature Space. ICLR 2024.
>
> [6]Northcutt, C. G., et al. (2021). Pervasive Label Errors in Test Sets Destabilize Machine Learning Benchmarks. ICML 2021.
>
> [7]Xiao, T., et al. (2015). Learning from Massive Noisy Labeled Data for Image Classification. CVPR 2015.
>
> [8]Li, W., Wang, L., Li, W., Agustsson, E., & Van Gool, L. (2017). "WebVision Database: Visual Learning and Understanding from Web Data." arXiv preprint arXiv:1708.02862.
>
> [9]Song, H., Kim, M., & Lee, J.-G. (2020). "Learning from Noisy Labels with Deep Neural Networks: A Survey." arXiv preprint arXiv:2007.08199.
>
> [10]Kuznetsova, A., Rom, H., Alldrin, N., et al. (2020). "The Open Images Dataset V4: Unified Image Classification, Object Detection, and Visual Relationship Detection at Scale." International Journal of Computer Vision (IJCV).

---

> > ### Author Response · Authors · 2024-11-22
> >
> > Dear reviewer wRDX,
> >
> > We thank you for serving as a reviewer for our paper. We would like to kindly remind you that the rebuttal period will conclude in less than a week. We would love to have your feedback to help us improve the paper, and we look forward to engaging in further discussion with you.

---

> > > ### Author Response · Authors · 2024-11-27
> > >
> > > Dear reviewer wRDX,
> > >
> > > This is a gentle reminder that the rebuttal period ends in less than one week. We would appreciate the opportunity to engage in a discussion and address any questions you may have.

---

> ### Comment · Reviewer_wRDX · 2024-11-28
> **Response to Rebuttal**
>
> Thanks to the authors for the rebuttal. Both of my questions were well addressed, and I have increased my score.

---

### Author Response · Authors · 2024-11-18
**General answer**

Dear Area Chair and Reviewers,

We sincerely thank you for your thoughtful comments and questions regarding our work. We appreciate the time and effort you have invested in reviewing our research and acknowledge the recognition of the significant effort involved in this study.

Combining extensive experimental validation with theoretical analysis on the double descent phenomenon is indeed rare. While there are notable papers that explore the double descent phenomenon in modern settings (e.g., [1,7,8,9]) and provide theoretical insights ([2,3,4,5,6]), our work aims to bridge these perspectives in a unique and innovative manner, particularly by linking it to Out of Distribution (OOD) Detection. We understand the challenges in reviewing such interdisciplinary contributions and we are grateful for your efforts.

We would like to emphasize the originality of our study. To the best of our knowledge, no one has previously undertaken such a detailed investigation of the link between OOD and Double Descent. We believe that our findings offer valuable contributions to the field.

In response to your feedback, we have addressed all the reviewers' comments and are committed to improving the clarity and quality of the manuscript. A revised version of the paper, incorporating the required corrections, will be submitted by Wednesday.

Thank you once again for your constructive feedback and engagement. We look forward to continuing this conversation with reviewers to further refine our work.

Best regards,


## References

[1] Nakkiran, P., et al. (2020). Deep Double Descent. ICLR 2020.

[2] Belkin, M., Hsu, D., Ma, S., & Mandal, S. (2019). Reconciling modern machine-learning practice and the classical bias–variance trade-off. Proceedings of the National Academy of Sciences, 116(32), 15849-15854.

[3] Louart, C., Liao, Z., & Couillet, R. (2018). A random matrix approach to neural networks. The Annals of Applied Probability, 28(2), 1190-1248.

[4] Liao, Z., Couillet, R., & Mahoney, M. W. (2020). A random matrix analysis of random fourier features: beyond the gaussian kernel, a precise phase transition, and the corresponding double descent. Advances in Neural Information Processing Systems, 33, 13939-13950.

[5] Belkin, M., Hsu, D., & Xu, J. (2020). Two models of double descent for weak features. SIAM Journal on Mathematics of Data Science, 2(4), 1167-1180.

[6]  Liu, F., Liao, Z., & Suykens, J. (2021, March). Kernel regression in high dimensions: Refined analysis beyond double descent. In International Conference on Artificial Intelligence and Statistics (pp. 649-657). PMLR.

[7] D'Angelo, E., et al. (2021). Can Neural Nets Learn the Same Model Twice? Investigating Reproducibility and Double Descent from the Decision Boundary Perspective. NeurIPS 2021.

[8] Unraveling the Enigma of Double Descent: An In-depth Analysis through the Lens of Learned Feature Space. ICLR 2024.

[9] Yang, Z., et al. (2020). Rethinking Bias-Variance Trade-off for Generalization of Neural Networks. ICML 2020.

---

> ### Author Response · Authors · 2024-11-20
>
> Dear Area Chair and Reviewers,
>
> We sincerely thank you for your constructive feedback, which has greatly helped improve the quality of our paper. We have carefully addressed all the comments and performed additional experiments. The updated manuscript incorporates these changes, and we have ensured that the results, analysis, and references are emphasized throughout the revised submission with a different color (orange). Below, we summarize the key updates:
>
> ## Clarification on the math formulas
> * $z$ dependence on $\\mathbf{x}$ is highlighted in the main paper with its values clearly illustrated (Remarques 4.1 and 4.2).
> * Our assumption on $\\hat{\\mathbf{w}}$ being close to $\\mathbf{w}^*$ is integrated to the main manuscript (Assumption 4.2).
>
>
> ## Significance of our theoretical framework
> * The connection between our theoretical results and practical OOD detection methods is clarified, with added experiments in section D.4 highlighting this.
> * The significance of our derived theorems is clarified, and the motivation behind its components is justified.
> * Concerns regarding the pertinence and utility of our model and data assumptions have been addressed.
> * Additional explanations have been provided for the reviewers regarding the connections between the different parts of our theoretical framework, and to clarify its various parts and components.
>
>
> ## Motivation and experiments
> * Experiments without label-noise have been added in section D.5. Its real-word nature and the utility of its integration are addressed.
> * The concerns about neural collapse (NC) sensitivity to the last-layer dimension have been addressed.
>
> * The motivations and implications of our theoretical and empirical work were explained.
> * The utility behind our OOD risk formula is explained, with the added experiments in section D.4 illustrating its links to confidence-based OOD detection methods.
> * Concerns regarding the possible experimental limitation from varying the model width are addressed.
>
> Attached, you will find a revised version of the paper incorporating all changes in orange color.
> We hope these revisions fully address the concerns raised and clarify our contributions. Please feel free to let us know if there are further issues that require our attention. Thank you for your time and effort in reviewing our work.
>
> Sincerely,
> Corresponding authors

---

> ### Author Response · Authors · 2024-11-27
>
> Dear Area Chair and Reviewers,
>
> We sincerely thank you for your constructive feedback. Thanks to you we have performed the following update :
>
> ### 1. Assumption 4.1
>
> Based on your feedback, **we have modified Assumption 4.1** to enhance clarity and readability. The former version of Assumption 4.1 about the nonsingular property of the matrix Sigma, which held for any strictly monotonically non-decreasing function, including the logistic function. To improve clarity, we have specifically restricted Assumption 4.1 to monotonically non-decreasing activation functions. The proof supporting this revised assumption is detailed in Lemma 4.1 in the Appendix.
>
> ### 2. Assumption 4.2
>
> Using the new version of Assumption 4.1, we have been able to **remove Assumption 4.2**, introduced in our previous reply, which required $\\hat{w}$ to be close to $w^*$. With the revised Assumption 4.1 and the application of the mean-value theorem, we can now derive the results of Theorem 1 without relying on the Taylor expansion or assumptions about $\\hat{w}$ being close to $w^*$. Details can be found in the proof of Theorem 1 in the Appendix.
>
> ### 3. OOD Risk
>
> We have performed **experiments with the out-of-distribution (OOD) risk** in a practical setting and observed that it also exhibits a double descent phenomenon in real-world scenarios. Figures can be found in Appendix D.
>
> Thank you again for your time and effort in reviewing our work. Do you have any additional comments or suggestions?
>
> Best regards,
>
> Authors

---

### Meta-Review · Area_Chair_MDkg · 2024-12-21

**Metareview:**

This paper investigates a double descent phenomenon for out-of-distribution (OOD) detection. Similar to Belkin's original double descent work (2019, 2020), the authors theoretically demonstrate the phenomenon in least squares classifiers with strict assumptions on the training data distribution. The authors back up the theory in this simplified setting with experiments with neural networks that identify a double descent curve corresponding to the under-parameterized and over-parameterized regimes.

Connecting out-of-distribution detection with double descent is novel and will likely be of interest to the OOD community. The general finding may not be surprising given the strong connection between Belkin's proof techniques and this paper. While the paper's theory aims to identify a peak at the interpolation threshold, this paper would be significantly strengthened if the authors made assumptions about the OOD distribution, which would allow for a theoretical characterization of the double descent curve away from the peak.

During the rebuttal period, the authors made non-trivial changes to the proofs in the paper, both to simplify assumptions and to correct errors pointed out by the reviewers. While all of the changes seem benign, the fact that such errors existed in the first place gives me pause, especially since the theorems generally follow the pattern of Belkin et al. (2020). I suggest another round of careful review to ensure the changes are theoretically sound.

Altogether, this paper could benefit from another round of reviewing. This would ensure that the theory is correct while allowing the authors to potentially extend their analysis to incorporate the OOD distribution, ultimately leading to a stronger and more impactful paper.

**Additional Comments On Reviewer Discussion:**

One reviewer had a critical view of the paper, citing (1) that the theoretical setting was overly simple and (2) errors in the proofs. I discounted (1) since many works on neural network theory make similar assumptions. (Beyond making analysis theoretically tractable, these assumptions aim to demonstrate that phenomena like double descent are consequences of high dimensional statistics rather than behaviour specific to neural networks.) Additionally, some of (2) pointed out only minor errors. Nevertheless, the authors' revisions amounted to substantial changes to the theory that could benefit from more eyes. In addition, minor errors (e.g., in Lemma 4.1) make me think that the theory could use more scrutiny, mainly because the proofs largely follow the framework set by Belkin et al.

Two reviewers mentioned that the theorems may not be very novel because they were largely based on Belkin et al. (2020). I think the author's focus on the OOD setting alleviates concerns about novelty.

Finally, one reviewer brought up the point that there were no assumptions about the OOD distribution, to which the authors responded that the primary objective was to highlight an OOD risk peak at the interpolation threshold. While I agree that the author's assumptions were sufficient for this goal, I (implicitly) agree with the reviewer that there would be an opportunity for OOD distribution-specific analysis if one made assumptions about the distribution, which would be of more interest to the community.

---

### Decision · Program_Chairs · 2025-01-22

Reject